# Differential response of plant transpiration to uptake of rainwater-recharged soil water for dominant tree species in the semiarid Loess Plateau

**Yakun Tang [1], Lina Wang [1], Yongqiang Yu [1], Dongxu Lu [1, 2]**

[1] State Key Laboratory of Soil Erosion and Dryland Farming on the Loess Plateau, Institute of Soil and Water Conservation, Northwest A&F University, Yangling, 712100, China

[2] State Key Laboratory of Soil Erosion and Dryland Farming on the Loess Plateau, Institute of Soil and Water Conservation, Chinese Academy of Sciences, Ministry of Water Resources, Yangling, 712100, China

*Correspondence to*: Yakun Tang (t453500@163.com)

**Abstract** Whether uptake of rainwater-recharged soil water (RRS) can increase plant transpiration in response to rainfall pulses requires investigation to evaluate plant adaptability, especially in water limited regions where rainwater is the only replenishable soil water source. In this study, the water sources from RRS and three soil layers, predawn ($\Psi_{pd}$), midday ($\Psi_m$) and gradient ($\Psi_{pd}-\Psi_m$) of leaf water potential, and plant transpiration in response to rainfall pulses were analyzed for two dominant tree species, *Hippophae rhamnoides* subsp. *sinensis* and *Populus tomentosa*, in pure and mixed plantations during the growing period (June–September). Mixed afforestation significantly enhanced $\Psi_{pd} - \Psi_m$, RRS uptake proportion (RUP), relative response of daily normalized sap flow ($SF_R$), and reduced the water source proportion from the deep soil layer (100–200 cm) for both species (P < 0.05). In pure and mixed plantations, the large $\Psi_{pd} - \Psi_m$ was consistent with high $SF_R$ for *H. rhamnoides*, and the small $\Psi_{pd} - \Psi_m$ was consistent with low $SF_R$ for *P. tomentosa*, in response to rainfall pulses. Therefore, *H. rhamnoides* and *P. tomentosa* exhibited anisohydric and isohydric behavior, respectively, and the former plant species was more sensitive to rainfall pulses than *P. tomentosa*. Furthermore, in pure plantations, the $SF_R$ was significantly affected by RUP and $\Psi_{pd} - \Psi_m$ for *H. rhamnoides*, and was

significantly influenced by $\Psi_{pd} - \Psi_m$ for *P. tomentosa* (P < 0.05). However, the $SF_R$ was significantly influenced by RUP and $\Psi_{pd} - \Psi_m$ for both species in the mixed plantation. These results indicate that mixed afforestation enhanced the influence of RRS uptake to plant transpiration for these different rainfall pulse sensitive plants. This study provides insights into suitable plantation species selection and management considering the link between RRS uptake and plant transpiration in water limited regions.

**Keywords:** Leaf water potential; Loess Plateau; Plant transpiration; Rainwater-recharged soil water; Water stable isotope

## 1 Introduction

Rainwater-recharged soil water (RRS) uptake by plants and plant transpiration in response to rainfall pulses drive the survival of plant species and ecosystem ecohydrological processes, especially in arid and semiarid regions where rainwater is the only replenishable soil water source (Berkelhammer et al., 2020; Gebauer and Ehleringer, 2000; West et al., 2012). Generally, RRS uptake after a rainfall pulse refers to the root uptake of soil water that was recharged by recent rainwater, and can be quantified through water stable isotopes (Cheng et al., 2006; Meier et al., 2018). The variability and intermittency of rainfall, which plays an important role in plant water uptake and transpiration (Swaffer et al., 2014; Wang et al., 2020a), have been predicted to increase in water limited regions (Mendham et al., 2011). Clarifying the influence of RRS uptake on plant transpiration after rainfall pulses is essential to understand the process of plant species adaptation in water limited regions (Meier et al., 2018; Tfwala et al., 2019).

The RRS uptake by plant is expected to increase plant transpiration after a rainfall pulse (Cheng et al., 2006; Liu et al., 2019). However, the uptake of RRS may also be mainly used to reduce the water uptake from deep soil layers or decrease the risk of cavitation in stems for some plant species (Plaut et al., 2013; Tfwala et al., 2019). The controversial rainfall pulse response between RRS uptake and plant transpiration may be mainly attributed to an inconsistent influence of plant leaf physiological characteristics (West et al., 2007), root morphology adjustment (Wang et al., 2020a), or environmental

conditions (Tfwala et al., 2019) on these two water processes. Generally, plant transpiration is observed to increase after rainfall pulses for plants with shallow (Liu et al., 2019) or dimorphic root systems (Swaffer et al., 2014); meanwhile, no increase or a decrease in plant transpiration is observed for plants with deep rooting systems (West et al., 2012). However, regardless of the root distribution, the plant leaf water potential gradient ($\Psi_{pd} - \Psi_m$, the difference between predawn ($\Psi_{pd}$) and midday ($\Psi_m$) leaf water potential) has been observed to regulate plant transpiration after rainfall pulses (Kumagai and Porporato, 2012; Liu et al., 2019). For example, plant species that show isohydric behavior generally maintain relative small $\Psi_{pd} - \Psi_m$ to protect stem hydraulic architecture, which is vulnerable to cavitation and limited plant transpiration under varied soil water conditions (Franks et al., 2007; McDowell et al., 2008). However, plant species that show anisohydric behavior are generally less vulnerable to cavitation and adopt relative large $\Psi_{pd} - \Psi_m$ to allow high plant transpiration after rainfall pulses (West et al., 2007; Klein, 2014; Ding et al., 2021). Thus, taking into consideration plant leaf physiological or root morphological parameters could help in understanding the mechanisms underlying the influence of RRS uptake on plant transpiration in response to rainfall pulses.

Uptake of contrasting water sources between coexisting species usually shows water source separation and can minimize water source competition (Munoz-Villers et al., 2020; Silvertown et al., 2015); however, overlapping water sources among plant species may lead to competition in arid and semiarid regions (Tang et al., 2019; Yang et al., 2020). Rainfall pulses have been observed to relieve or eliminate water competition among coexisting species and thus maintain or increase plant transpiration in some water limited regions (Wang et al., 2020a; Tfwala et al., 2019). Meanwhile, plant species with strong RRS uptake ability generally exhibit more competitiveness than coexisting weak RRS uptake ability species (Stahl et al., 2013; West et al., 2012). However, Liu et al. (2019) attribute opposite RRS uptake ability to the stable coexistence of species in mixed plantations in semiarid regions, where the rainfall events are variable and less RRS taken up by one of the coexisting plant species. In addition, coexisting species may also cope with or minimize water resource competition through plant leaf water potential or root distribution adjustment (Chen et al., 2015; Silvertown et al., 2015). It is still unclear whether these adjustments could influence the RRS uptake and plant transpiration for coexisting species

in water limited regions.

The "Grain for Green project" has increased vegetation coverage by 25% in the Loess Plateau through afforestation activities since the 1990s, to deal with vegetation degradation and water and soil loss (Tang et al., 2019; Wu et al., 2021). *Hippophae rhamnoides* subsp. *sinensis* and *Populus tomentosa* are typical deciduous broadleaved tree species, with similar leaf expansion (April) and falling (November) periods, and occupy nearly 30% of the plantation area in this region (Liu et al., 2017; Tang et al., 2019). Our previous study indicated that *H. rhamnoides* generally took up soil water from 0−40 cm or > 100 cm soil depths and adopted large value and variation of $\Psi_{pd}-\Psi_m$ to cope with varied soil water conditions in this region (Tang et al., 2019). Meanwhile, *P. tomentosa* generally took up soil water from > 100 cm soil depth throughout the growing season under varied soil water conditions (Xi et al., 2013). In addition, mixed plantations of these two species were widely promoted by local government due to the higher soil and water conservation capacity than pure plantations in the original afforestation stage (Tang et al., 2019; Wang et al., 2020a). Tang et al. (2019) also suggested that mixed afforestation with *Ulmus pumila*, a deciduous broadleaved tree species with similar leaf growth phenology to *H. rhamnoides*, increased the water source from 0−40 cm soil depth and enlarged the $\Psi_{pd}-\Psi_m$ for *H. rhamnoides* compared with these values for this species in pure plantation. Furthermore, rainfall events have obvious seasonal variability and the rainfall amount is generally lower than the reference evapotranspiration ($ET_0$) during the plant growth period in this semiarid region (Zhang et al., 2017). The imbalance between rainwater input and plant water demand may weaken the sustainability of plantations with further plant growth (Jia et al., 2020; Wu et al., 2021). To understand the adaptation of plantation species in this study, the plant transpiration, water sources from RRS and different soil layers, and plant leaf water potential for *H. rhamnoides* and *P. tomentosa* in pure and mixed plantations were analyzed. The specific objectives were as follow: (1) to investigate the influence of RRS uptake and leaf water potential on plant transpiration after rainfall events in pure plantation, and (2) to assess the mixed afforestation effect on these influences. Based on variations of plant water uptake from different soil layers and leaf water potential for these species in Xi et al. (2013) and Tang et al. (2019), we hypothesize that (1) the influence of RRS uptake and leaf water potential on plant transpiration may

differ for these species in pure plantations, and (2) these influences may differ for specific species in

pure and mixed plantations.

## 2 Materials and methods

### 2.1 Study site

The study was conducted in the Ansai Ecological Station in the semiarid Loess Plateau (36.55 °N,

109.16 °E, 1221 m above sea level), Northern China (Fig. S1). The study area has a semiarid continental

climate. The annual average (mean $\pm$ SD) rainfall amount and air temperature are 454.8 $\pm$ 105.2 mm

and 10.6 $\pm$ 0.4 °C (2000–2017), respectively, with higher monthly rainfall amount and air temperature

generally occurring during June–September and lower values during the other months (Fig. S1).

Three adjacent plantations were chosen for the study: pure *H. rhamnoides* plantation, pure *P.*

*tomentosa* plantation, and *H. rhamnoides–P. tomentosa* mixed plantation (Fig. S1), with corresponding

plantation slope of 5.2, 4.5, and 5.5 °. All plantations were planted on abandoned grassland in 2004,

where *Bothriochloa ischaemum* was the dominant herbaceous species at that time. Three adjacent plots

were selected (16 m $\times$ 10 m) for each plantation type, and no soil and water conservation measure was

conducted in the plantations. In pure plantations, the original planted spacing for each individual plant

was 2.0 m $\times$ 2.0 m. In the mixed plantation, *P. tomentosa* was originally planted between the 4.0 m gaps

in rows of *H. rhamnoides*, each individual plant was also spaced 2.0 m $\times$ 2.0 m. Based on a survey

performed in July 2018, in pure plantations, the average tree trunk diameter (at 1.2 m height above the

ground) and height were 50.5 $\pm$ 3.6 mm and 4.11 $\pm$ 0.81 m for *H. rhamnoides*, respectively, and the

corresponding values were 52 $\pm$ 4.6 mm and 4.05 $\pm$ 0.63 m for *P. tomentosa*. Meanwhile, in mixed

plantations, the average trunk diameter and tree height were 51.3 $\pm$ 2.9 mm and 4.49 $\pm$ 0.7 m for *H.*

*rhamnoides*, respectively, and the corresponding values were 56.3 $\pm$ 3.8 mm and 4.23 $\pm$ 0.79 m for *P.*

*tomentosa*. *B. ischaemum* and *Glycyrrhiza uralensis* were the dominant herbaceous species in *H.*

*rhamnoides* and *P. tomentosa* pure plantations, respectively; meanwhile, *B. ischaemum* was dominant in

the mixed plantation. Based on an experiment conducted in July 2018 using the cutting ring (Wu et al.,

2016), constant water head (Reynolds et al., 2002), and centrifugation (Qiao et al., 2019) method, the

soil bulk density, total porosity, saturated hydraulic conductivity, field capacity, and permanent wilting point at 0–200 cm soil depth were found to be similar in the three plantations. The average soil bulk density was $1.38 \pm 0.08$, $1.35 \pm 0.11$, and $1.35 \pm 0.09$ g cm$^{-3}$ for pure *H. rhamnoides*, pure *P. tomentosa*, and mixed plantations, respectively, and corresponding soil total porosity was $48.2 \pm 0.6$, $48.1 \pm 0.4$,

and $48.1 \pm 0.7\%$. The average soil saturated hydraulic conductivity was $0.44 \pm 0.08$, $0.46 \pm 0.09$, and $0.46 \pm 0.08$ mm min$^{-1}$ for pure *H. rhamnoides*, pure *P. tomentosa*, and mixed plantations, respectively. The average field capacity was $0.26 \pm 0.02$, $0.25 \pm 0.03$, and $0.25 \pm 0.02$ m$^3$ m$^{-3}$ for pure *H. rhamnoides*, pure *P. tomentosa*, and mixed plantations, respectively, and corresponding permanent wilting point was $0.06 \pm 0.02$, $0.06 \pm 0.01$, and $0.06 \pm 0.02$ m$^3$ m$^{-3}$. The soil is characterized as a silt loam soil according

to United States Department of Agriculture soil taxonomy, with average sand (2–0.05 mm), silt (0.05–0.002 mm), and clay (<0.002 mm) compositions were $24.3 \pm 1.3$, $63.2 \pm 1.1$, and $12.5 \pm 2.1\%$, respectively, for three plantation types at 0–200 cm soil depth. These compositions were determined using a Mastersize 2000 (Malvern Instruments Ltd., UK).

**2.2 Environmental parameter measurements and $ET_0$ calculation**

Net radiation ($R_n$, CNR4, Kipp &Zone Inc., Netherlands), atmospheric pressure (CS105, Vaisala Inc., Finland), air temperature ($T_a$) and relative humidity (HMP45D, Vaisala Inc.), and wind velocity (Ws, A100R, Vector Inc., UK) were measured using a weather station nearly 500 m from the research plots. Soil heat flux (G) and rainfall amount were measured at 5 cm below ground using two HFT-3 plates

(Campbell Scientific Inc., USA) and a TE525 rain gauge (Campbell Scientific Inc.), respectively. At each plot, soil water content (SW) was measured at 5, 20, 50, 100, 150, and 200 cm below ground ($SW_{5cm}$, $SW_{20cm}$, $SW_{50cm}$, $SW_{100cm}$, $SW_{150cm}$, and $SW_{200cm}$) by CS615 probes (Campbell Scientific Inc.). All these parameters were measured and stored at 30 min interval by a CR3000 datalogger (Campbell Scientific Inc.).

$ET_0$, considering both aerodynamic characteristics and energy balance, was used to indicate atmospheric evaporative demand (Allen et al., 1998):

$$ET_0 = (0.408 \times s \times (R_n - G) + \gamma \times \frac{900}{T_a + 273} \times W_s \times VPD) / (s + \gamma \times (1 + 0.34 \times W_s)) \tag{1}$$

where $\gamma$, $s$, and $VPD$ are the psychrometric constant (kPa K$^{-1}$), the slope between saturation vapor pressure and air temperature (kPa K$^{-1}$), and vapor pressure deficit (kPa), respectively. The units of $R_n$ and $G$ are MJ m$^{-2}$ day$^{-1}$, and of $W_s$ is m s$^{-1}$.

**2.3 Sap flow observation**

Three standard individuals, with approximately mean height and trunk diameter, for specific species were chosen in each of the nine plots (Table S1). In each plot in the mixed plantation, three individuals of *H. rhamnoides* were chosen firstly, then a neighboring *P. tomentosa* individual was selected at approximately 2 m distance from each chosen *H. rhamnoides* individual. The sap flow was monitored by a pair of Granier-type thermal dissipation probes (TDPs) 10 mm in length and 2 mm in diameter in 36 selected individuals. During the plant growing season and ranging from 11 May (DOY 132) to 30 September (DOY 273) in 2018, the 30 s original and 30 min average sap flow values were monitored using a CR3000 data logger (Campbell Scientific Inc.). Waterproof silicone and aluminum foil were used to avoid the impact of the external environment on and physical damage to TDPs (Du et al., 2011). The standard sap flow density ($F_d$, ml m$^{-2}$ s$^{-1}$) was calculated as follows (Granier, 1987):

$$F_d = 119((\Delta t_{max} - \Delta t) / \Delta t)^{1.231} \tag{2}$$

where $\Delta t$ and $\Delta t_{max}$ are the temperature difference of heated and unheated probes at 30 min intervals and the maximum $\Delta t$ in each day, respectively.

Steppe et al. (2010) suggested that $F_d$ should have a species-specific calibration to validate Eq. (2). Meanwhile, the possibility of underestimating the $F_d$ value with the Granier-type thermal dissipation method (Du et al., 2011) should be considered when the whole tree transpiration is calculated. However, with the lack of species-specific calibration for Eq. (2) in the present study, the daily normalized $F_d$ for each replicate individual was calculated as the index of plant transpiration, through dividing $F_d$ by the maximum value from DOY 132 to DOY 273. Thus, each monitored individual had a maximum daily normalized $F_d$ of 1. In each plantation type, the average daily normalized $F_d$ for specific species was

calculated in each plot to determine the plant transpiration characteristics rather than the absolute transpiration amount (Du et al., 2011).


## 2.4 Rainwater, plant stem, soil water, and leaf sample collection and measurement

From April to October 2018, at the end of each rainfall event, 19 rainwater samples were collected immediately using a polyethylene rain gauge cylinder placed in the weather station, and stored at 4 ℃. A funnel containing a ping-pang ball was connected at the top of rain gauge cylinder to avoid rainwater

evaporation (Yang et al., 2015). To avoid the influence of sample collection on sap flow observation, one standard individual for the specific species nearby each sap flow monitored individual was selected for plant stem and soil water collection. In the mixed plantation, the distance was approximately 2 m between the selected *H. rhamnoides* and *P. tomentosa* standard individuals in each plot for sample collection. For plant stem and soil water collection, 5 rainfall events were selected: 3.4 mm (DOY 194),

7.9 mm (DOY 265), 15.4 mm (DOY 249), 24 mm (DOY 204), and 35.2 mm (DOY 155–156). These rainfall events were selected with an interpulse period longer than 7 days to eliminate the potential influence of the previous rainfall event. In addition, no runoff was generated during the selected rainfall events in three plantations according to the simulated result from the HYDRUS-1D model (Appendix A), which is based on the Richards' equation to describe soil water dynamics (Šimůnek et al., 2008). This

model has been widely used to simulate the runoff and soil water dynamics in the Loess Plateau (Yi and Fan, 2016; Bai et al., 2020; Wang et al., 2020b).

At each of successive three days after every selected rainfall event, one suberized stem after removing the bark was collected at midday (11:30–13:30) for each standard individual. Meanwhile, approximately 0.5 m around the stem of each standard individual in the pure plantations and at the

middle between two species in the mixed plantation, one soil core at seven depths (0–10, 10–20, 20–30, 30–50, 50–100, 100–150, and 150–200 cm) was collected through soil drilling. The suberized stem and collected soil samples were placed into glass bottles. These bottles were sealed with parafilm and stored at −15 ℃. On the same day as plant stem and soil sample collections, one leaf was selected from each sap flow monitored individual for leaf water potential measurement. The $\Psi_{pd}$ and $\Psi_{m}$ were measured by

a PMS1515D analyzer (PMS Instrument, Corvallis Inc., OR, USA) at predawn (4:30–5:30) and midday (11:20–12:40), respectively.

All the plant stem, soil, and leaf samples collected on the first day after a rainfall pulse were used for analysis, with the detailed given in section "2.6 Statistical analysis". There were 180 stem and 945 soil samples for water extraction, and 180 leaf samples for $\Psi_{pd}$ and $\Psi_m$ measurement, respectively.

A vacuum line (LI-2100, LICA Inc., China) was used to extract water from soil samples and plant stems. The water isotopic values of rainwater, soil samples, and plant stems were determined using a DLT-100 water isotope analyzer (LGR Inc., USA), with accuracy of $\pm$ 0.1 ($\delta^{18}O$) and $\pm$ 0.3 ‰ ($\delta D$). The potential influence of organic matter on water isotopic values produced during water extraction from stems was eliminated using the method of Yang et al. (2015). The isotopic values (‰) were calculated as follows:

$$\delta^{18}O(D) = (R_{sample} - R_{standard}) / R_{standard} \times 1000 ‰$$

(3)

where $R_{standard}$ and $R_{sample}$ indicate the $^{18}O/^{16}O$ (D/H) molar ratios of water sample relative to the Vienna Standard Mean Ocean Water, respectively. The average water $\delta^{18}O$ and $\delta D$ of plant stems for specific species and corresponding soil samples was calculated in each plot for further analysis.

## 2.5 Plant fine root investigation

In August 2018, 4 soil cores were dug around each selected standard individual for plant stem and soil water collection, through a soil drill with diameter 20 cm to investigate plant fine roots. The collected soil depths were 0–10, 10–20, 20–30, 30–50, 50–70, 70–100, 100–130, 130–150, 150–200 cm, with approximately 0.5 m around the stem of each species standard individual. The sum of root samples for 4 soil cores at each soil depth for each selected standard individual was used for fine root distribution analysis, giving 324 fine root samples. WinRHIZO (Regent Instruments Inc., Quebec, Canada) was used to determine the fine root (diameter < 2 mm) surface area at each soil depth. The average fine root surface area for specific species at each soil depth was calculated in each plot for further analysis.

## 2.6 Statistical analysis

**2.6.1 Calculation of plant transpiration and leaf water potential in response to rainfall pulse**

In the present study, the first day after rainfall was the maximum normalized $F_d$ within 3 days for *H. rhamnoides* and *P. tomentosa* in both plantation types, except that the second day after 24 and 35.2 mm was the maximum normalized $F_d$ for *P. tomentosa* in pure plantation. However, for *P. tomentosa* in pure plantation, there was no significant difference (P > 0.05) in diurnal sap flow between the first and second day after each of these two rainfall events based on independent-sample *t*-test (Fig. S2). Therefore, the normalized $F_d$ on the first day after each selected rainfall amount was used in Eq. (4) to calculate the relative response of daily normalized $F_d$ (SF$_R$, %) to rainfall pulses:

$$SF_R = ((X_{after} - X_{before})/X_{before}) \times 100\%  \tag{4}$$

where $X_{after}$ and $X_{before}$ are the normalized $F_d$ on the first day after and on the day before the rainfall event, respectively.

Meanwhile, none of $\Psi_{pd}$, $\Psi_m$, or $\Psi_{pd}-\Psi_m$ showed significant differences between the first and second day after each rainfall event (P > 0.05) for these two species in both plantation types (Table S2). On the first day after each rainfall event, the average $\Psi_{pd}$, $\Psi_m$, and $\Psi_{pd}-\Psi_m$ for specific plant species in each plot were used in the following analysis to illustrate the influence of leaf water potential on SF$_R$ in response to rainfall pulses.

**2.6.2 Calculation of RRS uptake proportion and water sources from different soil layers**

The RRS uptake proportion (RUP, %) after a recent rainfall pulse for plant was calculated as the proportion of rainwater in plant stem as follows (Cheng et al., 2006):

$$\delta^{18}O\,(D)_P = RUP \times \delta^{18}O\,(D)_{rain} + (1-RUP) \times \delta^{18}O\,(D)_{swb}  \tag{5}$$

$$RUP = (\delta^{18}O(D)_p - \delta^{18}O(D)_{swb})/(\delta^{18}O(D)_{swa} - \delta^{18}O(D)_{swb}) \times 100\%  \tag{6}$$

where $\delta^{18}O(D)_{rain}$ and $\delta^{18}O(D)_p$ are the isotopic values for rainwater and plant stem after rainfall, respectively; $\delta^{18}O(D)_{swb}$ and $\delta^{18}O(D)_{swa}$ are the isotopic values of soil water immediately before and after rainfall, respectively. The Eq. (6) is derived through the linear mixing model for water isotopic value in plant stem after rainfall in Eq. (5). The RUP was the average value calculated in Eq. (6) based on $\delta^{18}O$ and $\delta D$, respectively, for specific plant species in each plot.

Equations (5) and (6) are based on the assumption that little or no soil water is lost through evaporation. Thus, in this study, only the values of plant stem and soil water collected on the first day immediately after rainfall were used, and only the RUP on the first day after each rainfall event was calculated.

In this study, the $\delta^{18}O(D)_{swb}$ could not be directly and accurately determined through soil water sample collection, due to unpredictable natural rainfall events. A linear mixed model can be used to calculate the $\delta^{18}O(D)_{swb}$, based on the isotopic values for rainwater and soil water after rainfall, and soil depth interval weighted SW before ($sw_b$, $m^3\ m^{-3}$) and after ($SW_a$, $m^3\ m^{-3}$) rainfall:

$$\delta^{18}O(D)_{swb} = SW_b/SW_a \times \delta^{18}O(D)_{swa} + (1 - SW_b/SW_a) \times \delta^{18}O(D)_{rain} \tag{7}$$

In addition to RUP, the water uptake proportions from different soil layers were calculated on the first day after a rainfall event using the MixSIR program, to complement the analysis of plant water source variations in response to rainfall pulses. The RUP method only calculated the proportion of recent rainwater in the plant stem and did not include soil water before the recent rainfall event (Gebauer and Ehleringer, 2000; Cheng et al., 2006). The water taken up from different soil layers by the plant is a mixture of soil water before the recent rainfall event and the recent rainwater.

Firstly, the seven soil depths (0–10, 10–20, 20–30, 30–50, 50–100, 100–150, and 150–200 cm) were combined into three soil layers (shallow, middle, and deep) based on the variation of soil water $\delta^{18}O$ and $\delta D$ and SW, to facilitate water source comparison (Wang et al., 2020a; Zhao et al., 2021). The shallow (0–30 cm) soil layer was vulnerable to rainfall, which exhibited high soil water $\delta^{18}O$ and $\delta D$ values and large water isotope and SW variations (Table S3, Fig. S3). The middle (30–100 cm) soil layer was less vulnerable to rainfall, with moderate soil water isotope values and water isotope and SW variations. The deep (100–200 cm) soil layer was relative stable, with lower soil water isotope values and smaller water isotope and SW variations compared with shallow and middle soil layers. In addition, based on one-way ANOVA followed by post hoc Tukey's test, significant difference ($P < 0.05$) was observed in soil water $\delta^{18}O$ and $\delta D$ among three soil layers in each plot. Then, the water uptake proportions from three soil layers were calculated using the MixSIR program (Moore and Semmens,

2008), with model input parameters being the average $\delta^{18}O$ and $\delta D$ values in plant stem water and soil water at each soil layer in each plot. The SD for $\delta^{18}O$ and $\delta D$ at each soil layer was also used to accommodate the uncertainties of these values. No fractionation was considered during water source uptake by these plant roots because none of the plants exhibited xerophytic or halophytic characteristics. Ellsworth and Williams (2007) and Moore and Semmens (2008) suggested that a water stable isotope fractionation generally occurred during root uptake by xerophytic or halophytic plants.

**2.6.3 Statistical analysis for plant transpiration, water sources, and leaf water potential in response to rainfall pulse**

A repeated ANOVA (ANOVAR) was used to analyze the differences in plant transpiration, water sources, and plant physiological parameters between these species in pure and mixed plantations, respectively. This analysis was conducted with $SF_R$, RUP, relative water uptake proportions from three soil depths, and $\Psi_{pd} - \Psi_m$ as response variables, and "species" and "rainfall" as between-subject and within-subject factors, respectively. The same analysis was used to detect mixed afforestation effect on response variables for each plant species, with "plantation type" and "rainfall" as between-subject and within-subject factors, respectively. Furthermore, significant differences in fine root proportion for each soil layer (shallow, middle, and deep) for specific species between pure and mixed plantations were detected through independent-sample $t$-test. All of these analyses were calculated with SPSS 18 (IBM Inc., New York, US), after data normal distribution and homogeneity of variance analysis were tested.

**3 Results**

**3.1 Variation in environmental parameters and plant fine root vertical distribution**

The rainfall amount during the study period (262.7 mm, DOY 132–273) was 15. 6% lower than the average value during 2000–2017. Rainfall varied seasonally with 36 consecutive days having no rainfall event (DOY 157–192) and 5 days having successive rainfall events (DOY 237–241) (Fig. 1). The $ET_0$ (554.7 mm) was approximately twice the rainfall amount during the study period, with the higher and lower values during the low (DOY 132–202) and high (DOY 203–273) rainfall event periods, respectively (Fig. 1). The $R_n$ and VPD also exhibited higher and lower values during the low and high

rainfall event periods, respectively (Fig. S4). The SW increased and subsequently decreased by different degrees following rainfall events, with shallow soil layer (0–30 cm) exhibited higher variation than the corresponding value below 30 cm in the three plantations (Fig. 1, Table S3). The coefficients of variation (CVs, SD/mean) for SW in the shallow soil layer were 18.2%, 16.7%, and 17.3% in *H. rhamnoides* and *P. tomentosa* pure plantations and the mixed plantation, respectively. The SW for shallow and middle (30–100 cm) soil layers exhibited lower values than some deep soil layers (100–200 cm) during the less rainfall event period (such as DOY 157–192) in three plantations. In addition, compared with shallow and middle soil layers, the deep soil layer SW exhibited a time lag response to rainfall events.

The *H. rhamnoides* and *P. tomentosa* in pure plantations exhibited different fine root vertical distributions, with more than 40% of fine roots observed in shallow and deep soil layers, respectively (Fig. 2). In the shallow soil layer, no significant changes in fine root proportion were observed for *H. rhamnoides* in pure and mixed plantations (P > 0.05). However, the fine root proportion of *P. tomentosa* in the shallow soil layer was significantly increased from 21.9% in pure plantation to 31.3% in the mixed plantation (P < 0.05).

**Figure 1.**

**Figure 2.**

**3.2 Variations in sap flow**

Daily normalized $F_d$ for *H. rhamnoides* and *P. tomentosa* fluctuated with rainfall events in pure and mixed plantations (Fig. 3). The variation of normalized $F_d$ for *H. rhamnoides* and *P. tomentosa* in mixed plantation was higher than the specific species in pure plantations, with corresponding CVs of 29.4% and 31.8% in the mixed plantation, and 19.8% and 24.9% in pure plantations (Fig. 3). The $SF_R$ after rainfall pulses was significantly influenced by both rainfall amount and plant species (P < 0.001) (Fig. 3, Table S4). Following large rainfall amounts (≥15.4 mm), the diurnal variation of sap flow was significantly higher than the value before rainfall (P < 0.05) for *H. rhamnoides* in pure plantation and for *P. tomentosa* in both plantation types (Figs. S5 and S6). The lowest rainfall amount (7.9 mm) that

significantly increased the diurnal variation of sap flow was observed for *H. rhamnoides* in the mixed plantation (Fig. S5). Furthermore, in response to rainfall pulses, the $SF_R$ for *H. rhamnoides* in pure (range 6.7 $\pm$ 1.2% to 106.3 $\pm$ 4.7%) and mixed (range 2.2 $\pm$ 0.5% to 190.9 $\pm$ 15.5%) plantations was significantly higher (P < 0.001) than corresponding values for *P. tomentosa*: ranges 4.2 $\pm$ 0.5% to 60.3 $\pm$ 5.7% and 3.1 $\pm$ 0.5% to 83.0 $\pm$ 14.2% (Table S4). Mixed afforestation significantly enhanced $SF_R$ for both species (P < 0.001) (Table S4).

**Figure 3.**

### 3.3 Variations in plant water sources

The soil water $\delta^{18}O$ and $\delta D$ for pure *H. rhamnoides*, pure *P. tomentosa*, and mixed plantations showed large vertical variation following small rainfall events ($\leq$7.9 mm), and exhibited relatively small vertical variations following large rainfall events ($\geq$15.4 mm) (Fig. S7). The average isotopic values of soil water decreased from shallow to deep soil layers (Table S3), and water isotopic values in shallow and middle soil layer were close to rainwater in the three plantations following large rainfall events.

Although no significant difference in RUP was observed between *H. rhamnoides* (14.2 $\pm$ 7.8%) and *P. tomentosa* (12.4 $\pm$ 7.3%) in pure plantations (Fig. 4, Table S4), the RUP was significantly higher for *H. rhamnoides* (19.2 $\pm$ 8.6%) than *P. tomentosa* (14.6 $\pm$ 5.9%) in the mixed plantation (P < 0.05) (Table S4). In addition, *H. rhamnoides* mainly uptake water from the middle soil layer in pure and mixed plantations based on the MixSIR result, with corresponding average values of 36.3 $\pm$ 2.4% and 44.1 $\pm$ 3.1% (Fig. 5). The water source for *P. tomentosa* in pure and mixed plantations was mainly from the deep and middle soil layers, respectively, with corresponding average values of 41.4 $\pm$ 15.2% and 40.2 $\pm$ 5.9%. In pure plantation, the water source from shallow and middle soil layers for *H. rhamnoides* was significantly higher than *P. tomentosa*; however, the water source from the deep soil layer was significantly lower for the former species (P < 0.05) (Table S5). No significant differences in water sources from each soil layer were observed between these species in the mixed plantation (Table S5). In addition, mixed afforestation significantly enhanced RUP and decreased the deep soil water uptake proportion for *H. rhamnoides* and *P. tomentosa* (P < 0.05) (Table S4, Figs. 4 and 5).

**Figure 4.**

**Figure 5.**

### 3.4 Variations in plant leaf water potential

In response to rainfall pulses, *H. rhamnoides* exhibited higher CVs for $\Psi_m$ and $\Psi_{pd}-\Psi_m$ than corresponding values for *P. tomentosa* in both plantation types, meanwhile, *H. rhamnoides* exhibited lower CVs for $\Psi_{pd}$ than *P. tomentosa* in pure (16.9% and 18.3%, respectively) and mixed (13.5% and 19.7%, respectively) plantations (Fig. 6). Compared with *P. tomentosa*, *H. rhamnoides* exhibited significantly higher $\Psi_{pd}$ in the pure plantation, lower $\Psi_m$ in the mixed plantation, and larger $\Psi_{pd}-\Psi_m$ in both plantation types ($P < 0.05$) (Table S6). Meanwhile, mixed afforestation significantly reduced the $\Psi_m$ and increased the $\Psi_{pd}$ for *H. rhamnoides* and *P. tomentosa* ($P < 0.05$), respectively, and significantly increased $\Psi_{pd}-\Psi_m$ for both species (Table S6).

**Figure 6.**

### 3.5 Influence of water sources and $\Psi_{pd}-\Psi_m$ on plant transpiration

The $SF_R$ significantly increased with increasing RUP and decreasing $\Psi_{pd}-\Psi_m$ for *H. rhamnoides* ($P < 0.01$) in both plantation types (Fig. 7). Meanwhile, $SF_R$ significantly increased with decreasing $\Psi_{pd}-\Psi_m$ for *P. tomentosa* in both plantation types ($P < 0.05$). However, a significant relationship between $SF_R$ and RUP was observed for *P. tomentosa* in the mixed plantation ($P < 0.05$) (Fig. 7).

**Figure 7.**

## 4 Discussion

### 4.1 RRS uptake enhances plant transpiration for *H. rhamnoides* but not *P. tomentosa* in pure plantations

Rainwater is the only replenished soil water source in the studied region (Shao et al., 2018), because plants cannot uptake ground water of approximately 150 m depth below the surface, which was determined through well observation (unpublished data). Small rainfall events generally only wet the

soil surface and may evaporate before plant root uptake (Gebauer and Ehleringer, 2000). However, large rainfall events are most likely recharge soil water and enhance the metabolic activity of plant fine roots (Hudson et al., 2018), thus enhancing plant water uptake. Furthermore, the $\delta^{18}O$ and $\delta D$ values in small rainfall events generally exhibit higher values than those in large rainfall events (Fig. S7). Salamalikis et al. (2016) attribute this phenomenon to the sub-cloud evaporation effect in dry conditions where rainwater in small rainfall event is more vulnerable subject to evaporation during their descent process compared in large rainfall event. Similar to *Salix psammophila* and *Caragana korshinskii* in the studied region (Zhao et al., 2021), both *H. rhamnoides* and *P. tomentosa* take up water from different soil layers under varied soil water conditions following rainfall pulses in pure plantations (Fig. 5). In pure plantations, large water uptake proportion from the deep soil layer after 3.4 mm of rainfall for *H. rhamnoides* (52.5 ±8.7%) and *P. tomentosa* (64.1 ±5.1%) (Fig. 5), suggested that this rainfall amount did not relieve the drought caused by 36 days (DOY 157−192) of no rainfall. The RUP for *H. rhamnoides* but not *P. tomentosa* significantly increased following an increase in rainfall amount ($P < 0.05$) (Fig. S8), indicating that water uptake was more sensitive to rainfall pulse for *H. rhamnoides*. This may be mainly due to the greater proportions of fine root surface area distributed in the shallow soil layer for *H. rhamnoides* (40.9 ±3.1%) compared to *P. tomentosa* (21.9 ±2.3%) (Fig. 2).

The RRS uptake does not permit plant transpiration increase after rainfall pulses especially in semiarid and arid environments (Grossiord et al., 2017; West et al., 2007), and the influence of water potential gradient ($\Psi_{pd}−\Psi_{m}$) on plant transpiration should also be considered (Hudson et al., 2018; Kumagai and Porporato, 2012). For example, although *Juniperus osteosperma*, a deep-rooted plant species, could uptake RRS after large rainfall events in the west of the United States, the plant transpiration did not increase with increasing rainfall amount (West et al., 2007). The asynchronization between RRS uptake and plant transpiration for *J. osteosperma* was mainly attributed to the uptake of RRS by plants being unable to reverse the cavitation in its roots and stems (Grossiord et al., 2017; West et al., 2007). Our previous investigations in the studied region indicated that *P. tomentosa* is relatively more vulnerable to cavitation than *H. rhamnoides*, with water potential at 50% loss of conductivity of −1.15 MPa (Zhang et al., 2013) and −1.49 MPa (Dang et al., 2017), respectively, based on stem

vulnerability curves. Being less vulnerable to stem cavitation allowed *H. rhamnoides* to experience a significantly lower $\Psi_m$ and larger $\Psi_{pd}-\Psi_m$ compared with *P. tomentosa* in response to soil water conditions after rainfall pulses. Meanwhile, the $\Psi_{pd}-\Psi_m$ was significantly higher for *H. rhamnoides* (0.54 $\pm$0.26 MPa) compared to *P. tomentosa* (0.2 $\pm$0.06 MPa) (P<0.01), indicated that *H. rhamnoides* and *P. tomentosa* exhibited anisohydric and isohydric behavior, respectively, based on definitions of Franks et al. (2007) and Klein (2014). Previous studies demonstrated that isohydric plants generally exhibit more conservative transpiration than anisohydric plants when contending with varied soil water conditions (West et al., 2007; McDowell et al., 2008; Ding et al., 2021). The significantly higher (P < 0.001) $SF_R$ for *H. rhamnoides* (56.9 $\pm$43.9 %) than *P. tomentosa* (35.1 $\pm$26.9 %) indicated that plant transpiration for *H. rhamnoides* was more sensitive to rainfall pulses than *P. tomentosa*. Furthermore, after rainfall events, the $SF_R$ for *H. rhamnoides* but not for *P. tomentosa* significantly increased following rainfall amount increases (P < 0.05) (Fig. S8), also confirmed the more sensitive to rainfall pulses for *H. rhamnoides* compared with *P. tomentosa*.

Consistent with the first hypothesis, the influence of RRS uptake and $\Psi_{pd}-\Psi_m$ on $SF_R$ was different for these species in pure plantations. The $SF_R$ was significantly influenced by RUP and $\Psi_{pd}-\Psi_m$ for *H. rhamnoides* in the pure plantation, indicating that RRS uptake and leaf physiological adjustment enhanced its plant transpiration (Figs. 7 and 8). However, the $SF_R$ was significantly influenced by $\Psi_{pd}-\Psi_m$ for *P. tomentosa* (Fig. 7), suggesting that its transpiration was mainly constrained by plant physiological characteristics. The $ET_0$ and VPD represent the atmospheric evaporative demand factors and $R_n$ represents the energy factor, and these factors have been observed to influence plant transpiration (Du et al., 2011; Iida et al., 2016; Li et al., 2021). However, in the present study, none of $ET_0$, $R_n$, and VPD after rainfall or relative response of $ET_0$, $R_n$, and VPD significantly influenced $SF_R$ for either species in pure plantations (Table S7). The influence of plant physiological characteristics (i.e. $\Psi_{pd} - \Psi_m$) on $SF_R$ for both species, may partially contribute to the lack of atmosphere evaporative demand and energy effect on plant transpiration in the studied region, although these species exhibited different rainfall pulse sensitivity.

**Figure 8.**

## 4.2 RRS uptake enhances plant transpiration for coexisting species in mixed plantation

Spatial water resource partitioning is considered one of the essential plant strategies to maintain coexistence in mixed plantations, especially in semiarid and arid regions (Munoz-Villers et al., 2020; Silvertown et al., 2015; Yang et al., 2020). However, water source competition has widely been observed among coexisting plant species according to the literature surveys by Silvertown et al. (2015) and Tang et al. (2018), in either water sufficient or limited regions. In the present study, the non-significant differences in xylem $\delta^{18}O$ and $\delta D$ (P > 0.05) and plant water sources for the three soil layers (Fig. 5, Table S5) indicated water competition between these species in the mixed plantation, although the RUP was significantly higher for *H. rhamnoides* (Table S4).

Generally, two types of adaptation can be adopted by plants to cope with resource competition: increased competition ability or minimized competition interactions (West et al., 2007). Consistent with the first adaptation type, mixed afforestation significantly increased the RUP for *H. rhamnoides* and *P. tomentosa* (P < 0.01) (Table S4). Although mixed afforestation did not significantly alter the $\Psi_{pd}$ and $\Psi_m$ for *H. rhamnoides* and *P. tomentosa*, respectively, significantly lower $\Psi_m$ and higher $\Psi_{pd}$ were observed for corresponding species (P < 0.01) (Table S6). Mixed afforestation significant increased $\Psi_{pd}$ for *P. tomentosa*, possibly due to the advantage of access to soil moisture recharged by rainwater through an increased root surface area in the shallow soil layer for this species in the mixed plantation (Fig. 2). Thus, plant physiological ($\Psi_m$) and root morphological adjustments were adopted by *H. rhamnoides* and *P. tomentosa* in the mixed plantation, respectively, to significantly enlarge $\Psi_{pd}-\Psi_m$ and increase RUP (Fig. 8). Similar to the results in pure plantations, the significant higher $\Psi_{pd}-\Psi_m$ (0.72 $\pm$ 0.32 MPa) and $SF_R$ (89.2 $\pm$ 80.2%) for *H. rhamnoides* compared to *P. tomentosa* (0.39 $\pm$ 0.09 MPa and 50.7 $\pm$ 38.1%, respectively) in mixed plantation (Figs. 3 and 6), suggested that *H. rhamnoides* and *P. tomentosa* exhibited anisohydric and isohydric behavior in mixed plantation, respectively, and the former plant species was more sensitive to rainfall pulses than *P. tomentosa*. In addition, the different influence of RUP and $\Psi_{pd}-\Psi_m$ on $SF_R$ for specific species in pure and mixed plantations was consistent with the second hypothesis. The significant influence of both RUP and $\Psi_{pd}-\Psi_m$ on $SF_R$ was observed for *P.*

*tomentosa* in mixed rather than in pure plantation (Fig. 7). Meanwhile, for *H. rhamnoides* in mixed plantation compared to specific value in pure plantation, larger and smaller slopes in linear regression were observed between $SF_R$ and RUP, and $SF_R$ and $\Psi_{pd}-\Psi_m$, respectively (Fig. 7). Furthermore, no significant relationship of $SF_R$ with $ET_0$, VPD, and $R_n$ after rainfall and of $SF_R$ with relative response of $ET_0$, VPD, and $R_n$ was observed for these species in the mixed plantation from DOY 132 to 273 and from DOY 203 to 273 (Table S7). This result also confirmed the influence of physiological or morphological factors on plant transpiration for these species in the mixed plantation in response to rainfall pulses.

Additionally, consistent with other studies in the Loess Plateau (Wang et al., 2020a; Wu et al., 2021), the deep soil layer generally exhibited lower SW than other soil layers in all plantation types in the present study (Fig. 1, Table S3). Jia et al. (2017) and Wang et al. (2020a) attributed the lower SW in deep soil layers to the imbalance between rainwater replenishment and plant uptake of water from this soil layer in the studied region. Silvertown et al. (2015) and Tang et al. (2019) suggested that coexisting plant species generally reduce water uptake from soil layers that exhibit low soil water content to avoid water source competition in these layers and maintain stable coexistence. In the present study, consistent with the second adaptation type, mixed afforestation significantly decreased the water uptake proportion from the deep soil layer for these species (Table S5). Thus, both increased rainwater-recharged soil water uptake and decreased water source competition from the deep soil layer were adopted by these species in the mixed plantation to minimize water sources competition under water limited conditions.

### 4.3 Implications for plantation species selection based on RRS uptake and plant transpiration

The RRS uptake and plant transpiration in response to rainfall pulses may influence plant physiological process and the water cycle (Meier et al., 2018; Zhao et al., 2021). In pure plantations, *H. rhamnoides* rather than *P. tomentosa* showed an advantage in RRS uptake due to the large $\Psi_{pd}-\Psi_m$ and high fine root surface area proportions distributed in the shallow soil layer for the former species. The excessive water uptake from the deep soil may desiccate deep soil (Wu et al., 2021), weakening plant

resilience to drought stress and thus plant community sustainability in this Loess Plateau region (Song et al., 2018; Zhao et al., 2021). West et al. (2012) and Wu et al. (2021) suggested that increased RRS uptake can reduce plant water uptake from deep soil layers, and is essential for plantation adaptation in water limited regions. In the present study, physiological (e.g., $\Psi_m$) and morphological (fine root distribution) adjustments were observed for *H. rhamnoides* and *P. tomentosa* in the mixed plantation,

respectively, to enlarge $\Psi_{pd}-\Psi_m$ and enhance the RUP and plant transpiration (Figs. 2 and 8). Meanwhile, the significantly increased RUP and decreased deep soil water uptake proportion for both species in mixed plantation may relieve deep soil water deficit and strengthen plantation sustainability (Tables S4 and S5). Furthermore, mixed afforestation also increased the total biomass of *H. rhamnoides* and *P. tomentosa*, calculated through the allometric equation indicated in Zhou et al. (2018) and Tang et

al. (2019) (Table S8). Thus, rainfall pulse sensitive species in pure plantation, and plant species in mixed plantation that can adopt physiological or morphological adjustment to enhance rainwater-recharged soil water uptake and reduce excessive water uptake from deep soil layers, should be more often considered for use in the studied region.

**5 Conclusions**

The influence of water sources and $\Psi_{pd}-\Psi_m$ on plant transpiration in response to rainfall pulses was determined for *H. rhamnoides* and *P. tomentosa* in the semiarid Loess Plateau region. In pure and mixed plantations, the large $\Psi_{pd}-\Psi_m$ was consistent with high $SF_R$ for *H. rhamnoides* suggesting that this species exhibited anisohydric behavior and sensitivity to rainfall pulses. Meanwhile, the small $\Psi_{pd}-\Psi_m$

was consistent with low $SF_R$ for *P. tomentosa* in both plantation types, and indicated that this species exhibited isohydirc behavior and less sensitivity to rainfall pulses. In addition, significantly lower plant $\Psi_m$ and increased fine root surface area were adopted by *H. rhamnoides* and *P. tomentosa* in the mixed plantation, respectively, to enlarge $\Psi_{pd}-\Psi_m$ and enhance RRS uptake and decrease water source competition from the deep soil layer. The $SF_R$ was significantly influenced by RUP and $\Psi_{pd}-\Psi_m$ for *H.*

*rhamnoides* in both plantation types, however, the $SF_R$ for *P. tomentosa* was significantly influenced by $\Psi_{pd}-\Psi_m$ in the pure plantation and by RUP and $\Psi_{pd}-\Psi_m$ in the mixed plantation. This study indicates

that, through plant physiological or morphological adjustment, RRS uptake can enhance plant transpiration in the mixed plantation regardless of species sensitivity to rainfall pulses in water limited regions.


## Appendix A: Runoff simulated from the HYDRUS-1D model

The HYDRUS-1D model is based on the Richards' equation (Richards, 1931) to describe soil water dynamics (Šimůnek et al., 2008; Šimůnek et al., 2013):

$$\partial\theta / \partial t = \partial / \partial z \left( K(h,z)((\partial h / \partial z)+1) \right) - S_r(z,t) \tag{A1}$$

where $\theta$, $t$, $h$, and $z$ are the soil moisture content (SW, cm$^3$ cm$^{-3}$), simulation time (day), pressure head (cm), and vertical coordinate (cm), respectively. $K(h, z)$ and $S_r(z, t)$ are the unsaturated hydraulic conductivity (cm day$^{-1}$) (Mualem, 1976; van Genuchten, 1980) and root water uptake (cm$^3$ cm$^{-3}$ day$^{-1}$), respectively.

This model has been widely used to simulate soil water hydrological processes with HYDRUS-1D

software (Šimůnek et al., 2013), such as soil water content dynamics and runoff in the Loess Plateau (Yi and Fan, 2016; Bai et al., 2020; Wang et al., 2020b). The model was used to calculate the runoff for each plantation type in this study, after calibration and validation this model using the observed SW. Based on suggestions in Yi and Fan (2016) and Bai et al. (2020), the atmospheric boundary condition with surface runoff and free drainage were selected as upper and lower boundary condition, respectively, to

calibrate and validate this model and calculate runoff (Fig. A1).

**Figure A1.**

## 1 Data sources

The observed meteorological, plant, and soil hydraulic parameters were the basic inputs for this model.

**1.1 Meteorological parameters**

The meteorological parameters required for HYDRUS-1D include relative humidity, wind speed ($W_S$), air temperature, rainfall amount, and reference evapotranspiration ($ET_0$). Daily relative humidity, maximum, minimum and average air temperatures, $W_S$, and rainfall amount were measured by a weather

station approximately 500 m from the research plots. The $ET_0$ (cm day$^{-1}$) was calculated through method described by Allen et al. (1998). The detailed information can be observed in "2.2 Environmental parameter measurements and $ET_0$ calculation" subsection in the manuscript.

**1.2 Plant parameters**

The plant parameters required for HYDRUS-1D include plant height, root depth, and potential transpiration rate. Plant height and root depth in each plantation type can be observed in Table S1 and Figure S4 in the manuscript, respectively. The leaf area index (LAI) was measured monthly, from May to September, for each plantation type using a LAI-2200 (LiCor Inc., Lincoln, USA). The potential transpiration rate (cm day$^{-1}$) was calculated using the Beer equation (Ritchie, 1972) based on the measured LAI and extinction coefficient value (0.39) suggested in Šimůnek et al. (2013).

**1.3 Soil hydraulic parameters**

The saturated soil water content ($\theta s$) and hydraulic conductivity ($Ks$), van Genuchten model parameters ($\alpha$ and $n$), and residual soil water content ($\theta r$) were required parameters for HYDRUS-1D. The $Ks$, $\theta s$, and soil bulk density (BD) at soil depth intervals of 0-20, 20-50, 50-100, and 100-200 cm were measured in July 2018 using the cutting ring (Wu et al., 2016) and constant water head (Reynolds et al., 2002) method in each plantation type. The soil particle composition was determined using a Mastersize 2000 (Malvern Instruments Ltd., UK). Additionally, the slopes for these three plantation types were required for HYDRUS-1D. The detailed information can be observed in "2.1 Study site" subsection in the manuscript. The measured soil hydraulic parameters for the three plantation types are shown in Table A1. The Rosetta pedotransfer function was used to calculate $\theta r$, $\alpha$, and $n$ (Jana and Mohanty, 2012; Bai et al., 2020).

**Table A1.**

**2 Model calibration and validation, and runoff calculation**

In each plantation type, SW was measured at 5, 20, 50, 100, 150, and 200 cm below ground (SW$_{5cm}$, SW$_{20cm}$, SW$_{50cm}$, SW$_{100cm}$, SW$_{150cm}$, and SW$_{200cm}$) by CS615 probes (Campbell Scientific Inc.). The detailed information can be observed in "2.1 Study site" subsection in the manuscript. The SW at each

soil depth in each plantation type from DOY 132 to 202 was used to calibrate HYDRUS-1D. The Ks, $\theta s$, $\theta r$, $a$, and $n$ were optimized using the inverse solution module in HYDRUS-1D (Table A2).

**Table A2.**

Subsequently, the SW values from DOY 203 to 273 in each plantation types were used to validate the model. The root mean square error (RMSE), Nash-Sutcliffe efficiency coefficient (NSE), and determinant coefficient ($R^2$) based on the observed and simulated SW were used to evaluate the model performance (Bai et al., 2020):

$$RMSE = \sqrt{\sum_{i=1}^{N} (SW_o - SW_s)^2 / N} \tag{A2}$$

$$NSE = 1 - (\sum_{i=1}^{N} (SW_s - SW_o)^2 / \sum_{i=1}^{N} (SW_o - SW_{oave})^2) \tag{A3}$$

$$R^2 = 1 - (\sum_{i=1}^{N} (SW_s - SW_o)^2 / \sum_{i=1}^{N} (SW_o - SW_{oave})^2) \tag{A4}$$

Where $SW_o$ and $SW_s$ are observed and simulated SW at time i at each soil depth, respectively. N is the observation number. $SW_{oave}$ is average observed *SW*. Lower of RMSE and (or) more close to 1 of NSE and $R^2$ indicated high accuracy of *SW* simulation.

The simulated SWs at different soil depths closely matched the variation of these values observed from DOY 203 to 273, the example can be observed in the *H. rhamnoides* pure plantation in Figs. A2 and A3. The RMSE ranged from 0.005–0.008, 0.006–0.009, and 0.006–0.01 in the *H. rhamnoides* pure, *P. tomentosa* pure, and mixed plantation, respectively (Table A3). The NSE ranged from 0.52–0.7, 0.57–0.67, and 0.54–0.76 in the *H. rhamnoides* pure, *P. tomentosa* pure, and mixed plantation, respectively, and the corresponding $R^2$ ranged from 0.71–0.84, 0.76–0.83, and 0.76–0.82. The calculated RMSE, NSE, and $R^2$ indicated that the simulated results were acceptable for three plantation types in this study (Table A3), based on the criterions suggested in Bai et al. (2020) and Wang et al. (2020b). The RMSE ranged from 0.022–0.036 and NSE ranged from -0.54–0.71 in Bai et al. (2020), and the RMSE ranged from 0.005–0.032 and $R^2$ ranged from 0.8–0.92 in Wang et al. (2020b) between the observed and simulated SW for this model.

**Figure A2.**

**Figure A3.**

**Table A3.**

Finally, for each plantation type, runoff was calculated based on HYDRUS-1D (Šimůnek et al., 2013). The results from the model indicated that no runoff was generated during the studied period from DOY 132 to 273 in *H. rhamnoides* pure plantation, *P. tomentosa* pure plantation, and mixed plantation. Thus, we expect that no runoff would be generated during the selected rainfall events.

**Data availability**

The data that support the findings of this study are available from the corresponding author upon request.

**Author contribution**

YKT designed the study, performed the statistical analyses and wrote the original manuscript draft. LNW and YQY performed the experiments and collected the data. DXL collected the data.

**Declaration of Competing Interest**

The authors declare that they have no conflict of interest.

**Acknowledgement**

This work was supported by the National Natural Science Foundation of China (41977425), the National Key Research and Development Program of China (2017YFA0604801). We acknowledge the insightful suggestions of editor and reviewers.

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

| Soil depth (cm) | Soil particle composition | | | Soil hydraulic parameter | | |
|---|---|---|---|---|---|---|
| | Sand (%) | Silt (%) | Clay (%) | $\theta_s$ ($cm^3\ cm^{-3}$) | BD ($g\ cm^{-3}$) | Ks ($cm\ day^{-1}$) |
| *H.* | 0-20 | 26.4 | 63.5 | 10.1 | 0.37 | 1.28 | 75.7 |
| *rhamnoides* | 20-50 | 22.2 | 61.6 | 16.2 | 0.34 | 1.35 | 70.3 |
| **pure** | 50-100 | 23.5 | 63.1 | 13.4 | 0.32 | 1.42 | 55.4 |
| **plantation** | 100-200 | 24.7 | 63.8 | 11.5 | 0.29 | 1.46 | 50.6 |
| | 0-20 | 25.8 | 62.2 | 12 | 0.35 | 1.21 | 82.4 |
| *P. tomentosa* | 20-50 | 23.7 | 62.5 | 13.8 | 0.35 | 1.33 | 73.7 |
| **pure** | 50-100 | 22.2 | 61.5 | 16.3 | 0.31 | 1.42 | 58.9 |
| **plantation** | 100-200 | 24.9 | 64.8 | 10.3 | 0.3 | 1.45 | 52.6 |
| | 0-20 | 25.5 | 63.8 | 10.7 | 0.36 | 1.25 | 78.5 |
| **Mixed** | 20-50 | 24.3 | 62.7 | 13 | 0.35 | 1.31 | 73.2 |
| **plantation** | 50-100 | 23.8 | 64.9 | 11.3 | 0.34 | 1.39 | 60.5 |
| | 100-200 | 24.6 | 63.7 | 11.7 | 0.31 | 1.45 | 53.4 |

$\theta_s$= saturated soil water content, *Ks*=saturated hydraulic conductivity, BD= soil bulk density

**Table A2.** Optimized soil hydraulic parameters in both pure and mixed plantations through HYDRUS-1D

| | Soil depth (cm) | $\theta r$ (cm$^3$ cm$^{-3}$) | $\theta s$ (cm$^3$ cm$^{-3}$) | Ks (cm day$^{-1}$) | $a$ | $n$ |
|---|---|---|---|---|---|---|
| **H. rhamnoides pure plantation** | 0-20 | 0.08 | 0.36 | 74.9 | 0.018 | 1.6 |
| | 20-50 | 0.08 | 0.34 | 71.2 | 0.018 | 1.6 |
| | 50-100 | 0.0823 | 0.31 | 56.2 | 0.01 | 1.45 |
| | 100-200 | 0.0823 | 0.3 | 51.5 | 0.01 | 1.43 |
| **P. tomentosa pure plantation** | 0-20 | 0.08 | 0.36 | 82.1 | 0.019 | 1.62 |
| | 20-50 | 0.08 | 0.35 | 73.5 | 0.018 | 1.6 |
| | 50-100 | 0.0821 | 0.31 | 59.2 | 0.01 | 1.51 |
| | 100-200 | 0.0822 | 0.31 | 51.6 | 0.011 | 1.47 |
| **Mixed plantation** | 0-20 | 0.08 | 0.37 | 79.2 | 0.018 | 1.61 |
| | 20-50 | 0.08 | 0.36 | 74.2 | 0.018 | 1.61 |
| | 50-100 | 0.0822 | 0.34 | 60.2 | 0.011 | 1.46 |
| | 100-200 | 0.0823 | 0.3 | 55.8 | 0.011 | 1.45 |

$\theta_r$= residual soil water content, $Ks$=saturated hydraulic conductivity, $\theta_s$= saturated soil water content, $a$ and $n$ = parameters of van Genuchten model.

**Table A3.** The RMSE, NSE, and $R^2$ between the observed and simulated SW during the HYDRUS-1D validation period (from DOY 203-273)

|  | Soil depth (cm) | RMSE | NSE | $R^2$ |
|---|---|---|---|---|
| *H. rhamnoides* **pure plantation** | 5 | 0.008 | 0.65 | 0.84 |
| | 20 | 0.006 | 0.58 | 0.83 |
| | 50 | 0.006 | 0.7 | 0.71 |
| | 100 | 0.008 | 0.56 | 0.85 |
| | 150 | 0.005 | 0.59 | 0.81 |
| | 200 | 0.006 | 0.52 | 0.78 |
| *P. tomentosa* **pure plantation** | 5 | 0.008 | 0.67 | 0.79 |
| | 20 | 0.008 | 0.62 | 0.76 |
| | 50 | 0.006 | 0.72 | 0.82 |
| | 100 | 0.009 | 0.59 | 0.75 |
| | 150 | 0.008 | 0.57 | 0.83 |
| | 200 | 0.009 | 0.61 | 0.78 |
| **Mixed plantation** | 5 | 0.009 | 0.61 | 0.81 |
| | 20 | 0.01 | 0.54 | 0.76 |
| | 50 | 0.008 | 0.68 | 0.82 |
| | 100 | 0.008 | 0.7 | 0.79 |
| | 150 | 0.006 | 0.76 | 0.82 |
| | 200 | 0.008 | 0.67 | 0.81 |

RMSE= root mean square error, NSE = Nash-Sutcliffe efficiency coefficient, $R^2$= determinant coefficient

**Figure Legends**

**Figure 1.** Variation in (a) rainfall amount, reference evapotranspiration ($ET_0$), and average (mean $\pm$ SD) soil water content (SW) in (b) *H. rhamnoides* pure plantation, (c) *P. tomentosa* pure plantation, and (d) mixed plantation from DOY 132 to 273 (11 May to 30 September) (n = 3). Standard deviation bars for SW at each soil layers are not shown to allow clear display of variation of SW for each plantation. Arrows in (a) indicate dates of sample collection at the first day after rainfall events: DOY 157 (6 June), DOY 194 (12 July), DOY 204 (23 July), DOY 249 (6 September), and DOY 266 (23 September).

**Figure 2.** Variation in average surface area of fine root at different soil depths for *H. rhamnoides* and *P. tomentosa* in (a) pure and (b) mixed plantations. Error bars indicate the standard deviation (n = 3).

**Figure 3.** Variation in (a) rainfall amount, and average daily normalized $F_d$ for *H. rhamnoides* in (a) pure and (b) mixed plantations and for *P. tomentosa* in (c) pure and (d) mixed plantations from DOY 132 to 273 (11 May to 30 September) (n = 3). Arrows in (a) indicate dates of sample collection at the first day after rainfall events: DOY 157 (6 June), DOY 194 (12 July), DOY 204 (23 July), DOY 249 (6 September), and DOY 266 (23 September).

**Figure 4.** Variation in average rainwater-recharged soil water uptake proportion (RUP) for *H. rhamnoides* and *P. tomentosa* in (a) pure and (b) mixed plantations after five rainfall events (n = 3).

**Figure 5.** Variation in average plant water sources from three soil layers (0–30, 30–100, and 100–200 cm) for *H. rhamnoides* in (a) pure and (b) mixed plantations, and for *P. tomentosa* in (c) pure and (d) mixed plantations after five rainfall events (n = 3).

**Figure 6.** Variation in average plant predawn ($\Psi_{pd}$), midday leaf water potential ($\Psi_m$), and leaf water potential gradient ($\Psi_{pd}-\Psi_m$) for (a–c) *H. rhamnoides* and (d–f) *P. tomentosa* in both plantation types after five rainfall events (n = 3).

**Figure 7.** Relationship between average relative response of normalized $F_d$ ($SF_R$) and (a, b) rainwater-recharged soil water uptake proportion (RUP), and between $SF_R$ and (c, d) leaf water potential gradient ($\Psi_{pd}-\Psi_m$) for *H. rhamnoides* and *P. tomentosa* in both plantation types (n = 3).

**Figure 8.** Schematic of rainwater-recharged soil water (RRS) uptake, leaf water potential gradient, and plant transpiration for *H. rhamnoides* and *P. tomentosa* in both plantation types. Both RRS uptake proportion (RUP) and leaf water potential gradient ($\Psi_{pd}-\Psi_m$) enhanced plant transpiration after rainfall pulses for *H. rhamnoides* in pure and mixed plantations, and for *P. tomentosa* in mixed plantation. However, $\Psi_{pd}-\Psi_m$ significantly influenced plant transpiration after rainfall pulses for *P. tomentosa* in the pure plantation. Mixed afforestation effect of these parameters for each species are indicated at the bottom half of the schematic, with "increase", "decrease" or "enlarge" indicating a significant difference (P < 0.05) for a species between pure and mixed plantations. Mixed afforestation significantly enhanced RUP and plant transpiration, decreased $\Psi_m$, and enlarged $\Psi_{pd}-\Psi_m$ for *H. rhamnoides*, and also significantly enhanced the RUP and plant transpiration, increased $\Psi_{pd}$, and enlarged $\Psi_{pd}-\Psi_m$ for *P. tomentosa*.

**Figure A1.** The upper and lower boundary conditions selection in HYDRUS-1D software (Version 4.15).

**Figure A2.** Variation in soil water content (SW) at 5, 50, 50, 100, 150, and 200 cm depths during the HYDRUS-1D (a-f) calibration (from DOY 132-202) and (g-l) validation (from DOY 203-273) period in H. rhamnoides pure plantation.

**Figure A3.** The relationship between observed ($SW_O$) and simulated ($SW_s$) soil water content at 5, 50, 50, 100, 150, and 200 cm depths during the HYDRUS-1D validation   period (from DOY 203-273) in *H. rhamnoides* pure plantation.

**Figure 1**

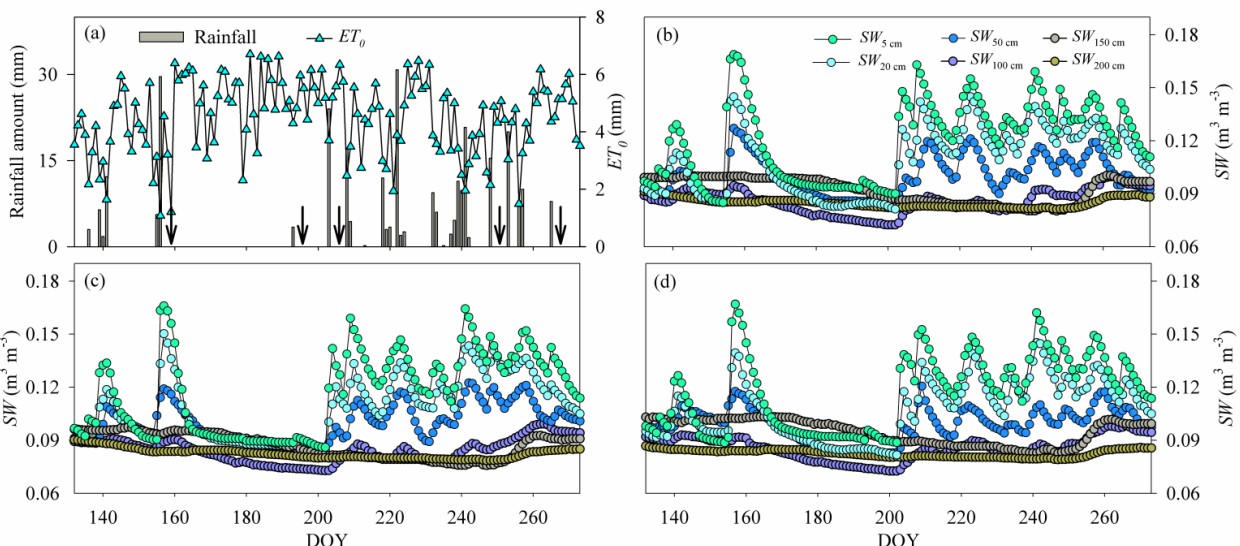

 **Figure 2**

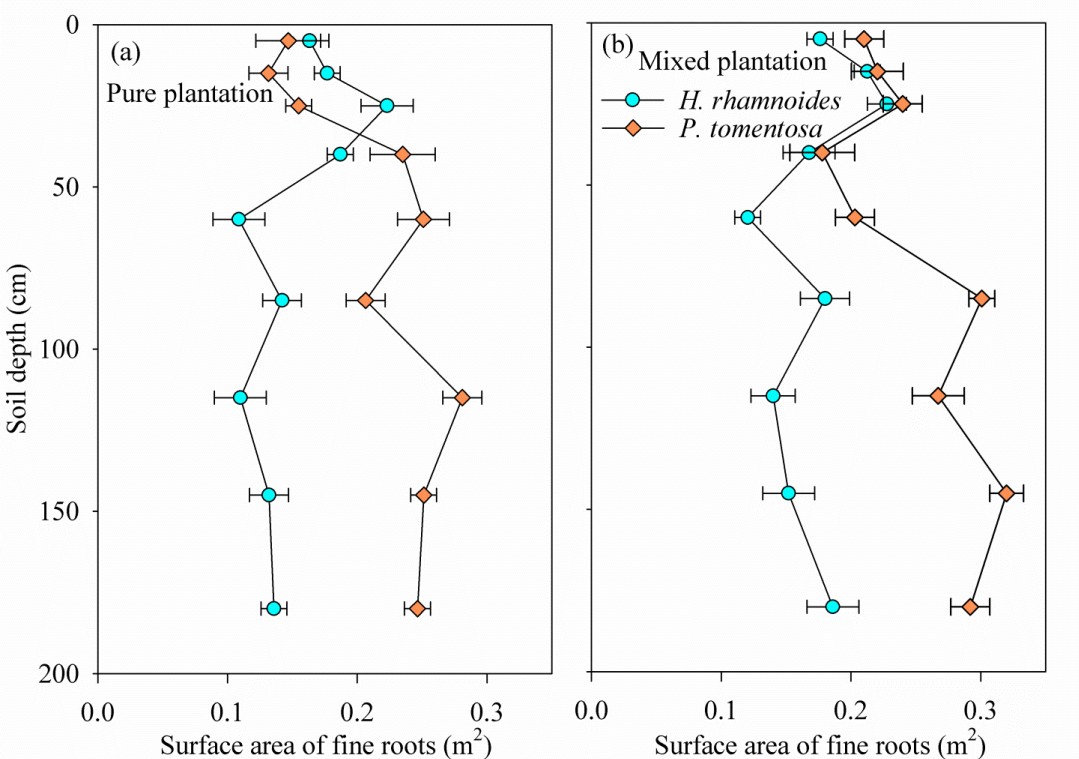

**Figure 3**

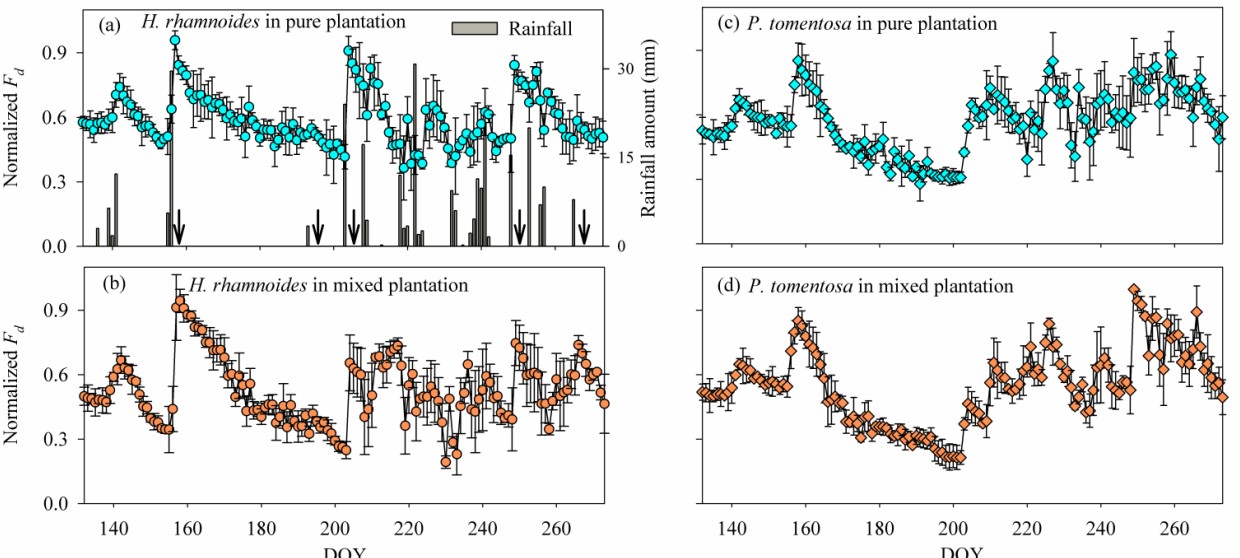

885

**Figure 4**

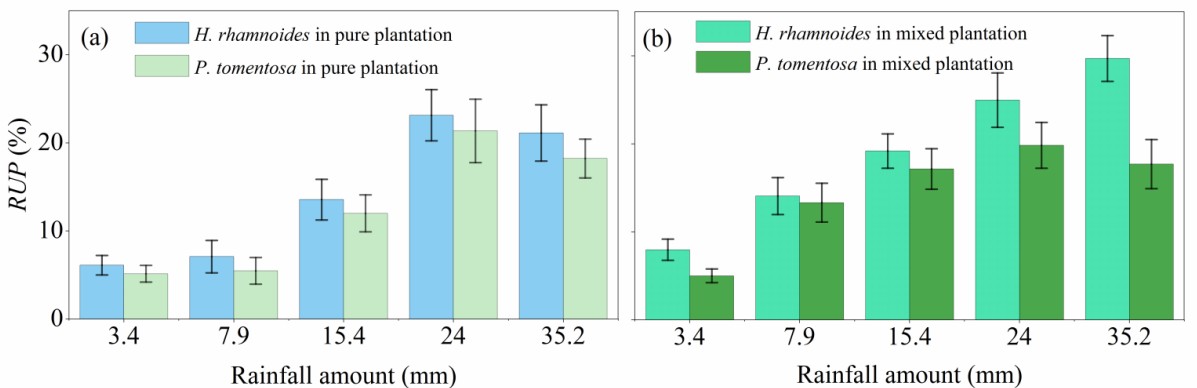

**Figure 5**

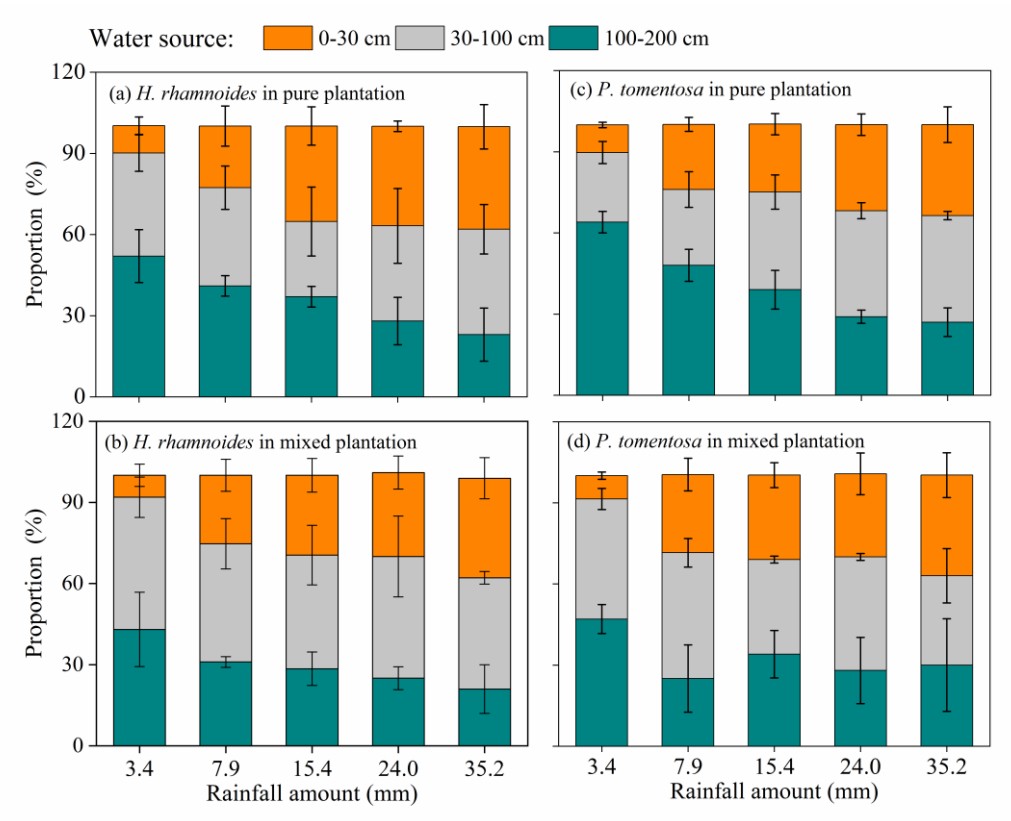

**Figure 6**

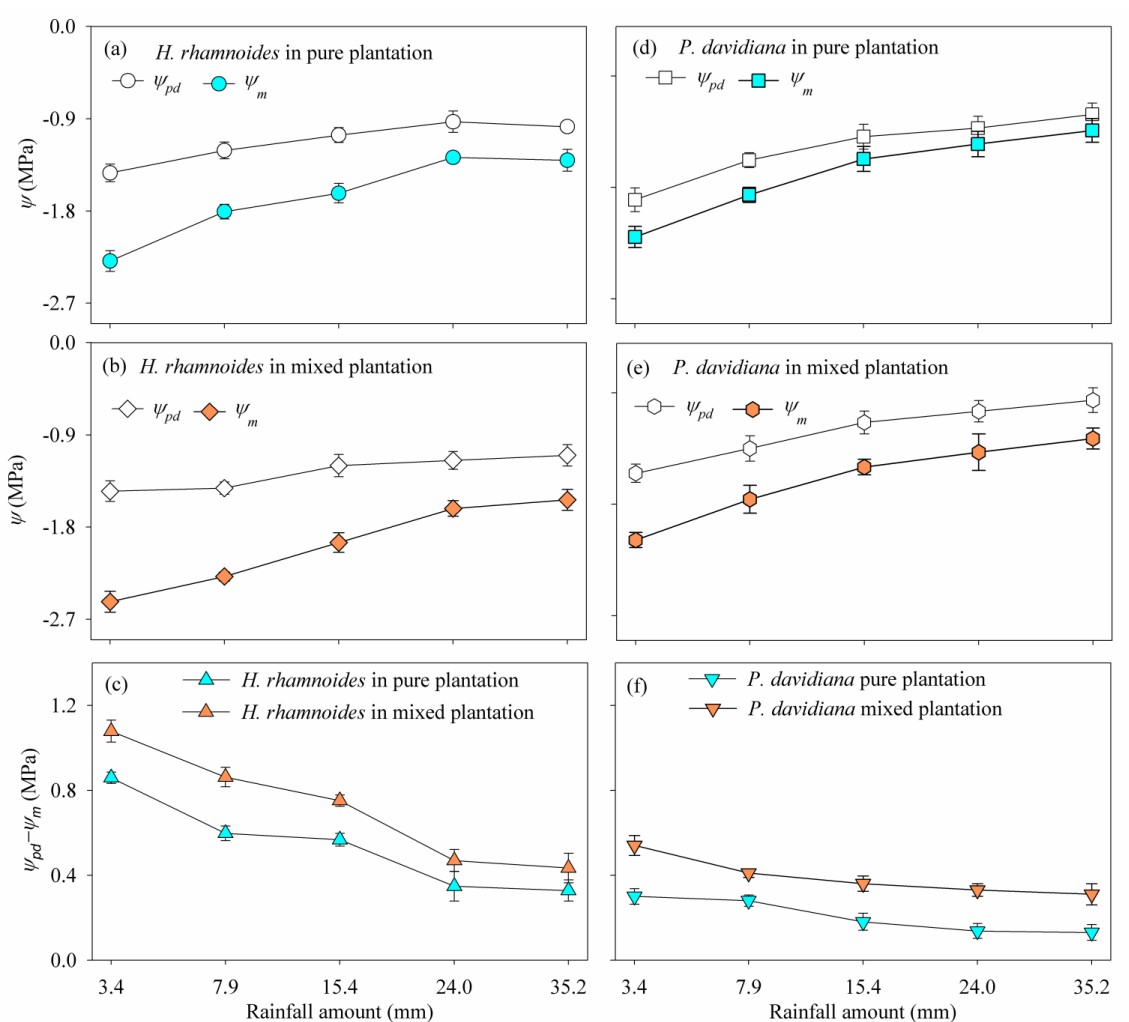

 **Figure 7**

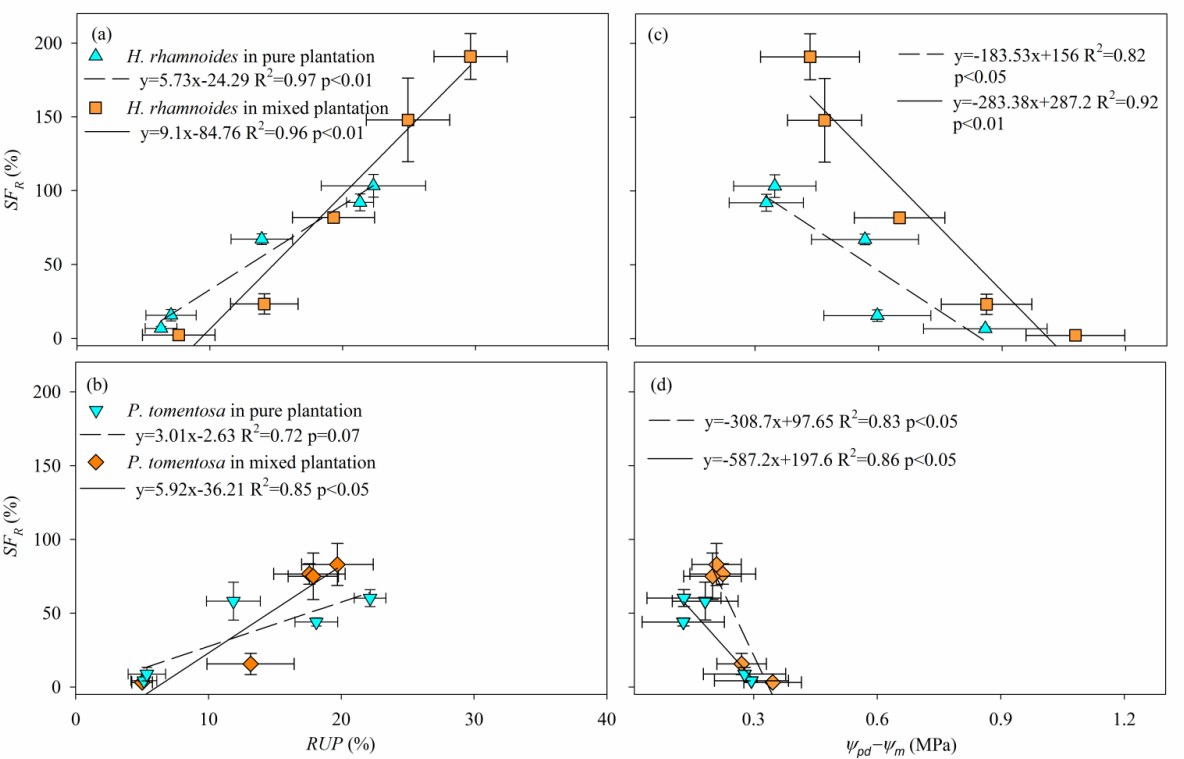

**Figure 8**

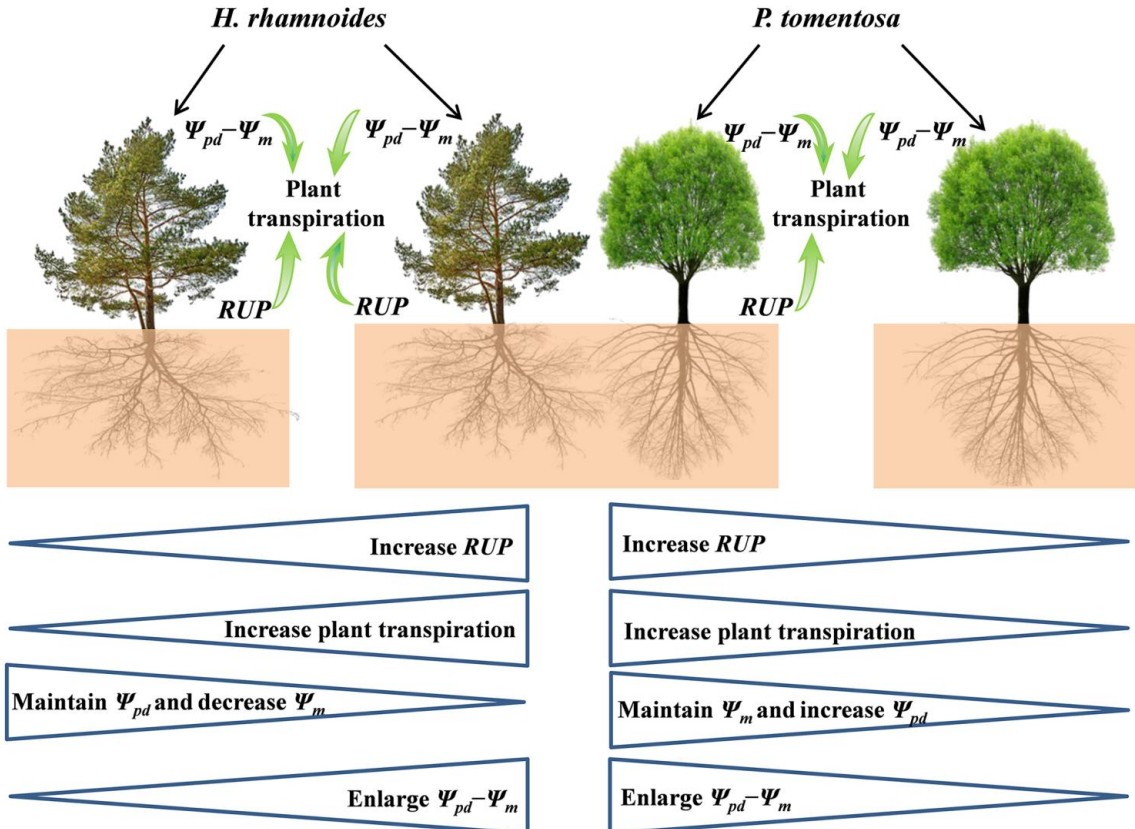

900

**Figure A1**

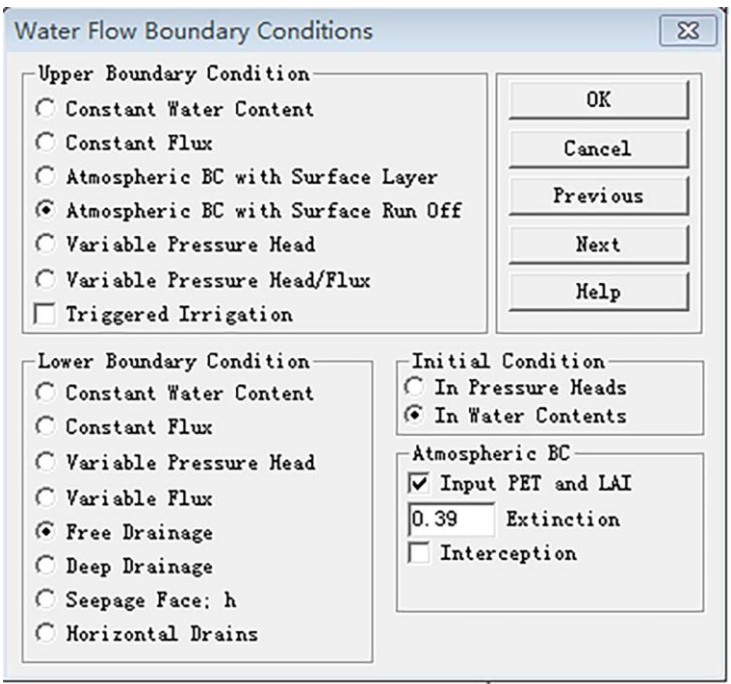

 **Figure A2**

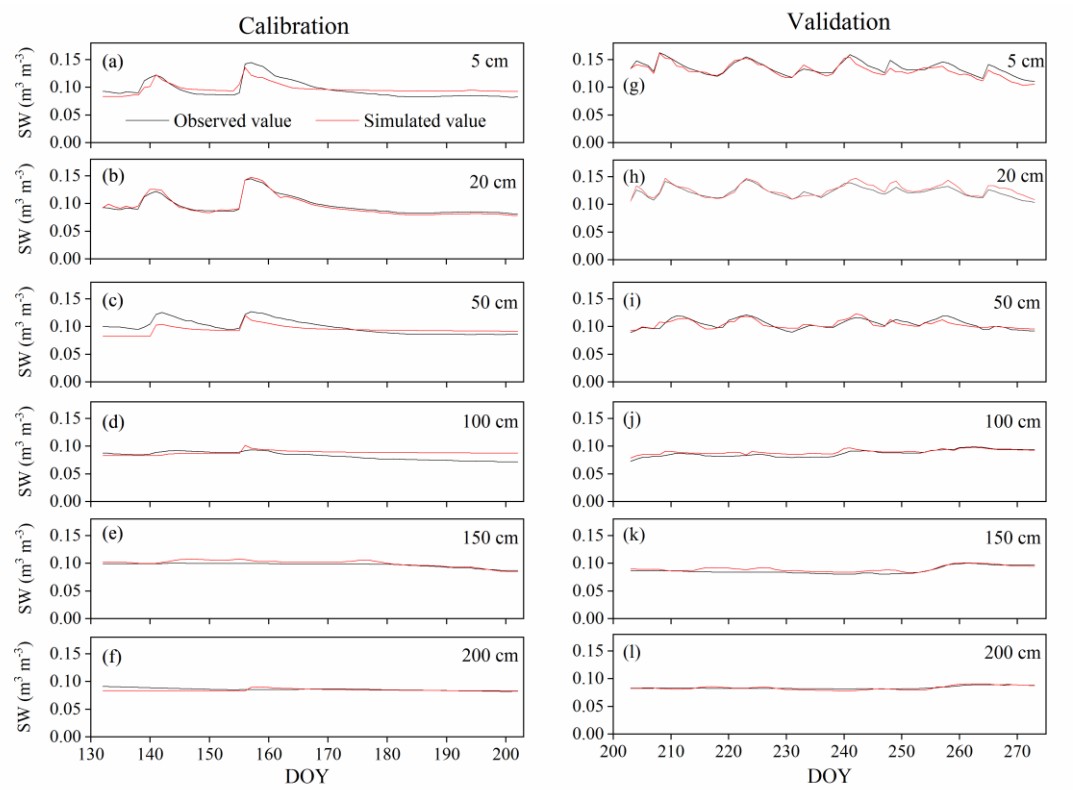

**Figure A3**

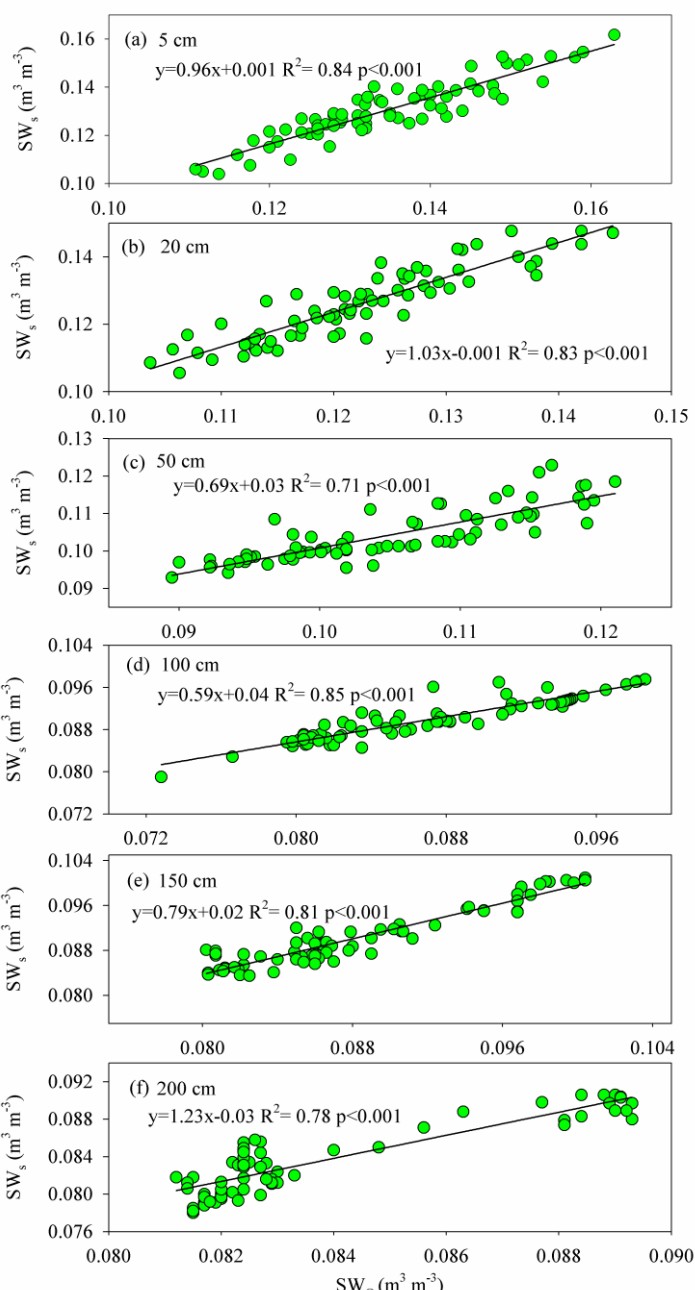