# Peer review of "Differential response of plant transpiration to uptake of rainwater-recharged soil water for dominant tree species in the semiarid Loess Plateau"

_Hydrology and Earth System Sciences, 2021_

## Author Comment (AC1)

Authors' responses to Reviewer comments on the manuscript of "Differential response of plant transpiration to uptake of rainwater-recharged soil water for dominant tree species in the semiarid Loess Plateau". Manuscript ID: hess-2021-351.

**Dear Reviewer,**

We deeply appreciate you for giving us an opportunity to revise our manuscript. The point-to-point responses (responses in upright Roman) to the Reviewer comments (*original comment and query in Itali*) can be observed in file named "**Response to Reviewer 1**".

**Reviewer 1**

**Major Comments:**

*1) Do you have any information about runoff generation of the studied plantation sites? Any runoff after rainfall pulse may influence the result of your manuscript since the contribution of precipitation to plant water uptake is central to your study, although precipitation amount was not the direct independent factor used during the data analysis. So, considered the potential runoff may strengthen and validity your result.*

**Response: Added and Clarified.** Thanks for this meaningful suggestion, the soil bulk densities, soil filtration properties, soil total porosity, and soil capillary porosity have been added in "2.1 Study site" subsection in "**2 Materials and methods**" section as follows: "Based on an experiment conducted in July 2017 through cutting ring method, the soil bulk density, filtration property, total porosity, and capillary porosity at 0–50 cm soil depth were similar in three plantations. The average soil bulk density was 1.34 $\pm$0.04, 1.31 $\pm$0.05, and 1.31 $\pm$0.05 g cm$^{-3}$ for pure *H. rhamnoides*, pure *P. tomentosa*, and mixed plantations, respectively, and corresponding soil saturated hydraulic conductivity was 0.97 $\pm$0.15, 0.96 $\pm$0.13, and 0.99 $\pm$0.11 mm min$^{-1}$. The average soil total porosity was 48.25 $\pm$0.52, 48.17 $\pm$0.48, and 48.03 $\pm$0.63% for pure *H. rhamnoides*, pure *P. tomentosa*, and mixed plantations, respectively, and corresponding soil capillary porosity was 38.89 $\pm$1.57, 39.02 $\pm$1.26, and 38.95 $\pm$1.87%."

In addition, the relative sentences have also been added in "4.3 Implications for plantation species and type selection based on RRS uptake and plant transpiration" subsection in "**4 Discussion**" section as follows: "In addition, no runoff was generated under 0.74 mm min$^{-1}$ rainfall intensity in silt loam soil in the Loess Plateau (Huang et al., 2014), which had no vegetation cover and similar soil saturated hydraulic conductivity (0.99 $\pm$0.15 g cm$^{-3}$) to that in the present study. Pan and Shuangguan (2005) also observed no runoff generation under 1.5 mm min$^{-1}$ rainfall intensity for vegetation covered plots with 15 °slope in the Loess Plateau. Direct observation for possible runoff after large rainfall events in further studies would be helpful for evaluating plantation species adaptability in the studied region, although Zhao et al. (2013) showed that the vegetation cover can enhance soil permeability and reduce water loss in the Loess Plateau. Furthermore, water conservation measures, such as water-fertilizer pits (60 $\times$60 $\times$40 cm) (Wang et al., 2020), that can intercept any possible runoff after large rainfall events and deliver it to deep soil layers may be appropriate for the studied region." (Page 23 Lines 526-536).

In the present study, we did no directly measure the runoff during the experiment period. The studied area is typical loess region with similar or smaller slopes compared with mentioned above studies, such as Pan and Shuangguan (2005) and Zhao et al. (2013). The soil bulk density and saturated hydraulic conductivity (0-50cm) was ranged from1.31 to 1.34 g cm$^{-3}$ and from 0.96 to 0.99 mm min$^{-1}$, respectively, in the present study, which is similar with the soil bulk density (1.35 g cm$^{-3}$) and saturated hydraulic conductivity value (0.99$\pm$0.15 g cm$^{-3}$) as the direct runoff experiment in Huang et al. (2014). Huang et al. (2014) suggested that no runoff was generated under 0.7 mm min$^{-1}$ rainfall intensity in silt loam soil in Loess Plateau, with simulated rainfall amount ranged from 53.1 to 77.1 mm.

Furthermore, Pan and Shuangguan (2005) also approved that no runoff generation under 1.5 mm min$^{-1}$ rainfall intensity at vegetation covered plot with 15 °slope in Loess Plateau. The largest rainfall amount we selected in our study is 35.2mm, which equals to approximately 0.05 mm min$^{-1}$ rainfall intensity and can be observed in **Figure explain 1** as follows, and the slops of our selected plots were approximately 5 °. Thus, we predicted that no runoff is generated during the time of our selected 5 rainfall events. We also suggested the possibility runoff after large rainfall events should be direct observation and would helpful for plantation species adaptability evaluation in the studied region.

According to plantation species and type selection, we also suggested that "Furthermore, water conservation measures, such as water-fertilizer pits (60 ×60 ×40 cm) (Wang et al., 2020), that can intercept any possible runoff after large rainfall events and deliver it to deep soil layers may be appropriate for the studied region." in "4.3 Implications for plantation species and type selection based on RRS uptake and plant transpiration" subsection in "**4 Discussion**" section (Page 23 Lines 534-536).

[Figure]

**Figure explain 1** The half-hour rainfall amount during DOY155-156. The rainfall during DOY 155-156 was considered as one rainfall event in the present study.

**References:**

Pan, C. Z., and Shangguan, Z. P.. Influence of forage grass on hydrodynamic characteristics of slope erosion. Journal of Hydraulic Engineering, 3, 371–377. https://doi.org/10.13243/j.cnki.slxb.2005.03.020 (In Chinese with English Abstract), 2005.

Huang, J., Wang, J., Zhao, X. N., Wu, P. T., Qi, Z. M., and Li, H. B.: Effects of permanent ground cover on soil moisture in jujube orchards under sloping ground: A simulation study, Agr Water Manage, 138, 68-77, 2014.

Zhao, X. I., Wu, P., Chen, X. L., Helmers, M. J., and Zhou, X. B.: Runoff and sediment yield under simulated rainfall on hillslopes in the Loess Plateau of China, Soil Res, 51, 50-58, 2013.

Wang, J., Fu, B. J., Wang, L. X., Lu, N., and Li, J. Y.: Water use characteristics of the common tree species in different plantation types in the Loess Plateau of China, Agr Forest Meteorol, 288, ARTN 108020, 10.1016/j.agrformet.2020.108020, 2020.

*2) Throughout the manuscript, there are also some instances where the term seems inappropriately use (e.g. only). I would suggest going through the entire paper and refining the language to more accurately reflect the result.*

**Response: Rewritten and clarified.** Thanks for your suggestion, the entire manuscript has been reviewed and the relative terms have been rewritten in the revised version.

For example, based on the suggestion by the other reviewer, the term "plant water consumption" and "rainwater uptake" has been revised to "plant transpiration" and "rainwater-recharged soil water", respectively, in the revised manuscript. The RRS was used as the abbreviation for "rainwater-recharged soil water" in the revised manuscript. And the **Title** of the revised manuscript has also been rewritten as "Differential response of plant transpiration to uptake of rainwater-recharged soil water for dominant tree species in the semiarid Loess Plateau"

For example, the "only" has also been deleted in the revised manuscript in "**Abstract**" section as follows: "In pure plantations, the relative response of daily normalized sap flow ($SF_R$) was significantly affected by RRS uptake proportion (RUP) and $\Psi_{pd}-\Psi_m$ for *H. rhamnoides*, and was significantly influenced by $\Psi_{pd}-\Psi_m$ for *P. tomentosa* ($P < 0.05$)."

*3) Potential/Reference Evapotranspiration is a key parameter indicator that reflect atmospheric evaporative demand, and also support some part of you conclusion. However, why the Reference evapotranspiration (ET0) was used in the study, because there are some other indicator also reflect the evaporative demand.*

**Response: Clarified and rewritten.** In response to this meaningful suggestion, the advantage of Reference evapotranspiration ($ET_0$) has been added in "2.2 Environmental parameter measurements and $ET_0$ calculation" subsection in "**2 Materials and methods**" section as follows: "$ET_0$, considering both aerodynamic characteristics and energy balance, was used to indicate atmospheric evaporative demand (Allen et al., 1998):"

Indeed, there are several Equations that calculated the potential or reference evapotranspiration. The $ET_0$ equation in the present study is used as the standard method by the FAO (Food and Agriculture Organization of the United Nations), and has been widely used for evaluate other $ET_0$ equations (Xiang et al., 2020). The advantage of the Equation that we used considered both aerodynamic aspects and energy balance, because evapotranspiration is a process that liquid water is converted vapor phase and

then the vapor moves. The detailed information can be observed in a review of difference of reference crop evapotranspiration in Xiang et al. (2020).

**References:**

Allen, R.G., Periera, L.S., Raes, D., and Smith, M.: Crop evapotranspiration: Guidelines for Computing Crop Requirements, Irrigation and Drainage paper NO.56, FAO, Rome, Italy, 300, 1998.

Xiang, K. Y., Li, Y., Horton, R., and Feng, H.: Similarity and difference of potential evapotranspiration and reference crop evapotranspiration - a review, Agr Water Manage, 232, 10.1016/J.Agwat.2020.106043, 2020.

*4) This manuscript should be looked over by a language editing service and/or a native English speaker - there are some grammatically incorrect and/or awkward phrasings.*

**Response: Rewritten.** Thanks for your suggestion; the entire revised manuscript has been reviewed and the language has been refined by *International Science Editing*.

[Figure]

**International Science Editing**
www.internationalscienceediting.com

DATE: January 26, 2022

Compuscript Ltd
T/A International Science Editing
Bay K, Shannon Industrial Park West
Shannon, Co Clare
Ireland
Phone +353 61 472818   Fax +353 61 472688

To whom it may concern,

The paper "Differential response of plant transpiration to uptake of rainwater-recharged soil water for dominant tree species in the semiarid Loess Plateau" by Yakun Tang was edited by International Science Editing. We were asked not to edit the references. Please contact us if you would like to view the edited paper.

Kindest regards,

David Cushley.

If the English is still not meet the standard, please give me another chance, I will revised the language by using another scientific editing service company again.

**Minor Comments:**

*1) Lines 22 "only" is too arbitrary*

**Response: Deleted.** This sentence has been rewritten in "**Abstract**" section as follows: "In pure plantations, the relative response of daily normalized sap flow ($SF_R$) was significantly affected by RRS uptake proportion (RUP) and $\Psi_{pd}-\Psi_m$ for *H. rhamnoides*, and was significantly influenced by $\Psi_{pd}-\Psi_m$ for *P. tomentosa* ($P < 0.05$)."

*2) Lines 30-32 "Regardless of sensitivity to rainfall pulses" ? this short sentence should be rewritten.*

**Response: Rewritten.** Thanks for this meaningful suggestion, this sentence has been rewritten in "**Abstract**" section as follows: "These results indicate that mixed afforestation enhanced the influence of RRS uptake to plant transpiration for these different rainfall pulse sensitive plants."

*3) Lines 54-57 The "water uptake" should also be clearly described.*

**Response: Rewritten.** In response to this meaningful suggestion, the sentence has been rewritten in "**1 Introduction**"as follows: "The controversial rainfall pulse response between RRS uptake and plant transpiration may be mainly attributed to an inconsistent influence of plant leaf physiological characteristics (West et al., 2007), root morphology adjustment (Wang et al., 2020), or environmental conditions (Tfwala et al., 2019) on these two water processes."

*4) Lines 69-71 the author should be clarified this sentence for pure or coexisting species? Because the similar meaning and sentence can be observed at Lines 57-60.*

**Response: Revised.** Thanks for this meaningful suggestion, this sentence has been revised in "**1 Introduction**" section as follows: "Rainfall pulses have been observed to relieve or eliminate water competition among coexisting species and thus maintain or increase plant transpiration in some water limited regions (Wang et al., 2020; Tfwala et al., 2019)."

Indeed, this sentence should be clarified the influence of rainfall pulses on water competition among

coexisting species.

*5) Lines 131-132 Please clarify why the Reference evapotranspiration (ET0) was used in the study, as a large number of indicators can reflect atmospheric evaporative demand.*

**Response: Clarified and rewritten.** In response to this meaningful suggestion, the advantage of Reference evapotranspiration ($ET_0$) has been added in "2.2 Environmental parameter measurements and $ET_0$ calculation" subsection in "**2 Materials and methods**" section as follows: "$ET_0$, considering both aerodynamic characteristics and energy balance, was used to indicate atmospheric evaporative demand (Allen et al., 1998):". The detailed explanation can be observed the **Tables and captions** at the end of this file.

*6) Lines 213-214 This sentence is nonsense and should be deleted.*

**Response: Deleted and Rewritten.** Thanks for this meaningful suggestion, this sentence has been deleted and rewritten in "2.6.2 Calculation of RRS uptake proportion and water sources from different soil layers" subsection in "**2 Materials and methods**" section as follows: "In addition to RUP, the water uptake proportions from different soil layers were calculated on the first day after a rainfall event using the MixSIR program, to complement the analysis of plant water source variations in response to rainfall pulses. The RUP method only calculated the proportion of recent rainwater in the plant stem and did not include soil water before the recent rainfall event (Gebauer and Ehleringer, 2000; Cheng et al., 2006). The water taken up from different soil layers by the plant is a mixture of soil water before the recent rainfall event and the recent rainwater."

**References:**

Cheng, X. L., An, S. Q., Li, B., Chen, J. Q., Lin, G. H., Liu, Y. H., Luo, Y. Q., and Liu, S. R.: Summer rain pulse size and rainwater uptake by three dominant desert plants in a desertified grassland ecosystem in northwestern China, Plant Ecol, 184, 1-12, 2006.

Gebauer, R. L. E., and Ehleringer, J. R.: Water and nitrogen uptake patterns following moisture pulses in a cold desert community, Ecology, 81, 1415-1424, 2000.

*7) Line 306 There are 7 Figures in the paper and the Tables 1-4 are the statistical analysis. These Tables are unnecessary list in the paper and its better remove to Supplementary file.*

**Response: Rewritten.** Thanks for this suggestion, all the Tables have been removed to Supplementary file. The Tables 1-4 have been renamed to Tables S 4-6, respectively, because the origin Tables 1-2 has been combined into Table S4. The detailed Table S can be observed in the revised *Supplementary file*.

*8) Line 415 Is synchronization correct in this sentence ? It's not correct, you should check it.*

**Response: Rewritten.** Indeed, the word "synchronization" is not correct in this sentence. The sentence has been rewritten in "4.1 RRS uptake enhances plant transpiration for *H. rhamnoides* but not *P. tomentosa* in pure plantations" subsection in "**4 Discussion**" section as follows: "The asynchronization between RRS uptake and plant transpiration for *J. osteosperma* was mainly attributed to the uptake of RRS by plants being unable to reverse the cavitation in its roots and stems (Grossiord et al., 2017; West et al., 2007)."

*9) Lines 478-480 Table S3 does not indicated the relationship between rainfall amount and water source proportion from deep soil layer.*

**Response: Deleted and rewritten.** Thanks for this meaningful suggestion, this sentence has been deleted and rewritten in "4.2 RRS uptake enhances plant transpiration for coexisting species in mixed plantation" subsection in "**4 Discussion**" section as follows: "Similar to other studies in the Loess Plateau (Wang et al., 2020; Wu et al., 2021), the deep soil layer exhibited lower SW than other soil layers in all plantation types in the present study (Fig. 1, Table S3). Jia et al. (2017) and Wang et al. (2020) attributed the lower SW in deep soil layers to the imbalance between rainwater replenishment and plant uptake of water from this layer. In addition, plants may expend more energy to uptake water from deep compared to shallow soil layers (Schenk, 2008), especially when the deep soil layer exhibits lower SW."

Indeed, the previous sentence is inappropriate interpretation, mainly because the decreased deep soil water source may influenced by both previous soil water conditions and precipitation amount. In

addition, the Table S3 in the previous manuscript is not the summary of relationship between rainfall amount and water source proportion from different soil layers. Therefore, we deleted the original sentence and rewritten these sentences mentioned above.

**Tables and captions**

**Table S1.** Plant height, trunk diameter, and estimated sapwood width for *H. rhamnoides* and *P. tomentosa* in both pure and mixed plantations.

| Plantation type | No. | Height (m) | Trunk diameter (mm) | Sapwood width (mm) |
|---|---|---|---|---|
| *H. rhamnoides* in pure plantation | 1 | 3.95 | 45 | 9 |
| | 2 | 4.26 | 53 | 11 |
| | 3 | 4.05 | 51 | 10 |
| | 4 | 4.13 | 49 | 9 |
| | 5 | 3.98 | 50 | 10 |
| | 6 | 4.1 | 51 | 11 |
| | 7 | 4.3 | 57 | 12 |
| | 8 | 3.86 | 44 | 9 |
| | 9 | 3.92 | 53 | 11 |
| *P. tomentosa* in pure plantation | 1 | 4.41 | 58 | 17 |
| | 2 | 3.9 | 52 | 9 |
| | 3 | 3.92 | 56 | 16 |
| | 4 | 4.35 | 56 | 17 |
| | 5 | 4.59 | 58 | 16 |
| | 6 | 4.2 | 53 | 13 |
| | 7 | 4.29 | 54 | 15 |
| | 8 | 3.86 | 51 | 9 |
| | 9 | 3.98 | 52 | 11 |
| *H. rhamnoides* in mixed plantation | 1 | 4.36 | 52 | 12 |
| | 2 | 3.9 | 49 | 11 |
| | 3 | 4.23 | 51 | 12 |
| | 4 | 4.5 | 56 | 13 |
| | 5 | 4.73 | 55 | 14 |
| | 6 | 3.96 | 49 | 11 |
| | 7 | 4 | 51 | 12 |
| | 8 | 4.52 | 53 | 12 |
| | 9 | 4.39 | 52 | 12 |
| *P. tomentosa* in mixed plantation | 1 | 4.12 | 53 | 11 |
| | 2 | 3.75 | 46 | 9 |
| | 3 | 4.5 | 57 | 13 |
| | 4 | 4.21 | 53 | 11 |
| | 5 | 4.2 | 53 | 11 |
| | 6 | 4.16 | 51 | 10 |
| | 7 | 3.8 | 45 | 9 |
| | 8 | 4.95 | 59 | 13 |
| | 9 | 4.16 | 51 | 10 |

The sapwood width was estimated through the equation established through 12 unmonitored individual core samples for specific species with different diameters. The core sample was obtained using an increment borer, and the colour difference between sapwood and heartwood was large. The equation between trunk diameter (mm) and sapwood width (mm) was $y=0.248x-2.296$ $R^2=0.84$ $p<0.01$ for *H. rhamnoides* in pure plantation; $y=0.348x-5.98$ $R^2=0.78$ $P<0.01$ for *H. rhamnoides* in mixed plantation; $y=1.126x-47.66$ $R^2=0.83$ $P<0.01$ for *P. tomentosa* in pure plantation; $y=0.317x-5.71$ $R^2=0.939$ $P<0.01$ for *P. tomentosa* in mixed plantation.

**Table S2.** Independent-sample $t$-test parameters for predawn ($\Psi_{pd}$), midday ($\Psi_m$), and gradient of leaf water potential ($\Psi_{pd}-\Psi_m$) between the first and second day after each rainfall amount.

| | Rainfall amount (mm) | df | $\Psi_{pd}$ | | $\Psi_m$ | | $\Psi_{pd}-\Psi_m$ | |
|---|---|---|---|---|---|---|---|---|
| | | | $t$ | $p$ | $t$ | $p$ | $t$ | $p$ |
| *H. rhamnoides* in pure plantation | 3.4 | 4 | 0.18 | 0.87 | 1.21 | 0.29 | -2.5 | 0.07 |
| | 7.9 | 4 | 0.33 | 0.75 | 0.79 | 0.58 | -8.01 | 0.47 |
| | 15.4 | 4 | 0.85 | 0.44 | 0.27 | 0.8 | 0.21 | 0.85 |
| | 24 | 4 | 0.97 | 0.39 | -0.67 | 0.54 | 2.13 | 0.1 |
| | 35.2 | 4 | -0.09 | 0.93 | -7.1 | 0.52 | 0.28 | 0.79 |
| *P. tomentosa* in pure plantation | 3.4 | 4 | 0.88 | 0.43 | 0.66 | 0.55 | 0.81 | 0.47 |
| | 7.9 | 4 | 0.34 | 0.08 | 0.75 | 0.49 | -1.8 | 0.14 |
| | 15.4 | 4 | 0.23 | 0.83 | 0.73 | 0.51 | -0.82 | 0.46 |
| | 24 | 4 | -2.08 | 0.11 | 1.14 | 0.32 | -0.85 | 0.45 |
| | 35.2 | 4 | -1.67 | 0.17 | 1.15 | 0.31 | -2.22 | 0.09 |
| *H. rhamnoides* in mixed plantation | 3.4 | 4 | 2.53 | 0.07 | 1.4 | 0.24 | -0.6 | 0.58 |
| | 7.9 | 4 | 1.24 | 0.28 | 2.02 | 0.11 | -1.87 | 0.14 |
| | 15.4 | 4 | -0.9 | 0.42 | 0.96 | 0.39 | -1.29 | 0.27 |
| | 24 | 4 | 1.74 | 0.16 | 2.04 | 0.11 | -1.22 | 0.29 |
| | 35.2 | 4 | 1.89 | 0.13 | 2.57 | 0.06 | -0.29 | 0.78 |
| *P. tomentosa* in mixed plantation | 3.4 | 4 | 0.07 | 0.95 | 1.9 | 0.13 | -0.35 | 0.72 |
| | 7.9 | 4 | 0.81 | 0.46 | 0.96 | 0.39 | -0.46 | 0.67 |
| | 15.4 | 4 | 0.7 | 0.52 | 2.12 | 0.1 | -0.53 | 0.62 |
| | 24 | 4 | 1.85 | 0.14 | 0.74 | 0.49 | 0.48 | 0.66 |
| | 35.2 | 4 | 2.23 | 0.09 | 1.21 | 0.3 | 0.55 | 0.61 |

**Table S3** The average (mean ± SD) and coefficients of variation (CVs, SD/mean) of soil water $\delta^{18}O$ and $\delta D$ on the first day after 5 selected rainfall events, and daily soil water content (SW) from DOY 152 to 273 (1 June to 30 September) in *H. rhamnoides* pure plantation, *P. tomentosa* pure plantation, and *H. rhamnoides*–*P. tomentosa* mixed plantation.

| | Soil depth | soil water $\delta^{18}O$ (‰) | | soil water $\delta D$ (‰) | | SW ($m^3$ $m^{-3}$) | |
|---|---|---|---|---|---|---|---|
| | | average | CV | average | CV | average | CV |
| *H. rhamnoides* pure plantation | 0–30 cm | -5.61±1.57 | 27.99 | -41.53±11.68 | 28.12 | 0.13±0.025 | 19.23 |
| | 30–100 cm | -7.14±0.92 | 12.89 | -52.37±6.47 | 12.35 | 0.1±0.012 | 12 |
| | 100–200 cm | -9.3±0.69 | 7.42 | -68.66±3.53 | 5.14 | 0.09±0.006 | 6.67 |
| *P. tomentosa* pure plantation | 0–30 cm | -5.43±1.69 | 31.12 | -42.08±11.91 | 28.3 | 0.13±0.026 | 20 |
| | 30–100 cm | -7.49±0.73 | 9.75 | -51.34±4.56 | 8.88 | 0.09±0.008 | 8.89 |
| | 100–200 cm | -9.39±0.34 | 3.62 | -67.36±3.79 | 5.63 | 0.08±0.005 | 6.25 |
| Mixed plantation | 0–30 cm | -5.68±1.73 | 30.46 | -41.67±10.67 | 25.61 | 0.12±0.021 | 17.5 |
| | 30–100 cm | -6.57±1.08 | 16.44 | -47.8±5.78 | 12.09 | 0.1±0.011 | 11 |
| | 100–200 cm | -9.07±0.5 | 5.51 | -64.47±2.45 | 3.8 | 0.09±0.005 | 5.56 |

There are 45, 30, and 30 data for calculated the average water $\delta^{18}O$ and $\delta D$ of shallow, middle, and deep soil layer in each plantation, respectively. The absolute value was used for CVs of soil water $\delta^{18}O$ and $\delta D$ calculation.

**Table S4.** Repeated ANOVA (ANOVAR) parameters for the relative response of normalized sap flow ($SF_R$) and rainwater-recharged soil water uptake proportion (RUP) after rainfall pulses of *H. rhamnoides* and *P. tomentosa* (n = 30).

| | Variation source | df | $SF_R$ | | RUP | |
|---|---|---|---|---|---|---|
| | | | F | p | F | p |
| Pure plantation | Rainfall | 4 | 97.91 | <0.001 | 385.02 | <0.01 |
| | Species | 1 | 121.13 | <0.001 | 21.02 | <0.05 |
| | Rainfall $\times$ Species | 4 | 27.35 | <0.001 | 0.83 | 0.52 |
| Mixed plantation | Rainfall | 4 | 489.9 | <0.001 | 17696.38 | <0.01 |
| | Species | 1 | 70.38 | <0.001 | 4089.12 | <0.01 |
| | Rainfall $\times$ Species | 4 | 249.17 | <0.001 | 1776.62 | <0.01 |
| *H. rhamnoides* | Rainfall | 4 | 42.63 | <0.001 | 496.72 | <0.01 |
| | Plantation type | 1 | 337.09 | <0.001 | 360.16 | <0.01 |
| | Rainfall $\times$ Plantation type | 4 | 215.43 | <0.001 | 17.62 | <0.01 |
| *P. tomentosa* | Rainfall | 4 | 10.05 | <0.001 | 1969.3 | <0.01 |
| | Plantation type | 1 | 32.36 | <0.01 | 54.83 | <0.01 |
| | Rainfall $\times$ Plantation type | 4 | 19.12 | <0.001 | 208.06 | <0.01 |

*df* = degree of freedom, Plantation type = pure and mixed plantation for each species. Pure and Mixed plantation indicate the result of $SF_R$ and RUP for both species in different plantation types, respectively; *H. rhamnoides* and *P. tomentosa* indicate the mixed afforestation effect on $SF_R$ and RUP for these species.

**Table S5.** Repeated ANOVA (ANOVAR) parameters for water uptake proportion from shallow (0–30 cm), middle (30–100 cm), and deep (100–200 cm) soil layer for *H. rhamnoides* and *P. tomentosa* (n = 30).

| | Variation source | $df$ | 0–30cm | | 30–100cm | | 100–200cm | |
|---|---|---|---|---|---|---|---|---|
| | | | $F$ | $p$ | $F$ | $p$ | $F$ | $p$ |
| Pure plantation | Rainfall | 4 | 153.45 | <0.01 | 145.04 | <0.01 | 176.79 | <0.01 |
| | Species | 1 | 8.69 | <0.05 | 10.56 | <0.05 | 11.08 | <0.05 |
| | Rainfall ×Species | 4 | 129.89 | <0.01 | 112.46 | <0.01 | 4.99 | <0.01 |
| Mixed plantation | Rainfall | 4 | 1.5 | 0.41 | 2.3 | 0.11 | 18.34 | <0.01 |
| | Species | 1 | 2.2 | 0.21 | 1.48 | 0.29 | 3.9 | 0.12 |
| | Rainfall ×Species | 4 | 0.9 | 0.48 | 2.41 | 0.09 | 1.9 | 0.16 |
| *H. rhamnoides* | Rainfall | 4 | 2.05 | 0.14 | 1.51 | 0.25 | 85.46 | <0.01 |
| | Plantation type | 1 | 1.07 | 0.36 | 1.32 | 0.32 | 10.08 | <0.05 |
| | Rainfall × Plantation type | 4 | 0.62 | 0.66 | 1.39 | 0.28 | 5.59 | <0.01 |
| *P. tomentosa* | Rainfall | 4 | 14.72 | <0.01 | 71.59 | <0.01 | 19.46 | <0.01 |
| | Plantation type | 1 | 4.1 | 0.12 | 5.68 | 0.08 | 123.27 | <0.01 |
| | Rainfall × Plantation type | 4 | 9.55 | <0.01 | 85.29 | <0.01 | 9.35 | <0.01 |

$df$ = degree of freedom, Plantation type = pure and mixed plantation for each species. Pure and Mixed plantation indicate the result of water sources from different soil layers for both species in different plantation types, respectively; *H. rhamnoides* and *P. tomentosa* indicate the mixed afforestation effect on water sources from different soil layers for these species.

**Table S6.** Repeated ANOVA (ANOVAR) parameters for predawn ($\Psi_{pd}$), midday leaf water potential ($\Psi_m$), and leaf water potential gradient ($\Psi_{pd}-\Psi_m$) for *H. rhamnoides* and *P. tomentosa* (n = 30).

| | Variation source | df | $\Psi_{pd}$ F | p | $\Psi_m$ F | p | $\Psi_{pd}-\Psi_m$ F | p |
|---|---|---|---|---|---|---|---|---|
| Pure plantation | Rainfall | 4 | 4.02 | <0.05 | 24.44 | <0.01 | 47.88 | <0.01 |
| | Species | 1 | 182.74 | <0.01 | 4.9 | <0.05 | 969.97 | <0.01 |
| | Rainfall × Species | 4 | 3.24 | <0.05 | 2.08 | 0.13 | 18.68 | <0.01 |
| Mixed plantation | Rainfall | 4 | 0.66 | 0.63 | 25.54 | <0.01 | 82.49 | <0.01 |
| | Species | 1 | 0.12 | 0.75 | 127.3 | <0.01 | 3420.1 | <0.01 |
| | Rainfall × Species | 4 | 1.8 | 0.18 | 3.7 | <0.05 | 35.92 | <0.01 |
| *H. rhamnoides* | Rainfall | 4 | 7.14 | <0.01 | 19.64 | <0.01 | 3.59 | <0.05 |
| | Plantation type | 1 | 27.05 | <0.01 | 496.66 | <0.01 | 1278.96 | <0.01 |
| | Rainfall × Plantation type | 4 | 1.69 | 0.202 | 3.32 | <0.05 | 6.66 | <0.01 |
| *P. tomentosa* | Rainfall | 4 | 30.78 | <0.01 | 12.39 | <0.01 | 7.38 | <0.01 |
| | Plantation type | 1 | 792.77 | <0.01 | 2.97 | 0.16 | 634.12 | <0.01 |
| | Rainfall × Plantation type | 4 | 3.8 | <0.05 | 0.09 | 0.98 | 3.83 | <0.05 |

df = degree of freedom, Plantation type = pure and mixed plantation for each species. Pure and Mixed plantation indicate the result of leaf water potential for both species in different plantation types, respectively; *H. rhamnoides* and *P. tomentosa* indicate the mixed afforestation effect on leaf water potential for these species.

**Table S7.** Regression of reference evapotranspiration ($ET_0$) and relative response of normalized sap flow ($SF_R$).

| Independent factors | *H. rhamnoides* in pure plantation | | *H. rhamnoides* in mixed plantation | | *P. tomentosa* in pure plantation | | *P. tomentosa* in mixed plantation | |
|---|---|---|---|---|---|---|---|---|
| | $R^2$ | p | $R^2$ | p | $R^2$ | p | $R^2$ | p |
| $ET_0$ | 0.18 | 0.47 | 0.11 | 0.59 | 0.44 | 0.22 | 0.39 | 0.26 |
| Relative response of $ET_0$ | 0.35 | 0.32 | 0.61 | 0.12 | 0.12 | 0.56 | 0.25 | 0.4 |

The regression equation is y=ax+b for all equations in this Table. Relative response of $ET_0$ is calculated as the same $SF_R$ in Eq. (4) in the manuscript, with before and the first day after rainfall event parameter is $ET_0$ instead.

**Table S8.** Parameters of allometric equation and average (mean ± SD) estimated biomass of leaf, branches, wood, and roots of *H. rhamnoides* and *P. tomentosa* in pure and mixed plantations (n=6).

| Species | | *a* | *b* | Biomass in pure plantation | Biomass in mixed plantation |
|---|---|---|---|---|---|
| | leaf | 0.017 | 0.541 | 0.51 ±0.02 | 0.55 ±0.04 |
| | branches | 0.013 | 0.042 | 0.16 ±0.05 | 0.14 ±0.01 |
| *H. rhamnoides* | wood | 0.036 | 0.721 | 2.4 ±0.09 | 2.6 ±0.07 |
| | roots | 0.019 | 0.732 | 1.51 ±0.06 | 1.79 ±0.04 |
| | total biomass | | | 4.58 ±1.01 | 5.08 ±1.13 |
| | leaf | 0.052 | 0.621 | 1.21 ±0.05 | 1.58 ±0.09 |
| | branches | 0.025 | 0.81 | 1.35 ±0.04 | 1.32 ±0.06 |
| *P. tomentosa* | wood | 0.0492 | 0.832 | 4.22 ±0.11 | 4.73 ±0.13 |
| | roots | 0.031 | 0.791 | 2.02 ±0.06 | 2.75 ±0.1 |
| | total biomass | | | 8.8 ±1.39 | 10.38 ±1.55 |

The allometric equation is $Y=a(D^2H)^b$, $Y$ is biomass (kg), $D$ is trunk diameter measured at 1.3 m above the ground (cm), $H$ is tree height (m). Six standard individuals of *H. rhamnoides* and *P. tomentosa* in pure and mixed plantations were selected for average $Y$ calculation.

---

## Author Comment (AC2)

Authors' responses to Reviewers comments on the manuscript of "Differential response of plant transpiration to uptake of rainwater-recharged soil water for dominant tree species in the semiarid Loess Plateau". Manuscript ID: hess-2021-351.

**Dear Reviewer,**

We deeply appreciate you for giving us an opportunity to revise our manuscript. The point-to-point responses (responses in upright Roman) to the Reviewer comments (*original comment and query in Itali*) can be observed in the PDF file named "**Response to Reviewer 2**". In addition, the revised manuscript is highlighted the changes by using the red colored text in the manuscript (track-changes version), and append at the end of this file. We know that the revised manuscript is no need to upload at this time, we added it at this time to facilitate review our responses and corresponding revisions. The revised manuscript at the end of this PDF file can be ignored if it is no need.

**Reviewer 2**

**Major Comments:**

*1. Personally, I find the terms 'rainwater uptake' and 'water consumption' (both central to this manuscript) rather ambiguous. I would recommend using 'transpiration' instead of 'water consumption. On the other hand, the term rainwater uptake can be confusing, as it seems to suggest that these trees take up water directly from rainfall. Some trees can indeed take up rainwater through their leaves, but this is not the case for the species included in this study. In my opinion, it would be better to refer instead to the 'uptake of recently recharged soil water' or similar (uptake of soil water that has been recharged from a recent rainfall event).*

**Response: Rewritten.** Thanks for this meaningful suggestion, the terms of "water consumption" and "rainwater uptake" have been changed to "transpiration" and "uptake of rainwater-recharged soil water", respectively, throughout the revised manuscript. For example, the manuscript **Title** has been rewritten as "Differential response of plant transpiration to uptake of rainwater-recharged soil water for dominant tree species in the semiarid Loess Plateau" (Page 1 Lines 1-2).

In addition, the "RRS", abbreviation of "rainwater-recharged soil water", was used in this study for convenient reading and understanding. This abbreviation has been added at the first sentence in "**Abstract**" section as "Whether uptake of rainwater-recharged soil water (RRS) can increase plant transpiration in response to rainfall pulses requires investigation to evaluate the plant adaptability, especially in water limited regions where rainwater is the only replenishable soil water source." (Page 1 Lines 14-16). And this abbreviation has also been added at the first sentence in "**Introduction**" section as "Rainwater-recharged soil water (RRS) uptake by plants and plant transpiration in response to rainfall pulses drive the survival of plant species and ecosystem ecohydrological processes, especially in arid and semiarid regions where rainwater is the only replenishable soil water source (Berkelhammer et al., 2020; Gebauer and Ehleringer, 2000; West et al., 2012)." (Page 2 Lines 39-42).

The illustration to "rainwater-recharged soil water (RRS)" has also been added in the "**1 Introduction**" section as follows: "Generally, RRS uptake after a rainfall pulse refers to the root uptake of soil water that was recharged by recent rainwater, and can be quantified through water stable isotopes (Cheng et al., 2006; Meier et al., 2018)." (Page 2 Lines 42-44)

Furthermore, the abbreviation of "rainwater recharged soil water" is used for "RRS" but not "RRSW", mainly because the "RRSW" has been used as the abbreviation of "resistance rivet spot welding" based on research in *web of science*.

*2. It would be very helpful if the authors could provide some additional information on the two studied tree species. The authors write (L.82-84): 'Hippophae rhamnoides and Populus tomentosa are typical dominant tree species, with high survival rate and drought tolerance, and occupy nearly 30% of the plantation area in this region (Liu et al., 2017; Tang et al., 2019)'. Could you give some species-specific information on e.g. their phenology or root system? How do the species differ, and are there any reasons to believe that they might respond differently to rain pulses in terms of transpiration and water source partitioning? Do you have any hypotheses? In addition, I would suggest the authors check the scientific names of the species. According to the World Flora Online, Hippophae rhamnoidesis not an accepted name but a synonym of Elaeagnus rhamnoides (L.) A.Nelson.*

**Response: Added and Corrected.** In response to this meaningful suggestion, the relative sentences of species-specific information, two hypotheses, and the correct scientific names of the species have been added and corrected in the revised manuscript.

**Firstly**, in response to plant phynology and species-specific differ for these two studied plantation species, the relative sentences have been rewritten in "**1 Introduction**" section as follows: "*Hippophae rhamnoides* subsp. *sinensis* and *Populus tomentosa* are typical deciduous broadleaved tree species, with similar leaf expansion (April) and falling (November) periods, and occupy nearly 30% of the plantation area in this region (Liu et al., 2017; Tang et al., 2019). Our previous study indicated that *H. rhamnoides* generally took up soil water from 0−40 cm or > 100 cm soil depths and adopted large leaf water potential variation to cope with varied soil water conditions in this region (Tang et al., 2019). Meanwhile, *P. tomentosa* generally took up soil water from > 100 cm soil depth throughout the growing season in varied soil water conditions (Xi et al., 2018). In addition, mixed plantations of these two species were widely promoted by local government due to the higher soil and water conservation capacity than pure plantations in the original afforestation stage (Tang et al., 2019; Wang et al., 2020). Tang et al. (2019) also suggested that mixed afforestation with *Ulmus pumila*, a deciduous broadleaved tree species with similar leaf growth phenology to *H. rhamnoides*, increased the water source from 0−40 cm soil depth and the leaf water potential variation for *H. rhamnoides* compared with these values for this species in pure plantation." (Pages 3-4 Lines 81-93).

In previous studies, the water sources from soil depths were different between the two studied plantation species (Xi et al., 2018; Tang et al., 2019), with *H. rhamnoides* generally shifted its main water sources depending on soil water conditions, however, *P. tomentosa* generally absorbed relative stable deep soil water throughout the growing season in varied soil water conditions. These two plantation species exhibited similar leaf growth phenology such as leaf expansion (April) and falling (November) time. In addition, mixed with other plant species may altered the water sources and leaf water potential for *H. rhamnoides* or *P. tomentosa*. All of these different soil water uptake patterns for these two species in pure plantation, as well as the possibility of altered plant water sources and leaf water potential in mixed plantation, may affect the influence of rainwater-recharged soil water (RRS)

uptake and leaf water potential on plant transpiration after rainfall pulses. And these affect may also different in pure and mixed plantations. Thus, "The specific objectives were as follow: (1) to investigate the influence of RRS uptake and leaf water potential on plant transpiration after rainfall events in pure plantation, and (2) to assess the mixed afforestation effect on these influences." (Page 4 Lines 100-102). And the two corresponding hypotheses can be observed in the second answer as follows.

**Secondly**, the two hypotheses have been added in the revised manuscript in "**1 Introduction**" section as follows: "Based on variations of plant water uptake from soil layers and/or leaf water potential for these species in Xi et al. (2018) and Tang et al. (2019), we hypothesize that (1) the influence of RRS uptake and leaf water potential on plant transpiration may differ for these species in pure plantations, and (2) these influences may differ for specific species in pure and mixed plantations." (Page 4 Lines 102-106).

The first hypothesis is mainly based on the majority of plant water sources from soil depths were different for these plantation species in pure plantation, which can be observed in Tang et al. (2019) and Xi et al. (2018). Thus, we hypothesis that the influence of RRS uptake on plant transpiration for these species may be different.

The second hypothesis is mainly based on our previous studies indicated that mixed afforestation between *H. rhamnoides* and *Ulmus pumila* altered both the majority of plant water sources from shallow soil layer and the leaf water potential for *H. rhamnoides*, compared with these values for this species in pure plantation (Tang et al., 2019). Thus, we hypothesis that mixed afforestation may alter the influence of RRS uptake and leaf water potential on plant transpiration for specific plant species, compared with these influences in pure plantation.

In addition, these tow hypotheses have also been test in "**4 Discussion**" section. In response to the first hypothesis, the relative sentence has been added: "Consistent with the first hypothesis, the influence of RRS uptake and physiological adjustment on plant transpiration was different for these species in pure plantations." (Page 20 Lines 445-446).

In response to the second hypothesis, the relative sentence has been added in "**4 Discussion**" section:

"In addition to these adjustments for specific plant species in mixed plantation, the significant influence of RUP and $\Psi_{pd}-\Psi_{m}$ on $SF_R$ for *P. tomentosa* in mixed plantation was also consistent with the second hypothesis (Fig. 6)" (Page 22 Lines 489-491).

**Thirdly**, in response to plant root distribution of these species, the plant root distribution was conducted in our present study. The method to investigate the plant root distribution were in the "2.5 Plant fine root investigation" subsection in "**2 Materials and methods**" section. (Page 9 Lines 220-227).

The results can be observed as in **Figure S2** as follows. The detailed sentences of these results to illustrated the plant water sources of these two species can be observed in "4.1 RRS uptake enhances plant transpiration for *H. rhamnoides* but not *P. tomentosa* in pure plantations" subsection in "**4 Discussion**" section as follows: "This may be mainly due to the greater proportions of fine root surface area distributed in the shallow soil layer for *H. rhamnoides* ($40.85 \pm 3.14\%$) compared to *P. tomentosa* ($21.94 \pm 2.3\%$) (Fig. S4)." (Page 19 Lines 422-424).

Also, in "4.2 RRS uptake enhances plant transpiration for coexisting species in mixed plantation" subsection in "**4 Discussion**" section as follows: "Mixed afforestation significant increased $\Psi_{pd}$ for *P. tomentosa*, possibly due to the advantage of access to soil moisture recharged by rainwater through an increased root surface area in the shallow soil layer for this species in the mixed plantation (Fig. S4)." (Page 21 Lines 484-486).

Thus, the description of root distribution between these two species was not emphasized "**1 Introduction**" section.

[Figure]

**Figure S4.** Variation in average (mean ± SD) surface area of fine root at different soil depths for *H. rhamnoides* and *P. tomentosa* in pure (a) and mixed (b) plantations. Error bars indicate the standard deviation (n = 3).

**Fourthly,** thanks for this careful examination; the Latin name of previous "*Hippophae rhamnoidesis*" has been corrected to "*Hippophae rhamnoides* subsp. *sinensis*" at the first appearance in "**Abstract**" and "**1 Introduction**" section, respectively, in the revised manuscript. Then, the acronym name "*H. rhamnoides*" was used throughout the manuscript.

The revised name of *Hippophae rhamnoides* subsp. *sinensis* can be observed in studies in Huang et al. (2018), and also in *World Flora Online* as follows:

[Figure]

**References:**

Huang, J. H., Li, G. Q., Li, J., Zhang, X. Q., Yan, M. J., and Du, S.: Projecting the Range Shifts in Climatically Suitable Habitat for Chinese Sea Buckthorn under Climate Change Scenarios, Forests, 9, 2018.

Tang, Y. K., Wu, X., Chen, C., Jia, C., and Chen, Y. M.: Water source partitioning and nitrogen facilitation promote coexistence of nitrogen-fixing and neighbor species in mixed plantations in the semiarid Loess Plateau, Plant Soil, 445, 289-305, 10.1007/s11104-019-04301-9, 2019.

Xi, B. Y., Wang, Y., Jia, L. M., Bloomberg, M., Li, G. D., and Di, N.: Characteristics of fine root system and water uptake in a triploid Populus tomentosa plantation in the North China Plain: Implications for irrigation water management, Agr Water Manage, 117, 83-92, 2013.

*3. The authors have done extensive field and lab work, which is extremely valuable. However, I find the material and method section a bit hard to follow given not only the number of measurements but also the use of multiple approaches to address the same question (for example RUP – rainwater uptake proportion - vs MixSIR, or MIXSir with 7 soil depth intervals vs. MIXSir with 3 soil layers). This affects as well the interpretation of the results. Therefore, I would suggest the authors clarify the different steps in the methodology better, whether the chosen approaches are complementary, and how.*

**Response: Added and Clarified.** Thanks for these meaning and helpful suggestions. The number of measurements and the complementary of RUP calculation and MixSIR method were added and clarified in "**2 Materials and methods**" section in the revised manuscript.

   **Firstly**, the sample numbers for plant stems and soil water collection have been added in the revised

manuscript in "2.4 Rainwater, plant stem, soil water, and leaf sample collection and measurement" subsection in "**2 Materials and methods**" section as follows: "There were 180 stem and 945 soil samples for water extraction, and 180 leaf samples for $\Psi_{pd}$ and $\Psi_m$ measurement, respectively." (Page 8 Lines 206-207).

Taken plant stem samples as an example: For *H. rhamnoides* in pure plantation: 1 stem for each individual in each plot$\times$3 individual in each plot$\times$3 plots for this plantation $\times$5 rainfall events= 45 stmes. Thus, the total plant stems were: 45 for *H. rhamnoides* in pure plantation+45 for *H. rhamnoides* in mixed plantation+45 for *P. tomentosa* in pure plantation+45 for *P. tomentosa* in mixed plantation=180 plant stems.

**Secondly**, the complementary between RUP and MixSIR method have been added in the revised manuscript in "2.6.2 Calculation of RRS uptake proportion and water sources from different soil layers" subsection in "**2 Materials and methods**" section as follows: "In addition to RUP, the water uptake proportions from different soil layers were calculated on the first day after a rainfall event using the MixSIR program, to complement the analysis of plant water source variations in response to rainfall pulses. The RUP method only calculated the proportion of recent rainwater in the plant stem and did not include soil water before the recent rainfall event (Gebauer and Ehleringer, 2000; Cheng et al., 2006). The water taken up from different soil layers by the plant is a mixture of soil water before the recent rainfall event and the recent rainwater." (Pages 10-11 Lines 265-270).

The RUP is the core analysis in the present study, which is calculated as the proportion of rainwater in plant stem (Gebauer and Ehleringer, 2000; Cheng et al., 2006), and indicated the uptake of rainwater-recharged soil water proportion (RUP) by plant from a recent rainfall event. The water uptake proportion from different soil layers is calculated through MixSIR method, and is used to complement analysis the plant water sources variation after rainfall pulses for *H. rhamnoides* or *P. tomentosa*. The difference of these two methods can be observed in **Figure explain 1** as follows. Indeed, the soil water after a recent rainfall event is the mixture of soil water before recent rainfall event and recent rainwater (**Figure explain 1**). The basic difference between RUP and results from MixSIR calculation, is the

former method only calculated the proportion of rainwater (rainwater-recharged soil water) in plant stem after a recent rainfall event and not including soil water before rainfall event.

[Figure]

**Figure explain 1** The schematic of plant uptake of rainwater-recharged soil water from a recent rainfall event and plant water source from 3 soil layers after rainfall event.

In the present study, the RUP and water sources from 3 soil layers is used to illustrate the mixed afforestation of these two species effect on plant water sources. The results indicated that mixed afforestation significantly increased the RUP for these two plant species, and significantly decreased the water uptake proportion from deep soil layer ($P < 0.05$).

The RUP and water sources from 3 soil layers were used to illustrate the two types of plant adaptation to cope with resource competition: (1) increased competition to rainwater-recharged soil water (RRS) from a recent rainfall event; and (2) minimized competition to deeps soil water, which is excessive consumption by plant and rainwater is the only replenish source to deep soil water. The detailed discussion of the first adaptation types can be observed in "4.2 RRS uptake enhances plant transpiration

for coexisting species in mixed plantation" subsection in "**4 Discussion**" section as follows: "Generally, two types of adaptation can be adopted by plants to cope with resource competition: increased competition ability or minimized competition interactions (West et al., 2007). Consistent with the first adaptation type, mixed afforestation enhanced the RUP for *H. rhamnoides* and *P. tomentosa* (Figs. 3 and 7, Table S4)." (Page 21 Lines 479-482). The detailed discussion of the second adaptation types can be observed in same subsection as follows: "Furthermore, consistent with the second adaptation type, mixed afforestation significantly decreased the water uptake proportion from the deep soil layer for these species (Table S5)." (Page 21 Lines 495-496).

Thus, the plant water sources from different soil layers based on MixSIR method is the complement analysis for RUP calculation in the present study. These two methods together were used to discuss the two types of plant adaptation to cope with resource competition.

**Thirdly**, the detailed method of how to calculated water uptake proportions from different soil layers through MixSIR method has been rewritten in "2.6.2 Calculation of RRS uptake proportion and water sources from different soil layers" subsection in "**2 Materials and methods**" section as follows: "Firstly, the 7 soil depths (0–10, 10–20, 20–30, 30–50, 50–100, 100–150, and 150–200 cm) were combined into three soil layers (shallow, middle, and deep) based on the variation of soil water $\delta^{18}O$ and $\delta D$ and SW, to facilitate water source comparisons. The shallow soil layer (0–30 cm) was vulnerable to rainfall, and exhibited higher soil water $\delta^{18}O$ and $\delta D$ values and larger water isotope and SW variations (Table S3, Fig. S3). The middle soil layer (30–100 cm) was less vulnerable to rainfall, with high soil water isotope values and large water isotope and SW variations. The deep soil layer (100–200 cm) was relative stable, with low soil water isotope values and small water isotope and SW variation compared with shallow and middle soil layers. In addition, based on independent-sample *t*-test, no significant difference (P > 0.05) in soil water $\delta^{18}O$ and $\delta D$ between different soil depths in the same soil layer in each plot ensured the feasibility of the combination of the three soil layers (Phillips et al., 2005). Then, the water uptake proportions from three soil layers were calculated using the MixSIR program (Moore and Semmens, 2008), with model input parameters being the average $\delta^{18}O$

and δD values in plant stem water and soil water at each soil layer in each plot." (Pages 11 Lines 271-283)

In the present study, firstly, as suggested in Phillips et al. (2005), differences in soil water $\delta^{18}O$ and δD at 7 depths were determined by independent-sample $t$-test, to ensure the feasibility of combining the three potential soil layers (0–30, 30–100, and 100–200 cm). Then, the plant water sources from these 3 soil layers were quantified using MixSIR method.

**Table S3** The average (mean ± SD) and coefficients of variation (CVs, SD/mean) of soil water $\delta^{18}O$ and δD on the first day after 5 selected rainfall events, and daily soil water content (SW) from DOY 152 to 273 (1 June to 30 September) in *H. rhamnoides* pure plantation, *P. tomentosa* pure plantation, and *H. rhamnoides–P. tomentosa* mixed plantation.

| | Soil depth | soil water $\delta^{18}O$ (‰) | | soil water δD (‰) | | SW ($m^3\ m^{-3}$) | |
| --- | --- | --- | --- | --- | --- | --- | --- |
| | | average | CV | average | CV | average | CV |
| *H. rhamnoides* pure plantation | 0–30 cm | -5.61±1.57 | 27.99 | -41.53±11.68 | 28.12 | 0.13±0.025 | 19.23 |
| | 30–100 cm | -7.14±0.92 | 12.89 | -52.37±6.47 | 12.35 | 0.1±0.012 | 12 |
| | 100–200 cm | -9.3±0.69 | 7.42 | -68.66±3.53 | 5.14 | 0.09±0.006 | 6.67 |
| *P. tomentosa* pure plantation | 0–30 cm | -5.43±1.69 | 31.12 | -42.08±11.91 | 28.3 | 0.13±0.026 | 20 |
| | 30–100 cm | -7.49±0.73 | 9.75 | -51.34±4.56 | 8.88 | 0.09±0.008 | 8.89 |
| | 100–200 cm | -9.39±0.34 | 3.62 | -67.36±3.79 | 5.63 | 0.08±0.005 | 6.25 |
| Mixed plantation | 0–30 cm | -5.68±1.73 | 30.46 | -41.67±10.67 | 25.61 | 0.12±0.021 | 17.5 |
| | 30–100 cm | -6.57±1.08 | 16.44 | -47.8±5.78 | 12.09 | 0.1±0.011 | 11 |
| | 100–200 cm | -9.07±0.5 | 5.51 | -64.47±2.45 | 3.8 | 0.09±0.005 | 5.56 |

There are 45, 30, and 30 data for calculated the average water $\delta^{18}O$ and δD of shallow, middle, and deep soil layer in each plantation, respectively. The absolute value was used for CVs of soil water $\delta^{18}O$ and δD calculation.

To be noticed, although the $\delta^{18}O$ and δD values of soil water was more positive at middle (30-100 cm) than in shallow (0-30 cm) soil layer after 15.4 mm as in red cycle in Figure S7 as follows, the averaged $\delta^{18}O$ and δD after 5 rainfall events were more negative in middle than in shallow soil layer during the

study period (Table S3).

[Figure]

**Figure S7.** Variation in average (mean ± SD) $\delta^{18}$O and δD of rainwater, stem water, and soil water at seven soil depths for *H. rhamnoides* in (a–e) pure and (k–o) mixed plantations and for *P. tomentosa* in (f–j) pure and (k–o) mixed plantations after 5 rainfall events. Error bars indicate the standard deviation (n = 3). The date of each 5 selected rainfall events is followed the corresponding rainfall amount value. The average rainwater $\delta^{18}$O and δD for each rainfall event is calculated with 3 rainwater subsamples, which was divided from one rainwater sample.

**References:**

Cheng, X. L., An, S. Q., Li, B., Chen, J. Q., Lin, G. H., Liu, Y. H., Luo, Y. Q., and Liu, S. R.: Summer rain pulse size and rainwater uptake by three dominant desert plants in a desertified grassland ecosystem in northwestern China, Plant Ecol, 184, 1-12, 2006.

Gebauer, R. L. E., and Ehleringer, J. R.: Water and nitrogen uptake patterns following moisture pulses in a cold desert community, Ecology, 81, 1415-1424, 2000.

Phillips, D. L., Newsome, S. D., and Gregg, J. W.: Combining sources in stable isotope mixing models: alternative methods, Oecologia, 144, 520-527, 2005.

*4. Where are the plantations where you conducted the measurements located? It would be good if you could provide a map to illustrate this. Also, what is the slope? Are there any terraces or other soil and water conservation measures? Are soil properties and land-use history similar across all nine plots included in the study? Do you have any information on the physical characteristics of the soils?*

**Response: Added.** Thanks for these meaningful plot basic information suggestions, the detailed information have been added in the revised manuscript.

**Firstly**, for plots location, the plantation sites on the Loess Plateau of China has been added in "2.1 Study site" subsection in "**2 Materials and methods**" as follows: "The study was conducted in the Ansai Ecological Station in the semiarid Loess Plateau (36.55 °N, 109.16 °E), Northern China (Fig. S1)." (Page 5 Lines 110-111)

[Figure]

**Figure S1.** The geographic location of (a) study area and (b) plantation site in the Loess Plateau of China, and (c) monthly average (mean ±SD) rainfall amount and air temperature (Ta) during 2000-2017, and monthly rainfall amount and average Ta in 2018**.** Plantation types including *H. rhamnoides* pure plantation, *P. tomentosa* pure plantation, and Mixed plantation. Three adjacent plots were selected (16 m ×10 m) for each plantation type, and the schematic diagram of these plantation types is in (b). The

China basic map can be obtained from

http://map.geoq.cn/arcgis/rest/services/ChinaOnlineCommunityENG/MapServer.

In addition, the (b) in Fig S1 is the clearest picture that we can get, and the size of each three adjacent plots (16 m $\times$ 10 m) is the schematic diagram. Indeed, the plantation areas of these types are larger than the plot in Fig S1. The relative areas can be observed as follows:

[Figure]

**Secondly**, for plantation slope, land-use history, and soil and water conservation measures, the relative sentences have been added in "2.1 Study site" subsection in "**2 Materials and methods**" as follows: "Three adjacent plantations were chosen for the study: pure *H. rhamnoides* plantation, pure *P. tomentosa* plantation, and *H. rhamnoides–P. tomentosa* mixed plantation (Fig. S1), with corresponding plantation slope of 5.2, 4.5, and 5.5°. All plantations were planted on abandoned grassland in 2004, where *Bothriochloa ischaemum* was the dominant herbaceous species at that time. Three adjacent plots were selected (16 m $\times$ 10 m) for each plantation type, and no soil and water conservation measure was conducted in the plantations." (Page 5 Lines 118-123)

**Thirdly**, for soil physical properties, the relative sentences have been added in "2.1 Study site" subsection in "**2 Materials and methods**" section as follows: "The soil is characterized as a silt loam

soil according to United States Department of Agriculture soil taxonomy, with 24.2% sand (2–0.05 mm), 62.5% silt (0.05–0.002 mm), and 13.3% clay (<0.002 mm) determined by Mastersize 2000 (Malvern Instruments Ltd., UK)." (Page 5 Lines 114-117).

"Based on an experiment conducted in July 2017 through cutting ring method, the soil bulk density, filtration property, total porosity, and capillary porosity at 0–50 cm soil depth were similar in three plantations. The average soil bulk density was 1.34 ±0.04, 1.31 ±0.05, and 1.31 ±0.05 g cm$^{-3}$ for pure *H. rhamnoides*, pure *P. tomentosa*, and mixed plantations, respectively, and corresponding soil saturated hydraulic conductivity was 0.97 ±0.15, 0.96 ±0.13, and 0.99 ±0.11 mm min$^{-1}$. The average soil total porosity was 48.25 ±0.52, 48.17 ±0.48, and 48.03 ±0.63% for pure *H. rhamnoides*, pure *P. tomentosa*, and mixed plantations, respectively, and corresponding soil capillary porosity was 38.89 ±1.57, 39.02 ± 1.26, and 38.95 ±1.87%." (Pages 5-6 Lines 133-140).

The detailed criterion to determine the soil texture can be observed in **Figure explain 2** as follows:

[Figure]

**Figure explain 2** The soil texture criterion based on the percent of sand (2-0.05 mm), silt (0.05-0.002 mm), and clay (<0.002 mm) in United States Department of Agriculture soil taxonomy. The red arrow lines represent the percent of sand, silt, and clay of our studied soil.

*5. The authors selected 5 distinct rainfall events of varying magnitude (ranging from 3.4 mm to 35.2 mm) to study the response of the tree species (in both pure and mixed stands) and how this varies according to the magnitude of the event. As stated in L. 168-169, 'These rainfall events were selected with an interpulse period longer than 7 days to eliminate the potential influence of the previous rainfall event.'.*

*However, I have serious doubts about this approach and the validity of the results from this specific analysis (e.g., L. 478-482: 'The increasing rainfall amount significantly decreased water source proportion from deep soil layer (P<0.05) for H. 480 rhamnoides and P. tomentosa in the mixed plantation (Table S3), with the corresponding values decreasing from 43.13 ±13.74% and 47.07 ±5.39% (both after 3.4 mm), respectively, to 21.54 ±8.9% (after 35.2 mm) and 28.66 ±12.26% (after 24 mm) (Fig 4)'). Unfortunately, the selected rainfall events not only differ in magnitude, but also in terms of antecedent conditions. For example, the 3.5 mm event (DOY 194) is the lowest rainfall event but also that following the most prolonged dry period (>30 days dry period from DOY 157 to 194). It is evident that when topsoil moisture content is low following a dry period, plants will tap into deeper, more reliable water sources. This is not so much related to a single rainfall event and its magnitude, but mostly to the antecedent conditions (prolonged dry period).*

**Response: Deleted and Rewritten.** In response to this meaningful and valuable question, these sentences in previous version has been deleted, and relative sentences have been rewritten in the revised manuscript in "4.2 RRS uptake enhances plant transpiration for coexisting species in mixed plantation" subsection in "**4 Discussion**" section as follows: "Furthermore, consistent with the second adaptation type, mixed afforestation significantly decreased the water uptake proportion from the deep soil layer for these species (Table S5). Similar to other studies in the Loess Plateau (Wang et al., 2020; Wu et al., 2021), the deep soil layer exhibited lower SW than other soil layers in all plantation types in the present study (Fig. 1, Table S3). Jia et al. (2017) and Wang et al. (2020) attributed the lower SW in deep soil layers to the imbalance between rainwater replenishment and plant uptake of water from this layer. In addition, plants may expend more energy to uptake water from deep compared to shallow soil layers (Schenk, 2008), especially when the deep soil layer exhibits lower SW. Thus, both increased rainwater-recharged soil water uptake and decreased water source competition from the deep soil layer were adopted by these species in the mixed plantation to minimize water sources competition under water limited conditions." (Page 22 Lines 495-504)

Indeed, the water sources from deep soil layer for these plant species may mainly influenced by antecedent soil water conditions. Thus, we rewrote these sentences mainly to illustrate the second

adaptation type of these plant species, that is "minimized competition interactions"—mixed afforestation significantly decreased the water uptake proportion from the deep soil layer. The lower SW in deep soil layer than in other soil layers is the similar soil water conditions to other studies in the Loess Plateau, and Jia et al. (2017) and Wang et al. (2020) attribute the lower SW in deep soil layer to the imbalance between rainwater replenishment and plant uptake water from this soil layer. In addition, more energy should be expend by plant for water uptake from deep than from shallow soil layer (Schenk, 2008). These sentences illustrated the second adaptation type "minimized competition interactions".

**References:**

Jia, X. X., Shao, M. A., Zhu, Y. J., and Luo, Y.: Soil moisture decline due to afforestation across the Loess Plateau, China, J Hydrol, 546, 113-122, 2017.

Wang, J., Fu, B. J., Wang, L. X., Lu, N., and Li, J. Y.: Water use characteristics of the common tree species in different plantation types in the Loess Plateau of China, Agr Forest Meteorol, 288, ARTN 108020, 10.1016/j.agrformet.2020.108020, 2020.

Wu, W. J., Li, H. J., Feng, H., Si, B. C., Chen, G. J., Meng, T. F., Li, Y., and Siddique, K. H. M.: Precipitation dominates the transpiration of both the economic forest (Malus pumila) and ecological forest (Robinia pseudoacacia) on the Loess Plateau after about 15 years of water depletion in deep soil, Agr Forest Meteorol, 297, ARTN 108244, 10.1016/j.agrformet.2020.108244, 2021.

Schenk, H. J.: Soil depth, plant rooting strategies and species' niches, New Phytol, 178, 223-225, 2008.

*6. I would strongly recommend that the authors include a plot of the local meteoric water line (LMWL). This should be relatively straightforward as they have analyzed the 19 collected rainfall samples for both δ18O and δH. On top of the LMWL I would then plot the signatures of the soil water at different depths. This would provide additional insights into the data and ease data interpretation (and can also be used to double-check that rainfall samples have not undergone evaporation). For instance, the rainfall signatures in Figure S5 could be visualized and interpreted much better in a dual-isotope plot.*

**Response: Added.** In response to this meaningful and useful suggestion, the Figure S3 including the local meteoric water line (LMWL) and the regression between $\delta^{18}O$ and $\delta D$ at three different soil layers (0-30, 30-100, and 100-200cm) has been added in the revised manuscript. Meanwhile, the relative sentences have also been added in "2.6 Statistical analysis" subsection in "**2 Materials and methods**"

section as follows: "The shallow soil layer (0–30 cm) was vulnerable to rainfall, and exhibited higher soil water $\delta^{18}O$ and $\delta D$ values and larger water isotope and SW variations (Table S3, Fig. S3). The middle soil layer (30–100 cm) was less vulnerable to rainfall, with high soil water isotope values and large water isotope and SW variations. The deep soil layer (100–200 cm) was relative stable, with low soil water isotope values and small water isotope and SW variation compared with shallow and middle soil layers." (Page 11 Lines 273-278).

[Figure]

**Figure S3.** The linear regression relationship between $\delta^2H$ and $\delta^{18}O$ for soil water at three layers (0–30, 30–100, and 100–200cm) in (a) *H. rhamnoides* pure plantation, (b) *P. tomentosa* pure plantation, and (c) Mixed plantation. The local meteoric water line (LMWL) is plotted in each panel for reference.

Indeed, the slope of LMWL is 7.29 based on 19 collected rainfall (**Figure explain 3**), which is slightly lower than the global meter water line (the slope is 8.0), but similar to the LMWL (the slope ranges from 7.47 to 7.76) results near the present study region in the semiarid Loess Plateau (Figure **explains 4 and 5**). The slopes between $\delta D$ and $\delta^{18}O$ for three soil layers were smaller than LMWL and

slightly decreased with soil layer increased. The slightly lower slope for each of 3 soil layer compared with LMWL, may mainly attribute to the mixture of soil water before rainfall event and rainwater, and the soil water before rainfall generally undergo evaporation. The lower soil water δD and $δ^{18}$O slope compared with LMWL can be observed previous studies such as in **Figures explain 4 and 5** in the semiarid Loess Plateau as follows**.**

[Figure]

**Figure explain 3** Linear regression between $δ^2$H and $δ^{18}$O in rainwater. There are 19 collected rainwater samples and 5 selected rainwater samples in the present study. The rainfall amount of each of 5 selected rainwater samples was added.

[Figure]

**Figure explain 4** Linear regression between $δ^2$H and $δ^{18}$O in soil water for an (a) apple orchard and (b) black locust forest. LMWL represents the local meteoric water line (y = 7.47x + 3.29, R2 = 0.95, P < 0.01). The isotopic composition values of xylem water from both species are also shown. This plot is from Wu et al. (2016).

[Figure]

**Figure explain 5** The linear regression relationship between $\delta^2H$ and $\delta^{18}O$ in soil water from three species (a) S. bungeana, (b) A. gmelinii, (c) V. negundo during the sampling periods. SWL represents soil water line based on isotopic data of soil water. LMWL represents the local meteoric water line (y = 7.76x + 5.14, $R^2$ = 0.91, p < 0.01). This plot is from Wang et al. (2017).

[Figure]

**Figure S1.** The geographic location of (a) study area and (b) plantation site in the Loess Plateau of China, and (c) monthly average (mean ±SD) rainfall amount and air temperature (Ta) during 2000-2017, and monthly rainfall amount and average Ta in 2018. Plantation types including *H. rhamnoides* pure plantation, *P. tomentosa* pure plantation, and Mixed plantation. Three adjacent plots were selected (16 m ×10 m) for each plantation type, and the schematic diagram of these plantation types is in (b). The China basic map can be obtained from http://map.geoq.cn/arcgis/rest/services/ChinaOnlineCommunityENG/MapServer.

*2. When Describing the soil texture (L.105) please indicate it is the texture you refer to and add the correct source (USDA). Besides the soil texture, kindly provide the soil class.*

**Response: Corrected and Added.** Thanks for this meaning suggestion, the texture, correct source (USDA), and the soil class has been written in "2.1 Study site" subsection in "**2 Materials and methods**" section as follows: "The soil is characterized as a silt loam soil according to United States Department of Agriculture soil taxonomy, with 24.2% sand (2–0.05 mm), 62.5% silt (0.05–0.002 mm), and 13.3% clay (<0.002 mm) determined by Mastersize 2000 (Malvern Instruments Ltd., UK)." (Page 5

Lines 114-117).

Because the "United States Department of Agriculture" occurred only once in the present study, thus the abbreviation named "USDA" of it was not used. The detailed criterion to determine the soil texture can be observed in **Figure explain 2** as follows:

[Figure]

**Figure explain 2** The soil texture criterion based on the percent of sand (2-0.05 mm), silt (0.05-0.002 mm), and clay (<0.002 mm). The red arrow lines represent the percent of sand, silt, and clay of our studied soil.

*3. L.131: Explain what VPD stands for after equation 1 and give its units (as you have done for the other variable sin equation 1).*

**Response: Added.** Thanks for this suggestion, the VPD explanation has been added in the revised manuscript in "2.2 Environmental parameter measurements and $ET_0$ calculation" subsection in "**2 Materials and methods**" section as follows: "where $\gamma$, $s$, and *VPD* are the psychrometric constant (kPa $K^{-1}$), the slope between saturation vapor pressure and air temperature (kPa $K^{-1}$), and vapor pressure deficit (kPa), respectively." (Page 6 Lines 155-156)

*4. L.145: What does the abbreviation TDPs mean?*

**Response: Added.** In response to this meaningful suggestion, the TDPs has been added at its first appearance sentence in "2.3 Sap flow observation" subsection in "**2 Materials and methods**" section

as follows: "The sap flow was monitored by a pair of Granier-type thermal dissipation probes (TDPs) 10 mm in length and 2 mm in diameter in 36 selected individuals." (Page 7 Lines 163-165).

*5. L.189, Formula 4: what is PAP?*

**Response: Corrected.** Thanks for this suggestion. It should be RUP but not PAP, and this equation has been revised in "2.6.2 Calculation of RRS uptake proportion and water sources from different soil layers" subsection in "**2 Materials and methods**" section as follows:

   "The RRS uptake proportion (RUP, %) after a recent rainfall pulse for plant was calculated as the proportion of rainwater in plant stem as follows (Cheng et al., 2006):

$$\delta^{18}O\ (D)_P = RUP \times \delta^{18}O\ (D)_{rain} + (1 - RUP) \times \delta^{18}O\ (D)_{swb} \tag{5}$$

$$RUP = (\delta^{18}O(D)_p - \delta^{18}O(D)_{swb})/(\delta^{18}O(D)_{swa} - \delta^{18}O(D)_{swb}) \times 100\% \tag{6}$$

   " (Page 10 Lines 247-250)

*6. L.208-209: Kindly provide a reference that supports this assumption ('no fractionation was considered during water source uptake by plant roots')*

**Response: Rewritten.** Thanks for this meaningful suggestion, this sentence has been rewritten in "**2 Materials and methods**" section as follows: "No fractionation was considered during water source uptake by these plant roots because none of the plants exhibited xerophytic or halophytic characteristics. Ellsworth and Williams (2007) and Moore and Semmens (2008) suggested that a water stable isotope fractionation generally occurred during root uptake by xerophytic or halophytic plants." (Page 11 Lines 284-288).

[Figure]

**Figure 1.** Variation in (a) rainfall amount, reference evapotranspiration ($ET_0$), and average (mean $\pm$SD) soil water content (SW) in (b) *H. rhamnoides* pure plantation, (c) *P. tomentosa* pure plantation, and (d) mixed plantation from DOY 152 to 273 (1 June to 30 September) (n = 3). Standard deviation bars for SW at each soil layers are not shown to allow clear display of variation of SW for each plantation. Arrows in (a) indicate dates of sample collection at the first day after rainfall events: DOY 157 (6 June), DOY 194 (12 July), DOY 204 (23 July), DOY 249 (6 September), and DOY 265 (22 September).

In addition, the first selected rainfall amount is 35.2 mm, occurred in DOY 155-156, can be observed in the red box in **Figure 1** as follows. The reason that we choose the rainfall amount during DOY 155-156, is mainly because the 35.2mm was continues occurred during these two successive days. DOY 155 only received 5.6 mm rainfall from 23:00-24:00, and DOY156 received 29.6mm. Therefore, the rainfall amount (5.6mm+29.6mm=35.2mm) received during DOY 155-156 was considered as one rainfall event in our present study. The half-hourly rainfall distribution during DOY 155-156 can be observed in **Figure explain 6** as follows:

[Figure]

**Figure 1.** Variation of (a) rainfall amount, reference evapotranspiration (ET$_0$), and averaged soil water content (SW) in (b) *H. rhamnoides* pure plantation, (c) *P. tomentosa* pure plantation, and (d) mixed plantation from DOY 152 to 273 (1 June to 30 September). Standard deviation bars for SW at each soil layers are not shown to allow clear display of variation of SW for each plot. Arrows in (a) indicate dates of sample collection: DOY 157–159 (6–8 June), DOY 194–196 (12 –14 July), DOY 204–206 (23–25 July), DOY 249–251 (6–8 September), and DOY 265–267 (22–24 September).

[Figure]

**Figure explain 6** The half-hour rainfall amount during DOY155-156. The rainfall during DOY 155-156 was considered as one rainfall event in the present study.

*8. Figure 2: It would be good to show the precipitation bars in this plot too.*

**Response: Added.** Both the precipitation amount and arrows indicate dates of sample collection have been added in the revised **Figure 2 (a)** in the revised manuscript as follows (Page 14 Lines 349-352):

[Figure]

**Figure 2.** Variation in (a) rainfall amount, and average daily normalized F_d for *H. rhamnoides* in (a) pure and (b) mixed plantations and for *P. tomentosa* in (c) pure and (d) mixed plantations (n = 3). Arrows in (a) indicate dates of sample collection at the first day after rainfall events: DOY 157 (6 June), DOY 194 (12 July), DOY 204 (23 July), DOY 249 (6 September), and DOY 265 (22 September).

*9. L.321: in figure S5, kindly add the date of rainfall events. Moreover, the dD signature of rainwater for the 3.4 and 7.9 mm events is very enriched. Could it be that there has been some evaporation of the sample going on? In any case, as I mentioned earlier, it would be really good if the authors could provide a plot of the LMWL.*

**Response: Added and Clarified.**

**Firstly**, the date of 5 selected rainfall events have been added in Figure S7 in the revised manuscript as follows:

[Figure]

**Figure S7.** Variation in average (mean ± SD) $\delta^{18}O$ and $\delta D$ of rainwater, stem water, and soil water at seven soil depths for *H. rhamnoides* in (a–e) pure and (k–o) mixed plantations and for *P. tomentosa* in (f–j) pure and (k–o) mixed plantations after 5 rainfall events. Error bars indicate the standard deviation (n = 3). The date of each 5 selected rainfall events is followed the corresponding rainfall amount value. The average rainwater $\delta^{18}O$ and $\delta D$ for each rainfall event is calculated with 3 rainwater subsamples, which was divided from one rainwater sample.

**Secondly**, a plot including the local meteoric water line (LMWL) and the regression between $\delta^{18}O$ and $\delta D$ at three different soil layers (0-30, 30-100, and 100-200cm) has been added in **Figure S3** as follows. Maybe there are two reasons that can illustrate the enriched $\delta^{18}O$ and $\delta D$ values for 3.4 and 7.9

mm.

*1)*, This may mainly attribute to the rainfall amount effect (Liu et al., 2014), that is the rainfall event with small rainfall amount generally had positive $\delta^{18}O$ and $\delta D$ values than those for large rainfall amount (**Figures explain 3 and 7**).

[Figure]

**Figure S3.** The linear regression relationship between $\delta^2H$ and $\delta^{18}O$ for soil water at three layers (0–30, 30–100, and 100–200cm) in (a) *H. rhamnoides* pure plantation, (b) *P. tomentosa* pure plantation, and (c) Mixed plantation. The local meteoric water line (LMWL) is plotted in each panel for reference.

[Figure]

**Figure explain 3** Linear regression between $\delta^2H$ and $\delta^{18}O$ in rainwater. There are 19 collected rainwater samples and 5 selected rainwater samples in the present study. The rainfall amount of each of 5 selected rainwater samples was added.

[Figure]

**Figure explain 7** Distribution of daily precipitation (PPT) and variation of $\delta^{18}O$ values for event rainwater and groundwater (GW) during the rainy/dry season cycle (2008–2009). The stippled bars at the top of the panel are phenophases for rubber trees, i.e. Fr representing fruit ripening, Ds dormant stage, Ls leaf shedding, Lf leaf flushing, Le leaf expansion, Fp flowering phase and Fs fruit setting. Vertical arrows indicate sampling dates. This plot is from Liu et al. (2014).

*2)*, Salamalikis et al. (2016) suggested that the positive $\delta^{18}O$ and $\delta D$ values for small rainfall amount may partially attribute to the sub-cloud evaporation effect in rainfall in dry conditions (**Figure explain 8**). In the present study, the 3.4 and 7.9 mm occurred with high $ET_0$ period, which reflects the dry condition (**red cycles in Fig 1 as follows**).

[Figure]

**Figure explain 8** Schematic of sub-cloud evaporation effect on rainwater. This plot is from Salamalikis et al. (2016).

[Figure]

**Figure 1.** Variation in (a) rainfall amount, reference evapotranspiration ($ET_0$), and average (mean $\pm$ SD) soil water content (SW) in (b) *H. rhamnoides* pure plantation, (c) *P. tomentosa* pure plantation, and (d) mixed plantation from DOY 152 to 273 (1 June to 30 September) (n = 3). Standard deviation bars for SW at each soil layers are not shown to allow clear display of variation of SW for each plantation. Arrows in (a) indicate dates of sample collection at the first day after rainfall events: DOY 157 (6 June), DOY 194 (12 July), DOY 204 (23 July), DOY 249 (6 September), and DOY 265 (22 September).

**DATE:** January 26, 2022

Compuscript Ltd
T/A International Science Editing
Bay K, Shannon Industrial Park West
Shannon, Co Clare
Ireland
Phone +353 61 472818   Fax +353 61 472688

To whom it may concern,

The paper "Differential response of plant transpiration to uptake of rainwater-recharged soil water for dominant tree species in the semiarid Loess Plateau" by Yakun Tang was edited by International Science Editing. We were asked not to edit the references. Please contact us if you would like to view the edited paper.

Kindest regards,

David Cushley.

If the English and our answers are not meet the standard, please give me another chance, I will revised the language and answered the relative questions again.

[revised manuscript text omitted]
 layer. In addition, plants may expend more energy to uptake water from deep compared to shallow soil layers (Schenk, 2008), especially when the deep soil layer exhibits lower SW. Thus, both increased rainwater-recharged soil water uptake and decreased water source competition from the deep soil layer were adopted by these species in the mixed plantation to minimize water sources competition under water limited conditions.

**4.3 Implications for plantation species and type selection based on RRS uptake and plant transpiration**

The RRS uptake and plant transpiration in response to rainfall pulses may influence plant physiological process and the water cycle (Meier et al., 2018; Zhao et al., 2021). In pure plantations, *H. rhamnoides* rather than *P. tomentosa* showed an advantage in RRS uptake due to the large $\Psi_{pd}-\Psi_m$ and high fine root surface area proportions distributed in the shallow soil layer for the former species, although both species exhibited plasticity in water sources. The excessive water uptake from the deep soil may desiccate deep soil (Wu et al., 2021), weakening plant resilience to drought stress and thus

plant community sustainability in this Loess Plateau region (Song et al., 2018; Zhao et al., 2021). West

et al. (2012) and Wu et al. (2021) suggested that increased RRS uptake can reduce plant water uptake

from deep soil layers, and is essential for plantation adaptation in water limited regions. In the present

study, physiological (e.g., $\Psi_m$) and morphological (fine root distribution) adjustments were observed for

*H. rhamnoides* and *P. tomentosa* in the mixed plantation, respectively, to enlarge $\Psi_{pd}-\Psi_m$ and enhance

the RUP and plant transpiration (Figs. 7 and S4). The significantly increased RUP and decreased deep

soil water uptake proportion for both species in mixed plantation may relieve deep soil water deficit and

strengthen plantation sustainability (Tables S4 and S5). Furthermore, mixed afforestation also increased

the total biomass of *H. rhamnoides* and *P. tomentosa*, calculated through the allometric equation

indicated in Zhou et al. (2018) and Tang et al. (2019) (Table S8). Thus, rainfall pulse sensitive species

in pure plantation, and plant species in mixed plantation that can adopt physiological or morphological

adjustment to enhance rainwater-recharged soil water uptake and reduce excessive water uptake from

deep soil layers, should be more often considered for use in the studied region. In addition, no runoff

was generated under 0.74 mm min$^{-1}$ rainfall intensity in silt loam soil in the Loess Plateau (Huang et al.,

2014), which had no vegetation cover and similar soil saturated hydraulic conductivity (0.99 $\pm$0.15 g

cm$^{-3}$) to that in the present study. Pan and Shuangguan (2005) also observed no runoff generation under

1.5 mm min$^{-1}$ rainfall intensity for vegetation covered plots with 15 $^\circ$slope in the Loess Plateau. Direct

observation for possible runoff after large rainfall events in further studies would be helpful for

evaluating plantation species adaptability in the studied region, although Zhao et al. (2013) showed that

the vegetation cover can enhance soil permeability and reduce water loss in the Loess Plateau.

Furthermore, water conservation measures, such as water-fertilizer pits (60 $\times$60 $\times$40 cm) (Wang et al.,

2020), that can intercept any possible runoff after large rainfall events and deliver it to deep soil layers

may be appropriate for the studied region.

**5 Conclusions**

The influence of water sources and $\Psi_{pd}-\Psi_m$ on plant transpiration in response to rainfall pulses was

determined for *H. rhamnoides* and *P. tomentosa* in the semiarid Loess Plateau region. In pure

plantations, the $SF_R$ was significantly influenced by RUP and $\Psi_{pd}-\Psi_m$ for *H. rhamnoides*, but the $SF_R$ was significantly influenced by $\Psi_{pd}-\Psi_m$ for *P. tomentosa*. Meanwhile, the lower value $\Psi_{pd}-\Psi_m$ was consistent with the high $SF_R$ for *H. rhamnoides*, and the higher value $\Psi_{pd}-\Psi_m$ was consistent with the low $SF_R$ for *P. tomentosa*, in response to rainfall pulses. Thus, *H. rhamnoides* and *P. tomentosa* exhibited sensitive and insensitive response to rainfall pulses, respectively. Furthermore, mixed afforestation enhanced the RRS uptake and plant transpiration for both species. Significantly lower plant $\Psi_m$ and increased fine root surface area were adopted by *H. rhamnoides* and *P. tomentosa* in the mixed plantation, respectively, to enlarge $\Psi_{pd}-\Psi_m$ and enhance RRS uptake and decrease water source competition from the deep soil layer. The $SF_R$ was significantly influenced by RUP and $\Psi_{pd}-\Psi_m$ for both species in the mixed plantation, and RRS uptake enhanced plant transpiration in the mixed plantation regardless of species sensitivity to rainfall pulses.

**Data availability**

The data that support the findings of this study are available from the corresponding author upon request.

**Author contribution**

YKT designed the study, performed the statistical analyses and wrote the original manuscript draft. LNW and YQY performed the experiments and collected the data. DXL collected the data.

**Declaration of Competing Interest**

The authors declare that they have no conflict of interest.

**Acknowledgement**

This work was supported by the National Natural Science Foundation of China (41977425), the National Key Research and Development Program of China (2017-YFA0604801).

[revised manuscript text omitted]

**Tables and captions**

**Table S1.** Plant height, trunk diameter, and estimated sapwood width for *H. rhamnoides* and *P. tomentosa* in both pure and mixed plantations.

| Plantation type | No. | Height (m) | Trunk diameter (mm) | Sapwood width (mm) |
|---|---|---|---|---|
| *H. rhamnoides* in pure plantation | 1 | 3.95 | 45 | 9 |
| | 2 | 4.26 | 53 | 11 |
| | 3 | 4.05 | 51 | 10 |
| | 4 | 4.13 | 49 | 9 |
| | 5 | 3.98 | 50 | 10 |
| | 6 | 4.1 | 51 | 11 |
| | 7 | 4.3 | 57 | 12 |
| | 8 | 3.86 | 44 | 9 |
| | 9 | 3.92 | 53 | 11 |
| *P. tomentosa* in pure plantation | 1 | 4.41 | 58 | 17 |
| | 2 | 3.9 | 52 | 9 |
| | 3 | 3.92 | 56 | 16 |
| | 4 | 4.35 | 56 | 17 |
| | 5 | 4.59 | 58 | 16 |
| | 6 | 4.2 | 53 | 13 |
| | 7 | 4.29 | 54 | 15 |
| | 8 | 3.86 | 51 | 9 |
| | 9 | 3.98 | 52 | 11 |
| *H. rhamnoides* in mixed plantation | 1 | 4.36 | 52 | 12 |
| | 2 | 3.9 | 49 | 11 |
| | 3 | 4.23 | 51 | 12 |
| | 4 | 4.5 | 56 | 13 |
| | 5 | 4.73 | 55 | 14 |
| | 6 | 3.96 | 49 | 11 |
| | 7 | 4 | 51 | 12 |
| | 8 | 4.52 | 53 | 12 |
| | 9 | 4.39 | 52 | 12 |
| *P. tomentosa* in mixed plantation | 1 | 4.12 | 53 | 11 |
| | 2 | 3.75 | 46 | 9 |
| | 3 | 4.5 | 57 | 13 |
| | 4 | 4.21 | 53 | 11 |
| | 5 | 4.2 | 53 | 11 |
| | 6 | 4.16 | 51 | 10 |
| | 7 | 3.8 | 45 | 9 |
| | 8 | 4.95 | 59 | 13 |
| | 9 | 4.16 | 51 | 10 |

The sapwood width was estimated through the equation established through 12 unmonitored individual core samples for specific species with different diameters. The core sample was obtained using an increment borer, and the colour difference between sapwood and heartwood was large. The equation between trunk diameter (mm) and sapwood width (mm) was y=0.248x-2.296 $R^2$=0.84 p<0.01 for *H. rhamnoides* in pure plantation; y=0.348x-5.98 $R^2$=0.78 P<0.01 for *H. rhamnoides* in mixed plantation; y=1.126x-47.66 $R^2$=0.83 P<0.01 for *P. tomentosa* in pure plantation; y=0.317x-5.71 $R^2$=0.939 P<0.01 for *P. tomentosa* in mixed plantation.

**Table S2.** Independent-sample *t*-test parameters for predawn ($\Psi_{pd}$), midday ($\Psi_m$), and gradient of leaf water potential ($\Psi_{pd} - \Psi_m$) between the first and second day after each rainfall amount.

| | Rainfall amount (mm) | df | $\Psi_{pd}$ | | $\Psi_m$ | | $\Psi_{pd} - \Psi_m$ | |
|---|---|---|---|---|---|---|---|---|
| | | | *t* | *p* | *t* | *p* | *t* | *p* |
| *H. rhamnoides* in pure plantation | 3.4 | 4 | 0.18 | 0.87 | 1.21 | 0.29 | -2.5 | 0.07 |
| | 7.9 | 4 | 0.33 | 0.75 | 0.79 | 0.58 | -8.01 | 0.47 |
| | 15.4 | 4 | 0.85 | 0.44 | 0.27 | 0.8 | 0.21 | 0.85 |
| | 24 | 4 | 0.97 | 0.39 | -0.67 | 0.54 | 2.13 | 0.1 |
| | 35.2 | 4 | -0.09 | 0.93 | -7.1 | 0.52 | 0.28 | 0.79 |
| *P. tomentosa* in pure plantation | 3.4 | 4 | 0.88 | 0.43 | 0.66 | 0.55 | 0.81 | 0.47 |
| | 7.9 | 4 | 0.34 | 0.08 | 0.75 | 0.49 | -1.8 | 0.14 |
| | 15.4 | 4 | 0.23 | 0.83 | 0.73 | 0.51 | -0.82 | 0.46 |
| | 24 | 4 | -2.08 | 0.11 | 1.14 | 0.32 | -0.85 | 0.45 |
| | 35.2 | 4 | -1.67 | 0.17 | 1.15 | 0.31 | -2.22 | 0.09 |
| *H. rhamnoides* in mixed plantation | 3.4 | 4 | 2.53 | 0.07 | 1.4 | 0.24 | -0.6 | 0.58 |
| | 7.9 | 4 | 1.24 | 0.28 | 2.02 | 0.11 | -1.87 | 0.14 |
| | 15.4 | 4 | -0.9 | 0.42 | 0.96 | 0.39 | -1.29 | 0.27 |
| | 24 | 4 | 1.74 | 0.16 | 2.04 | 0.11 | -1.22 | 0.29 |
| | 35.2 | 4 | 1.89 | 0.13 | 2.57 | 0.06 | -0.29 | 0.78 |
| *P. tomentosa* in mixed plantation | 3.4 | 4 | 0.07 | 0.95 | 1.9 | 0.13 | -0.35 | 0.72 |
| | 7.9 | 4 | 0.81 | 0.46 | 0.96 | 0.39 | -0.46 | 0.67 |
| | 15.4 | 4 | 0.7 | 0.52 | 2.12 | 0.1 | -0.53 | 0.62 |
| | 24 | 4 | 1.85 | 0.14 | 0.74 | 0.49 | 0.48 | 0.66 |
| | 35.2 | 4 | 2.23 | 0.09 | 1.21 | 0.3 | 0.55 | 0.61 |

**Table S3** The average (mean ± SD) and coefficients of variation (CVs, SD/mean) of soil water $\delta^{18}O$ and $\delta D$ on the first day after 5 selected rainfall events, and daily soil water content (SW) from DOY 152 to 273 (1 June to 30 September) in *H. rhamnoides* pure plantation, *P. tomentosa* pure plantation, and *H. rhamnoides–P. tomentosa* mixed plantation.

| | Soil depth | soil water $\delta^{18}O$ (‰) | | soil water $\delta D$ (‰) | | SW (m$^3$ m$^{-3}$) | |
|---|---|---|---|---|---|---|---|
| | | average | CV | average | CV | average | CV |
| *H. rhamnoides* pure plantation | 0–30 cm | -5.61±1.57 | 27.99 | -41.53±11.68 | 28.12 | 0.13±0.025 | 19.23 |
| | 30–100 cm | -7.14±0.92 | 12.89 | -52.37±6.47 | 12.35 | 0.1±0.012 | 12 |
| | 100–200 cm | -9.3±0.69 | 7.42 | -68.66±3.53 | 5.14 | 0.09±0.006 | 6.67 |
| *P. tomentosa* pure plantation | 0–30 cm | -5.43±1.69 | 31.12 | -42.08±11.91 | 28.3 | 0.13±0.026 | 20 |
| | 30–100 cm | -7.49±0.73 | 9.75 | -51.34±4.56 | 8.88 | 0.09±0.008 | 8.89 |
| | 100–200 cm | -9.39±0.34 | 3.62 | -67.36±3.79 | 5.63 | 0.08±0.005 | 6.25 |
| Mixed plantation | 0–30 cm | -5.68±1.73 | 30.46 | -41.67±10.67 | 25.61 | 0.12±0.021 | 17.5 |
| | 30–100 cm | -6.57±1.08 | 16.44 | -47.8±5.78 | 12.09 | 0.1±0.011 | 11 |
| | 100–200 cm | -9.07±0.5 | 5.51 | -64.47±2.45 | 3.8 | 0.09±0.005 | 5.56 |

There are 45, 30, and 30 data for calculated the average water $\delta^{18}O$ and $\delta D$ of shallow, middle, and deep soil layer in each plantation, respectively. The absolute value was used for CVs of soil water $\delta^{18}O$ and $\delta D$ calculation.

**Table S4.** Repeated ANOVA (ANOVAR) parameters for the relative response of normalized sap flow ($SF_R$) and rainwater-recharged soil water uptake proportion (RUP) after rainfall pulses of *H. rhamnoides* and *P. tomentosa* (n = 30).

| | Variation source | *df* | $SF_R$ | | RUP | |
|---|---|---|---|---|---|---|
| | | | *F* | *p* | *F* | *p* |
| Pure plantation | Rainfall | 4 | 97.91 | <0.001 | 385.02 | <0.01 |
| | Species | 1 | 121.13 | <0.001 | 21.02 | <0.05 |
| | Rainfall × Species | 4 | 27.35 | <0.001 | 0.83 | 0.52 |
| Mixed plantation | Rainfall | 4 | 489.9 | <0.001 | 17696.38 | <0.01 |
| | Species | 1 | 70.38 | <0.001 | 4089.12 | <0.01 |
| | Rainfall × Species | 4 | 249.17 | <0.001 | 1776.62 | <0.01 |
| *H. rhamnoides* | Rainfall | 4 | 42.63 | <0.001 | 496.72 | <0.01 |
| | Plantation type | 1 | 337.09 | <0.001 | 360.16 | <0.01 |
| | Rainfall × Plantation type | 4 | 215.43 | <0.001 | 17.62 | <0.01 |
| *P. tomentosa* | Rainfall | 4 | 10.05 | <0.001 | 1969.3 | <0.01 |
| | Plantation type | 1 | 32.36 | <0.01 | 54.83 | <0.01 |
| | Rainfall × Plantation type | 4 | 19.12 | <0.001 | 208.06 | <0.01 |

*df* = degree of freedom, Plantation type = pure and mixed plantation for each species. Pure and Mixed plantation indicate the result of $SF_R$ and RUP for both species in different plantation types, respectively; *H. rhamnoides* and *P. tomentosa* indicate the mixed afforestation effect on $SF_R$ and RUP for these species.

**Table S5.** Repeated ANOVA (ANOVAR) parameters for water uptake proportion from shallow (0–30 cm), middle (30–100 cm), and deep (100–200 cm) soil layer for *H. rhamnoides* and *P. tomentosa* (n = 30).

| | Variation source | $df$ | 0–30cm | | 30–100cm | | 100–200cm | |
|---|---|---|---|---|---|---|---|---|
| | | | $F$ | $p$ | $F$ | $p$ | $F$ | $p$ |
| Pure plantation | Rainfall | 4 | 153.45 | <0.01 | 145.04 | <0.01 | 176.79 | <0.01 |
| | Species | 1 | 8.69 | <0.05 | 10.56 | <0.05 | 11.08 | <0.05 |
| | Rainfall ×Species | 4 | 129.89 | <0.01 | 112.46 | <0.01 | 4.99 | <0.01 |
| Mixed plantation | Rainfall | 4 | 1.5 | 0.41 | 2.3 | 0.11 | 18.34 | <0.01 |
| | Species | 1 | 2.2 | 0.21 | 1.48 | 0.29 | 3.9 | 0.12 |
| | Rainfall ×Species | 4 | 0.9 | 0.48 | 2.41 | 0.09 | 1.9 | 0.16 |
| *H. rhamnoides* | Rainfall | 4 | 2.05 | 0.14 | 1.51 | 0.25 | 85.46 | <0.01 |
| | Plantation type | 1 | 1.07 | 0.36 | 1.32 | 0.32 | 10.08 | <0.05 |
| | Rainfall × Plantation type | 4 | 0.62 | 0.66 | 1.39 | 0.28 | 5.59 | <0.01 |
| *P. tomentosa* | Rainfall | 4 | 14.72 | <0.01 | 71.59 | <0.01 | 19.46 | <0.01 |
| | Plantation type | 1 | 4.1 | 0.12 | 5.68 | 0.08 | 123.27 | <0.01 |
| | Rainfall × Plantation type | 4 | 9.55 | <0.01 | 85.29 | <0.01 | 9.35 | <0.01 |

$df$ = degree of freedom, Plantation type = pure and mixed plantation for each species. Pure and Mixed plantation indicate the result of water sources from different soil layers for both species in different plantation types, respectively; *H. rhamnoides* and *P. tomentosa* indicate the mixed afforestation effect on water sources from different soil layers for these species.

**Table S6.** Repeated ANOVA (ANOVAR) parameters for predawn ($\Psi_{pd}$), midday leaf water potential ($\Psi_m$), and leaf water potential gradient ($\Psi_{pd}-\Psi_m$) for *H. rhamnoides* and *P. tomentosa* (n = 30).

| | Variation source | df | $\Psi_{pd}$ | | $\Psi_m$ | | $\Psi_{pd}-\Psi_m$ | |
|---|---|---|---|---|---|---|---|---|
| | | | F | p | F | p | F | p |
| Pure plantation | Rainfall | 4 | 4.02 | <0.05 | 24.44 | <0.01 | 47.88 | <0.01 |
| | Species | 1 | 182.74 | <0.01 | 4.9 | <0.05 | 969.97 | <0.01 |
| | Rainfall × Species | 4 | 3.24 | <0.05 | 2.08 | 0.13 | 18.68 | <0.01 |
| Mixed plantation | Rainfall | 4 | 0.66 | 0.63 | 25.54 | <0.01 | 82.49 | <0.01 |
| | Species | 1 | 0.12 | 0.75 | 127.3 | <0.01 | 3420.1 | <0.01 |
| | Rainfall × Species | 4 | 1.8 | 0.18 | 3.7 | <0.05 | 35.92 | <0.01 |
| *H. rhamnoides* | Rainfall | 4 | 7.14 | <0.01 | 19.64 | <0.01 | 3.59 | <0.05 |
| | Plantation type | 1 | 27.05 | <0.01 | 496.66 | <0.01 | 1278.96 | <0.01 |
| | Rainfall × Plantation type | 4 | 1.69 | 0.202 | 3.32 | <0.05 | 6.66 | <0.01 |
| *P. tomentosa* | Rainfall | 4 | 30.78 | <0.01 | 12.39 | <0.01 | 7.38 | <0.01 |
| | Plantation type | 1 | 792.77 | <0.01 | 2.97 | 0.16 | 634.12 | <0.01 |
| | Rainfall × Plantation type | 4 | 3.8 | <0.05 | 0.09 | 0.98 | 3.83 | <0.05 |

*df* = degree of freedom, Plantation type = pure and mixed plantation for each species. Pure and Mixed plantation indicate the result of leaf water potential for both species in different plantation types, respectively; *H. rhamnoides* and *P. tomentosa* indicate the mixed afforestation effect on leaf water potential for these species.

**Table S7.** Regression of reference evapotranspiration ($ET_0$) and relative response of normalized sap flow ($SF_R$).

| Independent factors | *H. rhamnoides* in pure plantation | | *H. rhamnoides* in mixed plantation | | *P. tomentosa* in pure plantation | | *P. tomentosa* in mixed plantation | |
|---|---|---|---|---|---|---|---|---|
| | $R^2$ | p | $R^2$ | p | $R^2$ | p | $R^2$ | p |
| $ET_0$ | 0.18 | 0.47 | 0.11 | 0.59 | 0.44 | 0.22 | 0.39 | 0.26 |
| Relative response of $ET_0$ | 0.35 | 0.32 | 0.61 | 0.12 | 0.12 | 0.56 | 0.25 | 0.4 |

The regression equation is y=ax+b for all equations in this Table. Relative response of $ET_0$ is calculated as the same $SF_R$ in Eq. (4) in the manuscript, with before and the first day after rainfall event parameter is $ET_0$ instead.

**Table S8.** Parameters of allometric equation and average (mean ± SD) estimated biomass of leaf, branches, wood, and roots of *H. rhamnoides* and *P. tomentosa* in pure and mixed plantations (n=6).

| Species | | *a* | *b* | Biomass in pure plantation | Biomass in mixed plantation |
|---|---|---|---|---|---|
| *H. rhamnoides* | leaf | 0.017 | 0.541 | 0.51 ± 0.02 | 0.55 ± 0.04 |
| | branches | 0.013 | 0.042 | 0.16 ± 0.05 | 0.14 ± 0.01 |
| | wood | 0.036 | 0.721 | 2.4 ± 0.09 | 2.6 ± 0.07 |
| | roots | 0.019 | 0.732 | 1.51 ± 0.06 | 1.79 ± 0.04 |
| | total biomass | | | 4.58 ± 1.01 | 5.08 ± 1.13 |
| *P. tomentosa* | leaf | 0.052 | 0.621 | 1.21 ± 0.05 | 1.58 ± 0.09 |
| | branches | 0.025 | 0.81 | 1.35 ± 0.04 | 1.32 ± 0.06 |
| | wood | 0.0492 | 0.832 | 4.22 ± 0.11 | 4.73 ± 0.13 |
| | roots | 0.031 | 0.791 | 2.02 ± 0.06 | 2.75 ± 0.1 |
| | total biomass | | | 8.8 ± 1.39 | 10.38 ± 1.55 |

The allometric equation is $Y=a(D^2H)^b$, $Y$ is biomass (kg), $D$ is trunk diameter measured at 1.3 m above the ground (cm), $H$ is tree height (m). Six standard individuals of *H. rhamnoides* and *P. tomentosa* in pure and mixed plantations were selected for average $Y$ calculation.

**Figure Legends**

**Figure S1.** The geographic location of (a) study area and (b) plantation site in the Loess Plateau of China, and (c) monthly average (mean ±SD) rainfall amount and air temperature (Ta) during 2000-2017, and monthly rainfall amount and average Ta in 2018. Plantation types including *H. rhamnoides* pure plantation, *P. tomentosa* pure plantation, and Mixed plantation. Three adjacent plots were selected (16 m × 10 m) for each plantation type, and the schematic diagram of these plantation types is in (b). The China basic map can be obtained from

http://map.geoq.cn/arcgis/rest/services/ChinaOnlineCommunityENG/MapServer.

**Figure S2.** Independent-sample *t*-test for diurnal variation of average (mean ±SD) sap flow between the first and second day after rainfall amount of (a) 24 and (b) 35.2 mm for *P. tomentosa* in pure plantation. Error bars indicate the standard deviation (n = 3).

**Figure S3.** The linear regression relationship between $\delta^2H$ and $\delta^{18}O$ for soil water at three layers (0–30, 30–100, and 100–200cm) in (a) *H. rhamnoides* pure plantation, (b) *P. tomentosa* pure plantation, and (c) Mixed plantation. The local meteoric water line (LMWL) is plotted in each panel for reference.

**Figure S4.** Variation in average (mean ±SD) surface area of fine root at different soil depths for *H. rhamnoides* and *P. tomentosa* in pure (a) and mixed (b) plantations. Error bars indicate the standard deviation (n = 3).

**Figure S5.** Independent-sample *t*-test for diurnal variation of average (mean ±SD) sap flow before and after 5 rainfall events for *H. rhamnoides* in pure (a–e) and mixed plantation (f–j). Before and after rainfall indicated the value in the day before and first day after a rainfall event. Error bars indicate the standard deviation (n = 3).

**Figure S6.** Independent-sample *t*-test for diurnal variation of average (mean ±SD) sap flow before and after 5 rainfall events for *P. tomentosa* in pure (a–e) and mixed plantation (f–j). Before and after rainfall indicated the value in the day before and first day after a rainfall event. Error bars indicate the standard

deviation (n = 3).

**Figure S7.** Variation in average (mean $\pm$ SD) $\delta^{18}$O and $\delta$D of rainwater, stem water, and soil water at seven soil depths for *H. rhamnoides* in (a–e) pure and (k–o) mixed plantations and for *P. tomentosa* in (f–j) pure and (k–o) mixed plantations after 5 rainfall events. Error bars indicate the standard deviation (n = 3). The date of each 5 selected rainfall events is followed the corresponding rainfall amount value. The average rainwater $\delta^{18}$O and $\delta$D for each rainfall event is calculated with 3 rainwater subsamples, which was divided from one rainwater sample.

**Figure S8.** Relationship between rainfall amount and (a) relative response of normalized sap flow (SF$_R$) and (b) rainwater-recharged soil water uptake proportion (RUP) for *H. rhamnoides* in both plantation types, and these corresponding relationships for *P. tomentosa* (c–d) in both plantation types (n=3).

**Figure S1.**

[Figure]

**Figure S2.**

[Figure]

**Figure S3.**

[Figure]

**Figure S4.**

[Figure]

**Figure S5.**

[Figure]

**Figure S6.**

[Figure]

**Figure S7.**

[Figure]

**Figure S8.**

[Figure]

---

## Author Comment (AC4)

*The authors provide a compelling case demonstrating the differential response of plant water consumption to rainwater uptake for dominant tree species (Hippophae rhamnoides and Populus davidiana) in the semiarid Loess Plateau. I appreciated that the study used multiple indicators such as plant physiology (leaf water potential) and root morphology, sap flow and rainwater uptake proportion to comprehensively address this topic. This study suggested that H. rhamnoides and P. davidiana exhibited sensitive and insensitive response to rainfall pulses, respectively, which provides insights into suitable plantation species selection. While, I have three small questions that I don't understand, could you please answer them if it's convenient?*

**Response:** Thanks for these meaningful suggestions; the detailed answer to these questions can be observed the response to specific question as follows.

*(1)    line 189-190 I haven't figured out the relationship between Eqs 4 and 5. What does PAP mean?*

**Response: Corrected.** In response to this meaningful suggestion, the PAP has been corrected to RUP, and the revised Equations can be observed as follows:

"The RRS uptake proportion (RUP, %) after a recent rainfall pulse for plant was calculated as the proportion of rainwater in plant stem as follows (Cheng et al., 2006):

$$\delta^{18}O\ (D)_P = RUP \times \delta^{18}O\ (D)_{rain} + (1 - RUP) \times \delta^{18}O\ (D)_{swb} \tag{5}$$

$$RUP = (\delta^{18}O(D)_p - \delta^{18}O(D)_{swb})/(\delta^{18}O(D)_{swa} - \delta^{18}O(D)_{swb}) \times 100\% \tag{6}$$

where $\delta^{18}O(D)_{rain}$ and $\delta^{18}O(D)_p$ are the isotopic values for rainwater and plant stem after rainfall, respectively; $\delta^{18}O(D)_{swb}$ and $\delta^{18}O(D)_{swa}$ are the isotopic values of soil water immediately before and after rainfall, respectively. The Eq. (6) is derived through the linear mixing model for water isotopic value in plant stem after rainfall in Eq. (5). The RUP was the average value calculated in Eq. (6) based on $\delta^{18}O$ and $\delta D$, respectively, for specific plant species in each plot."

*(2) line 190 This study calculated RUP using D and 18O, respectively. Are the results of these two stable isotopes consistent?*

**Response: Added and Clarified.** Thanks for this meaningful suggestion, the relative sentence has been added in the revised manuscript in "**2.6 Statistical analysis**" section as follows: "The RUP was the average value calculated in Equation (6) based on $\delta^{18}O$ and $\delta D$, respectively, for specific plant species in each plot."

Indeed, the result of RUP calculated using $\delta^{18}$O and $\delta$D have the similar result in Equation (6). There are two reasons that can illustrate these results.

Firstly, the two studied plant species did not exhibited xerophytic or halophytic characteristic, which may cause water isotopic value fraction during water source uptake by these two plantation species. We also revised these two sentences in "**2.6 Statistical analysis**" section as follows: "No fractionation was considered during water source uptake by these plant roots because none of the plants exhibited xerophytic or halophytic characteristics. Ellsworth and Williams (2007) and Moore and Semmens (2008) suggested that a water stable isotope fractionation generally occurred during root uptake by xerophytic or halophytic plants."

Secondly, the water sources from three different soil layers calculated through MixSIR program (Moore and Semmens, 2008) also using both the $\delta^{18}$O and $\delta$D. And the result of water sources from rainwater recharged water after recent rainfall event, and the water sources from three soil layers were combined to analysis the water source variation response of these two plantation species. The relative sentences can be observed in "**2.6 Statistical analysis**" section as follows: "In addition to RUP, the water uptake proportions from different soil layers were calculated on the first day after a rainfall event using the MixSIR program, to complement the analysis of plant water source variations in response to rainfall pulses. The RUP method only calculated the proportion of recent rainwater in the plant stem and did not include soil water before the recent rainfall event (Gebauer and Ehleringer, 2000; Cheng et al., 2006). The water taken up from different soil layers by the plant is a mixture of soil water before the recent rainfall event and the recent rainwater."

**References:**

Gebauer, R. L. E., and Ehleringer, J. R.: Water and nitrogen uptake patterns following moisture pulses in a cold desert community, Ecology, 81, 1415-1424, 2000.

Cheng, X. L., An, S. Q., Li, B., Chen, J. Q., Lin, G. H., Liu, Y. H., Luo, Y. Q., and Liu, S. R.: Summer rain pulse size and rainwater uptake by three dominant desert plants in a desertified grassland ecosystem in northwestern China, Plant Ecol, 184, 1-12, 2006.

Ellsworth, P. Z., and Williams, D. G.: Hydrogen isotope fractionation during water uptake by woody xerophytes, Plant Soil, 291, 93-107, 10.1007/s11104-006-9177-1, 2007.

Moore, J. W., and Semmens, B. X.: Incorporating uncertainty and prior information into stable isotope mixing models, Ecol Lett, 11, 470-480, 10.1111/j.1461-0248.2008.01163.x, 2008.

*(3)    The study calculated the use of precipitation by plants after five rainfall events. I guess the use of precipitation by plants depends not only on the magnitude of the rainfall, but also on the antecedent soil water condition. How do you consider the potential impact that differences in antecedent soil water conditions may have on the results?*

**Response: Clarified.** Thanks for this meaningful suggestion. This study mainly focused on the influence of rainwater recharged soil water to plant transpiration after rainfall pulses. Theoretically, the rainfall amount, plant physiological adjustment, and antecedent soil water condition may influence the rainwater recharged soil water uptake proportion (RUP,%) for plant.

In the present study, the antecedent soil water content did not significantly influenced RUP for these two studied plantation species in either pure or mixed plantation types (P>0.05). In addition, the physiological adjustment ($\Psi_{pd}-\Psi_m$, difference between predawn and midday leaf water potential) significantly influenced the relative response of daily normalized sap flow ($SF_R$) for these species in pure and mixed plantations. These results also suggest that the physiological adjustment regulated the water absorb or consumption for these species in water limited regions.

---

## Author Response (AR1)

Authors' responses to Editor and Reviewers comments on the manuscript of "Differential response of plant transpiration to uptake of rainwater-recharged soil water for dominant tree species in the semiarid Loess Plateau". Manuscript ID: hess-2021-351.

**Dear Editor and Reviewers,**

We deeply appreciate you for giving us an opportunity to revise our manuscript. The marked-up manuscript version is highlighted the changes by using the red colored text in the manuscript (track-changes version). Here are the point-to-point responses (responses in upright Roman) to the Editor and Reviewers comments (*original comment in Itali*).

**Editor**

**Major Comments:**

*1) Your manuscript was read by four reviewers, who were generally interested in the study and made some useful comments to improve the manuscript. I see you have already made a large effort to improve the manuscript and it is good to see already the revised manuscript accompanying your response to the reviews. I think several corrections have improved the paper, amongst others the change in terminology made the paper easier to read. I do think that some of the responses to the reviews led to new unclarities or were not always very much to the point. As an example I add here some comments on your responses to reviewer 1. Seeing that you have made a significant effort and I do believe the manuscript has improved, I recommend to reconsider the article after major revision and will send it to the reviewers to get their opinion on the revised manuscript.*

**Response: Suggestions accepted.** Thanks for these meaningful suggestions, the relative sentences for runoff calculation and plant uptake deep soil water description in response to **Reviewer 1**, and for two hypotheses description in response to **Reviewer 2** have been rewritten in the revised manuscript.

**1)** In response to runoff calculation, we added the relative sentence in "2.4 Rainwater, plant stem, soil water, and leaf sample collection and measurement" subsection in "**2 Materials and methods**" section as follows: "In addition, no runoff was generated during the selected rainfall events in three plantations

according to the simulated result from the HYDRUS-1D model (Appendix A), which is based on the Richards' equation to describe soil water dynamics (Šimůnek et al., 2008). This model has been widely used to simulate the runoff and soil water dynamics in the Loess Plateau (Yi and Fan, 2016; Bai et al., 2020; Wang et al., 2020b)." (Page 8 Lines 195-199).

In the revised manuscript, the HYDRUS-1D model was used to calculate the runoff during the selected rainfall events in three plantations. This model has been widely used to simulate the runoff and soil water dynamics in the Loess Plateau (Yi and Fan, 2016; Bai et al., 2020; Wang et al., 2020b). The detailed information of this model, the basic data sources should be inputs in this model, the calibration and validation of this model, and the runoff result calculated by this model can be observed in **Appendix A** in the revised manuscript.

In Appendix A, **firstly**, we described the equation of the HYDRUS-1D model. After that, we pointed that this model has been widely used to simulate the runoff and soil water dynamics in the Loess Plateau (Yi and Fan, 2016; Bai et al., 2020; Wang et al., 2020b). The relative sentences can be observed as follows: "

The HYDRUS-1D model is based on the Richards' equation (Richards, 1931) to describe soil water dynamics (Šimůnek et al., 2008; Šimůnek et al., 2013):

$$\partial \theta / \partial t = \partial / \partial z \left( K(h,z)((\partial h / \partial z)+1) \right) - S_r(z,t) \tag{1}$$

where $\theta$, $t$, $h$, and $z$ are the soil moisture content (SW, $cm^3\ cm^{-3}$), simulation time (day), pressure head (cm), and vertical coordinate (cm), respectively. $K(h, z)$ and $S_r(z, t)$ are the unsaturated hydraulic conductivity ($cm\ day^{-1}$) (Mualem, 1976; van Genuchten, 1980) and root water uptake ($cm^3\ cm^{-3}\ day^{-1}$), respectively.

This model has been widely used to simulate soil water hydrological processes with HYDRUS-1D software (Šimůnek et al., 2013), such as soil water content dynamics and runoff in the Loess Plateau (Yi and Fan, 2016; Bai et al., 2020; Wang et al., 2020b). The model was used to calculate the runoff for each plantation type in this study, after calibration and validation this model using the observed SW. Based on suggestions in Yi and Fan (2016) and Bai et al. (2020), the atmospheric boundary condition with surface

runoff and free drainage were selected as upper and lower boundary condition, respectively, to calibrate and validate this model and calculate runoff (Fig. A1).

[Figure]

**Figure A1.** The upper and lower boundary conditions selection in HYDRUS-1D software (Version 4.15).

" (Pages 1-2 Lines 2-18 in Appendix A)

**Secondly**, the basic data sources should be inputs in this model were clearly described. These data sources included the observed meteorological, plant, and soil hydraulic parameters.

For meteorological parameters, the relative sentences can be observed in "**1.1 Meteorological parameters**" subsection in "**Appendix A**" as follows: "The meteorological parameters required for HYDRUS-1D include relative humidity, wind speed ($W_S$), air temperature, rainfall amount, and reference evapotranspiration ($ET_0$). Daily relative humidity, maximum, minimum and average air temperatures, $W_S$, and rainfall amount were measured by a weather station approximately 500 m from the research plots. The $ET_0$ (cm day$^{-1}$) was calculated through method described by Allen et al. (1998). The detailed information can be observed in "2.2 Environmental parameter measurements and ET$_0$ calculation" subsection in the manuscript." (Pages 2-3 Lines 24-29 in Appendix A)

For plant parameters, the relative sentences can be observed in "**1.2 Plant parameters**" subsection in "**Appendix A**" as follows: "The plant parameters required for HYDRUS-1D include plant height, root

depth, and potential transpiration rate. Plant height and root depth in each plantation type can be observed in Table S1 and Figure S4 in the manuscript, respectively. The leaf area index (LAI) was measured monthly, from May to September, for each plantation type using a LAI-2200 (LiCor Inc., Lincoln, USA). The potential transpiration rate (cm day$^{-1}$) was calculated using the Beer equation (Ritchie, 1972) based on the measured LAI and extinction coefficient value (0.39) suggested in Šimůnek et al. (2013)." (Page 3 Lines 31-36 in Appendix A)

For soil hydraulic parameters, the relative sentences can be observed in "**1.3 Soil hydraulic parameters**" subsection in "**Appendix A**" as follows: "The saturated soil water content ($\theta s$) and hydraulic conductivity ($Ks$), van Genuchten model parameters ($\alpha$ and $n$), and residual soil water content ($\theta r$) were required parameters for HYDRUS-1D. The $Ks$, $\theta s$, and soil bulk density (BD) at soil depth intervals of 0-20, 20-50, 50-100, and 100-200 cm were measured in July 2018 using the cutting ring (Wu et al., 2016) and constant water head (Reynolds et al., 2002) method in each plantation type. The soil particle composition was determined using a Mastersize 2000 (Malvern Instruments Ltd., UK). Additionally, the slopes for these three plantation types were required for HYDRUS-1D. The detailed information can be observed in "2.1 Study site" subsection in the manuscript. The measured soil hydraulic parameters for the three plantation types are shown in Table A1. The Rosetta pedotransfer function was used to calculate $\theta r$, $\alpha$, and $n$ (Jana and Mohanty, 2012; Bai et al., 2020).

**Table A1.** Measured soil hydraulic parameters and particle composition in both pure and mixed plantations

| | Soil depth (cm) | Soil particle composition | | | Soil hydraulic parameter | | |
| --- | --- | --- | --- | --- | --- | --- | --- |
| | | Sand (%) | Silt (%) | Clay (%) | $\theta_s$ (cm$^3$ cm$^{-3}$) | BD (g cm$^{-3}$) | Ks (cm day$^{-1}$) |
| | 0-20 | 26.4 | 63.5 | 10.1 | 0.37 | 1.28 | 75.7 |
| *H. rhamnoides* | 20-50 | 22.2 | 61.6 | 16.2 | 0.34 | 1.4 | 70.3 |
| pure plantation | 50-100 | 23.5 | 63.1 | 13.4 | 0.32 | 1.44 | 55.4 |
| | 100-200 | 24.7 | 63.8 | 11.5 | 0.29 | 1.48 | 50.6 |
| *P. tomentosa* | 0-20 | 25.8 | 62.2 | 12 | 0.35 | 1.21 | 82.4 |

| | | | | | | | |
|---|---|---|---|---|---|---|---|
| pure plantation | 20-50 | 23.7 | 62.5 | 13.8 | 0.35 | 1.33 | 73.7 |
| | 50-100 | 22.2 | 61.5 | 16.3 | 0.31 | 1.42 | 58.9 |
| | 100-200 | 24.9 | 64.8 | 10.3 | 0.3 | 1.45 | 52.6 |
| | 0-20 | 25.5 | 63.8 | 10.7 | 0.36 | 1.25 | 78.5 |
| Mixed | 20-50 | 24.3 | 62.7 | 13 | 0.35 | 1.31 | 73.2 |
| plantation | 50-100 | 23.8 | 64.9 | 11.3 | 0.34 | 1.39 | 60.5 |
| | 100-200 | 24.6 | 63.7 | 11.7 | 0.31 | 1.45 | 53.4 |

$\theta_s$= saturated soil water content, $Ks$=saturated hydraulic conductivity, BD= soil bulk density "

" (Pages 3-4 Lines 38-50 in Appendix A)

**Thirdly,** the measured SW in the present study in each plantation type was used for model calibration and validation. The SW at each soil depths in each plantation type from DOY 132 to 202 was used to calibrate the HYDRUS-1D. And some soil hydraulic parameters were optimized through the inverse solution module in HYDRUS-1D during the model calibration (Table A2).

**Table A2.** Optimized soil hydraulic parameters in both pure and mixed plantations through HYDRUS-1D

| | Soil depth (cm) | $\theta r$ (cm$^3$ cm$^{-3}$) | $\theta s$ (cm$^3$ cm$^{-3}$) | Ks (cm day$^{-1}$) | $a$ | $n$ |
|---|---|---|---|---|---|---|
| | 0-20 | 0.08 | 0.36 | 74.9 | 0.018 | 1.6 |
| *H. rhamnoides* pure | 20-50 | 0.08 | 0.34 | 71.2 | 0.018 | 1.6 |
| plantation | 50-100 | 0.0823 | 0.31 | 56.2 | 0.01 | 1.45 |
| | 100-200 | 0.0823 | 0.3 | 51.5 | 0.01 | 1.43 |
| | 0-20 | 0.08 | 0.36 | 82.1 | 0.019 | 1.62 |
| | 20-50 | 0.08 | 0.35 | 73.5 | 0.018 | 1.6 |
| *P. tomentosa* pure plantation | 50-100 | 0.0821 | 0.31 | 59.2 | 0.01 | 1.51 |
| | 100-200 | 0.0822 | 0.31 | 51.6 | 0.011 | 1.47 |

| | Soil depth | | | | |
|---|---|---|---|---|---|
| | 0-20 | 0.08 | 0.37 | 79.2 | 0.018 | 1.61 |
| Mixed plantation | 20-50 | 0.08 | 0.36 | 74.2 | 0.018 | 1.61 |
| | 50-100 | 0.0822 | 0.34 | 60.2 | 0.011 | 1.46 |
| | 100-200 | 0.0823 | 0.3 | 55.8 | 0.011 | 1.45 |

$\theta_r$= residual soil water content, $Ks$=saturated hydraulic conductivity, $\theta_s$= saturated soil water content, $a$ and $n$ = parameters of van Genuchten model.

**Subsequently**, the SW from DOY 203 to 273 in each plantation type was used to validate the model. The root mean square error (RMSE), Nash-Sutcliffe efficiency coefficient (NSE), and determinant coefficient ($R^2$) based on the observed and simulated SW was used to evaluate the model performance (Bai et al., 2020). The calculated RMSE, NSE, and $R^2$ indicated that the simulated results were acceptable for three plantation types in this study (Table A3), based on the criterions suggested in Bai et al. (2020) and Wang et al. (2020b). The detailed sentences can be observed in "2 Model calibration and validation, and runoff calculation" section in **"Appendix A".** (Pages 5-10 Lines 53-96 in Appendix A)

**Table A3.** The RMSE, NSE, and $R^2$ between the observed and simulated SW during the HYDRUS-1D validation period (from DOY 203-273)

| | Soil depth (cm) | RMSE | NSE | $R^2$ |
|---|---|---|---|---|
| *H. rhamnoides* pure plantation | 5 | 0.008 | 0.65 | 0.84 |
| | 20 | 0.006 | 0.58 | 0.83 |
| | 50 | 0.006 | 0.7 | 0.71 |
| | 100 | 0.008 | 0.56 | 0.85 |
| | 150 | 0.005 | 0.59 | 0.81 |
| | 200 | 0.006 | 0.52 | 0.78 |
| *P. tomentosa* pure plantation | 5 | 0.008 | 0.67 | 0.79 |
| | 20 | 0.008 | 0.62 | 0.76 |
| | 50 | 0.006 | 0.72 | 0.82 |
| | 100 | 0.009 | 0.59 | 0.75 |

| | | | |
|---|---|---|---|
| | 150 | 0.008 | 0.57 | 0.83 |
| | 200 | 0.009 | 0.61 | 0.78 |
| | 5 | 0.009 | 0.61 | 0.81 |
| | 20 | 0.01 | 0.54 | 0.76 |
| | 50 | 0.008 | 0.68 | 0.82 |
| Mixed plantation | 100 | 0.008 | 0.7 | 0.79 |
| | 150 | 0.006 | 0.76 | 0.82 |
| | 200 | 0.008 | 0.67 | 0.81 |

RMSE= root mean square error, NSE = Nash-Sutcliffe efficiency coefficient, $R^2$= determinant coefficient

The simulated SWs at different soil depths closely matched the variation of these values observed from DOY 203 to 273, the example can be observed in the *H. rhamnoides* pure plantation in Figs. A1 and A2.

[Figure]

**Figure A1.** Variation in soil water content (SW) at 5, 50, 50, 100, 150, and 200 cm depths during the HYDRUS-1D (a-f) calibration (from DOY 132-202) and (g-l) validation (from DOY 203-273) period in *H. rhamnoides* pure plantation.

[Figure]

**Figure A2.** The relationship between observed ($SW_O$) and simulated ($SW_s$) soil water content at 5, 50, 50, 100, 150, and 200 cm depths during the HYDRUS-1D validation   period (from DOY 203-273) in *H. rhamnoides* pure plantation.

**Fourthly,** the runoff was calculated through HYDRUS-1D in three plantation types. The result from this model indicated that no runoff was generated during the studied period from DOY 132 to 273. Thus, we expect that no runoff was generated during the time of our selected rainfall events. The detailed sentences can be observed in "2 Model calibration and validation, and runoff calculation" section in **"Appendix A".** (Page 10 Lines 98-101 in Appendix A)

**2)** In response to the description of plant uptake deep soil water, the relative sentences have been rewritten in "4.2 RRS uptake enhances plant transpiration for coexisting species in mixed plantation" subsection in "**4 Discussion**" section as follows: "Furthermore, similar to other studies in the Loess Plateau (Wang et al., 2020a; Wu et al., 2021), the deep soil layer generally exhibited lower SW than other soil layers in all plantation types in the present study (Fig. 1, Table S3). Jia et al. (2017) and Wang et al. (2020a) attributed the lower SW in deep soil layers to the imbalance between rainwater replenishment and plant uptake of water from this soil layer in the studied region. Silvertown et al. (2015) and Tang et al. (2019) suggested that coexisting plant species generally reduce water uptake from soil layers that exhibit low soil water content to avoid water source competition in these layers and maintain stable coexistence. In the present study, consistent with the second adaptation type, mixed afforestation significantly decreased the water uptake proportion from the deep soil layer for these species (Table S5). Thus, both increased rainwater-recharged soil water uptake and decreased water source competition from the deep soil layer were adopted by these species in the mixed plantation to minimize water sources competition under water limited conditions." (Page 23 Lines 504-514)

This paragraph mainly described the second adaptation type of coexisting plant species to cope with resource competition. Two types of adaptation can be adopted by plants among plant species to cope with resource competition in mixed plantations. The sentence has been mentioned in the previous paragraph: "Generally, two types of adaptation can be adopted by plants to cope with resource competition: increased competition ability or minimized competition interactions (West et al., 2007)." (Page 22 Lines 485-486).

In this paragraph, **Firstly,** we pointed out that the deep soil layer generally exhibited lower SW than other soil layers in all plantation types in the present study (Fig. 1, Table S3). Jia et al. (2017) and Wang et al. (2020a) attributed the lower SW in deep soil layers to the imbalance between rainwater replenishment and plant uptake of water from this layer. This is mainly because, during the dry period in this semiarid region, the SW in shallow and middle soil layers may similar or lower than the value in deep soil layer. For example, during DOY 180-200, the SW at 5 cm and 20 cm was lower than the value at 150cm (red cycle in Figure 1d as follows). Under this water limited conditions, plant generally absorbed more water from deep than from other soil layers. In general, at seasonal or annual timescale, the imbalance between rainwater replenishment and plant uptake of water from deep soil layer result to the lower SW in deep soil layers in the Loess Plateau (Figure 1). Coexisting plant species may reduce their water uptake proportion from deep soil layer and thus reduce water competition at this soil layer, compared with water sources for specific species in pure plantation.

Silvertown et al. (2015) and Tang et al. (2019) suggested that coexisting plant species generally reduce water uptake from soil layers that exhibit low soil water content to avoid water source competition in these layers and maintain stable coexistence. In addition, our result indicated that "mixed afforestation significantly decreased the water uptake proportion from the deep soil layer for these species". Thus, we point out that "In the present study, consistent with the second adaptation type, mixed afforestation significantly decreased the water uptake proportion from the deep soil layer for these species (Table S5)." (Page 23 Lines 510-512).

[Figure]

**Figure 1.** Variation in (a) rainfall amount, reference evapotranspiration ($ET_0$), and average (mean $\pm$ SD) soil water content (SW) in (b) *H. rhamnoides* pure plantation, (c) *P. tomentosa* pure plantation, and (d) mixed plantation from DOY 132 to 273 (11 May to 30 September) (n = 3). Standard deviation bars for SW at each soil layers are not shown to allow clear display of variation of SW for each plantation. Arrows in (a) indicate dates of sample collection at the first day after rainfall events: DOY 157 (6 June), DOY 194 (12 July), DOY 204 (23 July), DOY 249 (6 September), and DOY 265 (22 September).

**Secondly,** combined with the first adaptation type discussed in the previous paragraph (Page 22 Lines 486-495), we suggested that "Thus, both increased rainwater-recharged soil water uptake and decreased water source competition from the deep soil layer were adopted by these species in the mixed plantation to minimize water sources competition under water limited conditions." (Page 23 Lines 512-514)

**3)** Based on suggestion by **Reviewer 2,** two hypotheses and these hypotheses verification have been added in the previous revised manuscript. We have been rewritten the verification of these hypotheses in the current revised manuscript.

Two hypotheses have been added in the revised manuscript in "**1 Introduction**" section as follows: "Based on variations of plant water uptake from different soil layers and leaf water potential for these species in Xi et al. (2013) and Tang et al. (2019), we hypothesize that (1) the influence of RRS uptake

and leaf water potential on plant transpiration may differ for these species in pure plantations, and (2) these influences may differ for specific species in pure and mixed plantations." (Page 4 Lines 102-106). RRS is the abbreviation for "Rainwater-recharged soil water", which has been mentioned at the first appearance in the first sentence in "**1 Introduction**" section (Page 2 Lines 39-42).

These tow hypotheses have also been verified in "**4 Discussion**" section. In response to the first hypothesis, the sentence has been rewritten in "4.1 RRS uptake enhances plant transpiration for *H. rhamnoides* but not *P. tomentosa* in pure plantations" subsection as follows: "Consistent with the first hypothesis, the influence of RRS uptake and $\Psi_{pd}-\Psi_m$ on $SF_R$ was different for these species in pure plantations." (Page 20 Lines 451-452). The $SF_R$ was significantly influenced by RUP and $\Psi_{pd}-\Psi_m$ for *H. rhamnoides* in the pure plantation, indicating that RRS uptake and leaf physiological adjustment enhanced its plant transpiration (Figs. 6 and 7). However, the $SF_R$ was significantly influenced by $\Psi_{pd}-\Psi_m$ for *P. tomentosa* (Fig. 6), suggesting that its transpiration was mainly constrained by plant physiological characteristics.

In response to the second hypothesis, the relative sentence has been added in "4.2 RRS uptake enhances plant transpiration for coexisting species in mixed plantation" subsection: "The different influence of RUP and $\Psi_{pd}-\Psi_m$ on $SF_R$ for specific species in pure and mixed plantations was consistent with the second hypothesis. The significant influence of RUP and $\Psi_{pd}-\Psi_m$ on $SF_R$ was observed for *P. tomentosa* in mixed plantation (Fig. 6). Meanwhile, for *H. rhamnoides* in mixed plantation compared to specific value in pure plantation, larger and smaller slopes in linear regression were observed between $SF_R$ and RUP, and $SF_R$ and $\Psi_{pd}-\Psi_m$, respectively (Fig. 6)." (Page 22 Lines 495-499).

[Figure]

**Figure 6.** Relationship of average (a, b) rainwater-recharged soil water uptake proportion (RUP) and (c, d) leaf water potential gradient ($\Psi_{pd}-\Psi_{m}$) with relative response of normalized $F_d$ ($SF_R$) for *H. rhamnoides* and *P. tomentosa* in both plantation types (n = 3).

$$\partial\theta / \partial t = \partial / \partial z \big( K(h, z)((\partial h / \partial z) + 1) \big) - S_r(z, t)$$

(1)

where $\theta$, $t$, $h$, and $z$ are the soil moisture content (SW, cm$^3$ cm$^{-3}$), simulation time (day), pressure head (cm), and vertical coordinate (cm), respectively. $K(h, z)$ and $S_r(z, t)$ are the unsaturated hydraulic conductivity (cm day$^{-1}$) (Mualem, 1976; van Genuchten, 1980) and root water uptake (cm$^3$ cm$^{-3}$ day$^{-1}$), respectively.

This model has been widely used to simulate soil water hydrological processes with HYDRUS-1D software (Šimůnek et al., 2013), such as soil water content dynamics and runoff in the Loess Plateau (Yi and Fan, 2016; Bai et al., 2020; Wang et al., 2020b). The model was used to calculate the runoff for each plantation type in this study, after calibration and validation this model using the observed SW. Based on suggestions in Yi and Fan (2016) and Bai et al. (2020), the atmospheric boundary condition with surface runoff and free drainage were selected as upper and lower boundary condition, respectively, to calibrate and validate this model and calculate runoff (Fig. A1).

[Figure]

**Figure A1.** The upper and lower boundary conditions selection in HYDRUS-1D software (Version 4.15).

" (Pages 1-2 Lines 2-18 in Appendix A)

**Secondly**, the basic data sources should be inputs in this model were clearly described. These data sources included the observed meteorological, plant, and soil hydraulic parameters.

For meteorological parameters, the relative sentences can be observed in "**1.1 Meteorological parameters" subsection in "Appendix A"** as follows: "The meteorological parameters required for HYDRUS-1D include relative humidity, wind speed ($W_S$), air temperature, rainfall amount, and reference evapotranspiration ($ET_0$). Daily relative humidity, maximum, minimum and average air temperatures, $W_S$, and rainfall amount were measured by a weather station approximately 500 m from the research plots. The $ET_0$ (cm day$^{-1}$) was calculated through method described by Allen et al. (1998). The detailed information can be observed in "2.2 Environmental parameter measurements and $ET_0$ calculation" subsection in the manuscript." (Pages 2-3 Lines 24-29 in Appendix A)

For plant parameters, the relative sentences can be observed in "**1.2 Plant parameters" subsection in "Appendix A"** as follows: "The plant parameters required for HYDRUS-1D include plant height, root depth, and potential transpiration rate. Plant height and root depth in each plantation type can be observed in Table S1 and Figure S4 in the manuscript, respectively. The leaf area index (LAI) was measured monthly, from May to September, for each plantation type using a LAI-2200 (LiCor Inc., Lincoln, USA). The potential transpiration rate (cm day$^{-1}$) was calculated using the Beer equation (Ritchie, 1972) based on the measured LAI and extinction coefficient value (0.39) suggested in Šimůnek et al. (2013)." (Page 3 Lines 31-36 in Appendix A)

For soil hydraulic parameters, the relative sentences can be observed in "**1.3 Soil hydraulic parameters" subsection in "Appendix A"** as follows: "The saturated soil water content ($\theta s$) and hydraulic conductivity ($Ks$), van Genuchten model parameters ($\alpha$ and $n$), and residual soil water content ($\theta r$) were required parameters for HYDRUS-1D. The $Ks$, $\theta s$, and soil bulk density (BD) at soil depth intervals of 0-20, 20-50, 50-100, and 100-200 cm were measured in July 2018 using the cutting ring (Wu et al., 2016) and constant water head (Reynolds et al., 2002) method in each plantation type. The soil particle composition was determined using a Mastersize 2000 (Malvern Instruments Ltd., UK). Additionally, the slopes for these three plantation types were required for HYDRUS-1D. The detailed information can be observed in "2.1 Study site" subsection in the manuscript. The measured soil

[revised manuscript text omitted]

*2) Throughout the manuscript, there are also some instances where the term seems inappropriately use (e.g. only). I would suggest going through the entire paper and refining the language to more accurately reflect the result.*

**Response: Rewritten and clarified.** Thanks for your suggestion, the entire manuscript has been reviewed and the relative terms have been rewritten in the revised version.

For example, based on the suggestion by the other reviewer, the term "plant water consumption" and "rainwater uptake" has been revised to "plant transpiration" and "rainwater-recharged soil water",

respectively, in the revised manuscript. The RRS was used as the abbreviation for "rainwater-recharged soil water" in the revised manuscript. And the **Title** of the revised manuscript has also been rewritten as "Differential response of plant transpiration to uptake of rainwater-recharged soil water for dominant tree species in the semiarid Loess Plateau" (Page 1 Lines 1-2).

For example, the "only" has also been deleted in the revised manuscript in "**Abstract**" section as follows: "In pure plantations, the relative response of daily normalized sap flow ($SF_R$) was significantly affected by RRS uptake proportion (RUP) and $\Psi_{pd}-\Psi_m$ for *H. rhamnoides*, and was significantly influenced by $\Psi_{pd}-\Psi_m$ for *P. tomentosa* ($P < 0.05$)." (Page 1 Lines 20-22)

*3) Potential/Reference Evapotranspiration is a key parameter indicator that reflect atmospheric evaporative demand, and also support some part of you conclusion. However, why the Reference evapotranspiration (ET0) was used in the study, because there are some other indicator also reflect the evaporative demand.*

**Response: Clarified and rewritten.** In response to this meaningful suggestion, the advantage of Reference evapotranspiration ($ET_0$) has been added in "2.2 Environmental parameter measurements and $ET_0$ calculation" subsection in "**2 Materials and methods**" section as follows: "$ET_0$, considering both aerodynamic characteristics and energy balance, was used to indicate atmospheric evaporative demand (Allen et al., 1998):" (Page 6 Lines 153-154).

Indeed, there are several Equations that calculated the potential or reference evapotranspiration. The $ET_0$ equation in the present study is used as the standard method by the FAO (Food and Agriculture Organization of the United Nations), and has been widely used for evaluate other $ET_0$ equations (Xiang et al., 2020). The advantage of the Equation that we used considered both aerodynamic aspects and energy balance, because evapotranspiration is a process that liquid water is converted vapor phase and then the vapor moves. The detailed information can be observed in a review of difference of reference crop evapotranspiration in Xiang et al. (2020).

DATE: March 9, 2022

Compuscript Ltd
T/A International Science Editing
Bay K, Shannon Industrial Park West
Shannon, Co Clare
Ireland
Phone +353 61 472818   Fax +353 61 472688

To whom it may concern,

The paper "Differential response of plant transpiration to uptake of rainwater-recharged soil water for dominant tree species in the semiarid Loess Plateau" by Yakun Tang was edited by International Science Editing. We were asked not to edit the references. Please contact us if you would like to view the edited paper.

Kindest regards,

David Cushley.

If the English is still not meet the standard, please give me another chance, I will revised the language by another scientific editing service company again.

**Minor Comments:**

*1) Lines 22 "only" is too arbitrary*

**Response: Deleted.** This sentence has been rewritten in "**Abstract**" section as follows: "In pure plantations, the relative response of daily normalized sap flow ($SF_R$) was significantly affected by RRS

uptake proportion (RUP) and $\Psi_{pd}-\Psi_m$ for *H. rhamnoides*, and was significantly influenced by $\Psi_{pd}-\Psi_m$ for *P. tomentosa* (P < 0.05)." (Page 1 Lines 20-22).

*2) Lines 30-32 "Regardless of sensitivity to rainfall pulses" ? this short sentence should be rewritten.*

**Response: Rewritten.** Thanks for this meaningful suggestion**,** this sentence has been rewritten in "**Abstract**" section as follows: "These results indicate that mixed afforestation enhanced the influence of RRS uptake to plant transpiration for these different rainfall pulse sensitive plants." (Page 2 Lines 30-31).

*3) Lines 54-57 The "water uptake" should also be clearly described.*

**Response: Rewritten.** In response to this meaningful suggestion, the sentence has been rewritten in "**1 Introduction**"as follows: "The controversial rainfall pulse response between RRS uptake and plant transpiration may be mainly attributed to an inconsistent influence of plant leaf physiological characteristics (West et al., 2007), root morphology adjustment (Wang et al., 2020a), or environmental conditions (Tfwala et al., 2019) on these two water processes." (Pages 2-3 Lines 53-56).

*4) Lines 69-71 the author should be clarified this sentence for pure or coexisting species? Because the similar meaning and sentence can be observed at Lines 57-60.*

**Response: Revised.** According to this suggestion, this sentence has been revised in "**1 Introduction**" section as follows: "Rainfall pulses have been observed to relieve or eliminate water competition among coexisting species and thus maintain or increase plant transpiration in some water limited regions (Wang et al., 2020a; Tfwala et al., 2019)." (Page 3 Lines 68-70)

Indeed, this sentence should be clarified the influence of rainfall pulses on water competition among coexisting species.

*5) Lines 131-132 Please clarify why the Reference evapotranspiration (ET0) was used in the study, as a large number of indicators can reflect atmospheric evaporative demand.*

**Response: Clarified and rewritten.** In response to this meaningful suggestion, the advantage of Reference evapotranspiration ($ET_0$) has been added in "2.2 Environmental parameter measurements and $ET_0$ calculation" subsection in "**2 Materials and methods**" section as follows: "$ET_0$, considering both aerodynamic characteristics and energy balance, was used to indicate atmospheric evaporative demand (Allen et al., 1998):" (Page 6 Lines 153-154). The detailed explanation can be observed the response to *Major Comments 3)*.

*6) Lines 213-214 This sentence is nonsense and should be deleted.*

**Response: Deleted and Rewritten.** In response to this suggestion, the sentence in the previous manuscript has been deleted and rewritten in the revised manuscript in "2.6.2 Calculation of RRS uptake proportion and water sources from different soil layers" subsection in "**2 Materials and methods**" section as follows: "In addition to RUP, the water uptake proportions from different soil layers were calculated on the first day after a rainfall event using the MixSIR program, to complement the analysis of plant water source variations in response to rainfall pulses. The RUP method only calculated the proportion of recent rainwater in the plant stem and did not include soil water before the recent rainfall event (Gebauer and Ehleringer, 2000; Cheng et al., 2006). The water taken up from different soil layers by the plant is a mixture of soil water before the recent rainfall event and the recent rainwater." (Page 11 Lines 270-275)

[Figure]

**Figure 1.** Variation in (a) rainfall amount, reference evapotranspiration (ET$_0$), and average (mean ±SD) soil water content (SW) in (b) *H. rhamnoides* pure plantation, (c) *P. tomentosa* pure plantation, and (d) mixed plantation from DOY 132 to 273 (11 May to 30 September) (n = 3). Standard deviation bars for SW at each soil layers are not shown to allow clear display of variation of SW for each plantation. Arrows in (a) indicate dates of sample collection at the first day after rainfall events: DOY 157 (6 June), DOY 194 (12 July), DOY 204 (23 July), DOY 249 (6 September), and DOY 265 (22 September).

**Secondly,** combined with the first adaptation type discussed in the previous paragraph (Page 22 Lines 486-495), we suggested that "Thus, both increased rainwater-recharged soil water uptake and decreased water source competition from the deep soil layer were adopted by these species in the mixed plantation to minimize water sources competition under water limited conditions." (Page 23 Lines 512-514)

These tow hypotheses have also been verified in "**4 Discussion**" section. In response to the first hypothesis, the sentence has been rewritten in "4.1 RRS uptake enhances plant transpiration for *H. rhamnoides* but not *P. tomentosa* in pure plantations" subsection as follows: "Consistent with the first hypothesis, the influence of RRS uptake and $\Psi_{pd}-\Psi_m$ on $SF_R$ was different for these species in pure plantations." (Page 20 Lines 451-452). The $SF_R$ was significantly influenced by RUP and $\Psi_{pd}-\Psi_m$ for *H. rhamnoides* in the pure plantation, indicating that RRS uptake and leaf physiological adjustment enhanced its plant transpiration (Figs. 6 and 7). However, the $SF_R$ was significantly influenced by $\Psi_{pd}-\Psi_m$ for *P. tomentosa* (Fig. 6), suggesting that its transpiration was mainly constrained by plant physiological characteristics.

In response to the second hypothesis, the relative sentence has been added in "4.2 RRS uptake enhances plant transpiration for coexisting species in mixed plantation" subsection: "The different influence of RUP and $\Psi_{pd}-\Psi_m$ on $SF_R$ for specific species in pure and mixed plantations was consistent with the second hypothesis. The significant influence of RUP and $\Psi_{pd}-\Psi_m$ on $SF_R$ was observed for *P. tomentosa* in mixed plantation (Fig. 6). Meanwhile, for *H. rhamnoides* in mixed plantation compared to specific value in pure plantation, larger and smaller slopes in linear regression were observed between $SF_R$ and RUP, and $SF_R$ and $\Psi_{pd}-\Psi_m$, respectively (Fig. 6)." (Page 22 Lines 495-499).

[Figure]

**Figure 6.** Relationship of average (a, b) rainwater-recharged soil water uptake proportion (RUP) and (c, d) leaf water potential gradient ($\Psi_{pd}-\Psi_{m}$) with relative response of normalized $F_d$ ($SF_R$) for *H. rhamnoides* and *P. tomentosa* in both plantation types (n = 3).

**3)** In response to plant root distribution of these species, the plant root distribution has been conducted in our present study. The method to investigate the plant root distribution was in the "2.5 Plant fine root investigation" subsection in "**2 Materials and methods**" section as follows: "In August 2018, 4 soil cores were dug around each selected standard individual for plant stem and soil water collection, through a soil drill with diameter 20 cm to investigate plant fine roots. The collected soil depths were 0–10, 10–20, 20–30, 30–50, 50–70, 70–100, 100–130, 130–150, 150–200 cm, with approximately 0.5 m around the stem of each species standard individual. The sum of root samples for 4 soil cores at each soil depth for each selected standard individual was used for fine root distribution analysis, giving 324 fine root samples. WinRHIZO (Regent Instruments Inc., Quebec, Canada) was used to determine the fine root (diameter < 2 mm) surface area at each soil depth. The average fine root surface area for specific species at each soil depth was calculated in each plot for further analysis." (Page 9 Lines 225-232).

The results can be observed as in **Figure S4** as follows. These fine root surface area distribution was

mainly used to illustrate the plant water sources of these two species in pure and mixed plantations in "**3 Results**" and "**4 Discussion**" section.

The relative sentences can be observed in "3.1 Variation in environmental parameters and plant fine root vertical distribution" subsection in "**3 Results**" section as follows: "The *H. rhamnoides* and *P. tomentosa* in pure plantations exhibited different fine root vertical distributions, with more than 40% of fine roots observed in shallow and deep soil layers, respectively (Fig. S4). In the mixed plantation, approximately 40% of *H. rhamnoides* fine roots were in the shallow soil layer. Meanwhile, no significant differences in fine root proportion were observed for *H. rhamnoides* for each soil layer in pure and mixed plantations (P > 0.05). The fine root proportion of *P. tomentosa* in the shallow soil layer was significantly increased from 21.94% in pure plantation to 31.28% in the mixed plantation (P < 0.05)." (Pages 12-13 Lines 321-327).

In "4.1 RRS uptake enhances plant transpiration for *H. rhamnoides* but not *P. tomentosa* in pure plantations" subsection in "**4 Discussion**" section as follows: "This may be mainly due to the greater proportions of fine root surface area distributed in the shallow soil layer for *H. rhamnoides* (40.85 ± 3.14%) compared to *P. tomentosa* (21.94 ± 2.3%) (Fig. S4)." (Page 19 Lines 428-430).

Also, in "4.2 RRS uptake enhances plant transpiration for coexisting species in mixed plantation" subsection in "**4 Discussion**" section as follows: "Mixed afforestation significant increased $\Psi_{pd}$ for *P. tomentosa*, possibly due to the advantage of access to soil moisture recharged by rainwater through an increased root surface area in the shallow soil layer for this species in the mixed plantation (Fig. S4)." (Page 22 Lines 490-492).

Thus, the description of root distribution for these two species was not emphasized in "**1 Introduction**" section.

[Figure]

**Figure S4.** Variation in average (mean ± SD) surface area of fine root at different soil depths for *H. rhamnoides* and *P. tomentosa* in pure (a) and mixed (b) plantations. Error bars indicate the standard deviation (n = 3).

**4)** Thanks for this careful examination; the Latin name of previous "*Hippophae rhamnoidesis*" has been corrected to "*Hippophae rhamnoides* subsp. *sinensis*" at the first appearance in "**Abstract**" and "**1 Introduction**" section, respectively, in the revised manuscript. Then, the acronym name "*H. rhamnoides*" was used throughout the manuscript.

The revised name of *Hippophae rhamnoides* subsp. *sinensis* can be observed in Huang et al. (2018), and also in *World Flora Online* as follows:

[Figure]

manuscript in "2.4 Rainwater, plant stem, soil water, and leaf sample collection and measurement" subsection in "**2 Materials and methods**" section as follows: "At each of successive three days after every selected rainfall event, one suberized stem after removing the bark was collected at midday (11:30–13:30) for each standard individual. Meanwhile, approximately 0.5 m around the stem of each standard individual in the pure plantations and at the middle between two species in the mixed plantation, one soil core at seven depths (0–10, 10–20, 20–30, 30–50, 50–100, 100–150, and 150–200 cm) was collected through soil drilling. The suberized stem and collected soil samples were placed into glass bottles. These bottles were sealed with parafilm and stored at −15 ℃. On the same day as plant stem and soil sample collections, one leaf was selected from each sap flow monitored individual for leaf water potential measurement. The $\Psi_{pd}$ and $\Psi_m$ were measured by a PMS1515D analyzer (PMS Instrument, Corvallis Inc., OR, USA) at predawn (4:30–5:30) and midday (11:20–12:40), respectively.

All the plant stem, soil, and leaf samples collected on the first day after a rainfall pulse were used for analysis, with the detailed given in section "2.6 Statistical analysis". There were 180 stem and 945 soil samples for water extraction, and 180 leaf samples for $\Psi_{pd}$ and $\Psi_m$ measurement, respectively." (Page 8 Lines 200-212)

Taken plant stem samples as an example: For *H. rhamnoides* in pure plantation: 1 stem for each individual in each plot×3 individual in each plot×3 plots for this plantation ×5 rainfall events= 45 stems. Thus, the total plant stems were: 45 for *H. rhamnoides* in pure plantation+45 for *H. rhamnoides* in mixed plantation+45 for *P. tomentosa* in pure plantation+45 for *P. tomentosa* in mixed plantation=180 plant stems.

**2)** The complementary between RUP and MixSIR method have been added in the revised manuscript in "2.6.2 Calculation of RRS uptake proportion and water sources from different soil layers" subsection in "**2 Materials and methods**" section as follows: "In addition to RUP, the water uptake proportions from different soil layers were calculated on the first day after a rainfall event using the MixSIR program, to complement the analysis of plant water source variations in response to rainfall pulses. The RUP method only calculated the proportion of recent rainwater in the plant stem and did not include soil

water before the recent rainfall event (Gebauer and Ehleringer, 2000; Cheng et al., 2006). The water taken up from different soil layers by the plant is a mixture of soil water before the recent rainfall event and the recent rainwater." (Page 11 Lines 270-275).

The RUP is the core analysis in the present study, which is calculated as the proportion of rainwater in plant stem (Gebauer and Ehleringer, 2000; Cheng et al., 2006), and indicated the uptake of rainwater-recharged soil water proportion (RUP) by plant from a recent rainfall event. The water uptake proportion from different soil layers is calculated through MixSIR method, and is used to complement analysis the plant water sources variation after rainfall pulses for *H. rhamnoides* or *P. tomentosa*. The difference of these two methods can be observed in **Figure explain 1** as follows. The soil water after a recent rainfall event is the mixture of soil water before recent rainfall event and recent rainwater (**Figure explain 1**). The basic difference between RUP and results from MixSIR method is the former method only calculated the proportion of rainwater (rainwater-recharged soil water) in plant stem after a recent rainfall event and not including soil water before rainfall event.

[Figure]

**Plant water source from 3 soil layers: plant uptake of soil water after rainfall event through MixSIR method.**
It is calculated as the proportion of soil water from 3 soil layers in plant stem. In this method, no "rainwater" term is considered, all the water source is from soil, and rainwater is integrated into soil water.

**Figure explain 1** The schematic of plant uptake of rainwater-recharged soil water from a recent rainfall

event and plant water source from 3 soil layers after rainfall event.

In the present study, the RUP and water sources from 3 soil layers was also used to illustrate the mixed afforestation effect of these two species on plant water sources after rainfall pulses. The results indicated that mixed afforestation significantly increased the RUP for these two plant species, and significantly decreased the water uptake proportion from deep soil layer ($P < 0.05$).

The RUP and water sources from 3 soil layers were used to illustrate the two types of plant adaptation to cope with resource competition: (1) increase competition to rainwater-recharged soil water (RRS) from a recent rainfall event; and (2) minimize competition to deeps soil water, which is excessive consumption by plant and rainwater is the only replenish source to deep soil water.

The detailed discussion of the first adaptation types can be observed in "4.2 RRS uptake enhances plant transpiration for coexisting species in mixed plantation" subsection in "**4 Discussion**" section as follows: "Generally, two types of adaptation can be adopted by plants to cope with resource competition: increased competition ability or minimized competition interactions (West et al., 2007). Consistent with the first adaptation type, mixed afforestation enhanced the RUP for *H. rhamnoides* and *P. tomentosa* (Figs. 3 and 7, Table S4)." (Page 22 Lines 485-488).

The detailed discussion of the second adaptation types can be observed in same subsection as follows: "Furthermore, similar to other studies in the Loess Plateau (Wang et al., 2020a; Wu et al., 2021), the deep soil layer generally exhibited lower SW than other soil layers in all plantation types in the present study (Fig. 1, Table S3). Jia et al. (2017) and Wang et al. (2020a) attributed the lower SW in deep soil layers to the imbalance between rainwater replenishment and plant uptake of water from this soil layer in the studied region. Silvertown et al. (2015) and Tang et al. (2019) suggested that coexisting plant species generally reduce water uptake from soil layers that exhibit low soil water content to avoid water source competition in these layers and maintain stable coexistence. In the present study, consistent with the second adaptation type, mixed afforestation significantly decreased the water uptake proportion from the deep soil layer for these species (Table S5)." (Page 23 Lines 504-512).

Thus, in the present study, the plant water sources from different soil layers based on MixSIR method

is the complement analysis for RUP calculation. These two methods together were used to discuss the two types of plant adaptation to cope with resource competition.

**3)** The detailed method of how to calculate water uptake proportions from different soil layers through MixSIR method has been rewritten in "2.6.2 Calculation of RRS uptake proportion and water sources from different soil layers" subsection in "**2 Materials and methods**" section as follows: "Firstly, the seven soil depths (0–10, 10–20, 20–30, 30–50, 50–100, 100–150, and 150–200 cm) were combined into three soil layers (shallow, middle, and deep) based on the variation of soil water $\delta^{18}O$ and $\delta D$ and SW, to facilitate water source comparison (Wang et al., 2020a; Zhao et al., 2021). The shallow (0–30 cm) soil layer was vulnerable to rainfall, which exhibited high soil water $\delta^{18}O$ and $\delta D$ values and large water isotope and SW variations (Table S3, Fig. S3). The middle (30–100 cm) soil layer was less vulnerable to rainfall, with moderate soil water isotope values and water isotope and SW variations. The deep (100–200 cm) soil layer was relative stable, with lower soil water isotope values and smaller water isotope and SW variations compared with shallow and middle soil layers. In addition, based on one-way ANOVA followed by post hoc Tukey's test, significant difference ($P <$ 0.05) was observed in soil water $\delta^{18}O$ and $\delta D$ among three soil layers in each plot. Then, the water uptake proportions from three soil layers were calculated using the MixSIR program (Moore and Semmens, 2008), with model input parameters being the average $\delta^{18}O$ and $\delta D$ values in plant stem water and soil water at each soil layer in each plot." (Pages 11 Lines 276-288)

In the present study, seven soil depths were combined into three soil layers to facilitate water source comparison. These soil depths were combined based on the variation of soil water $\delta^{18}O$ and $\delta D$ and SW. The similar soil depths combination can also be observed in previous studies in the semiarid Loess Plateau (such as Wang et al., 2020a; Zhao et al., 2021), to facilitate water source comparison. In the present study, the shallow (0–30 cm) soil layer was vulnerable to rainfall, which exhibited high soil water $\delta^{18}O$ and $\delta D$ values and large water isotope and SW variations. The middle (30–100 cm) soil layer was less vulnerable to rainfall, with moderate soil water isotope values and water isotope and SW variations. The deep (100–200 cm) soil layer was relative stable, with lower soil water isotope values

and smaller water isotope and SW variations compared with shallow and middle soil layers.

Then, the water uptake proportions from three soil layers were calculated using the MixSIR program (Moore and Semmens, 2008), with model input parameters being the average $\delta^{18}O$ and $\delta D$ values in plant stem water and soil water at each soil layer in each plot.

**Table S3** The average (mean $\pm$ SD) and coefficients of variation (CVs, SD/mean) of soil water $\delta^{18}O$ and $\delta D$ on the first day after 5 selected rainfall events, and daily soil water content (SW) from DOY 132 to 273 (11 May to 30 September) in *H. rhamnoides* pure plantation, *P. tomentosa* pure plantation, and *H. rhamnoides–P. tomentosa* mixed plantation.

| | Soil depth | soil water $\delta^{18}O$ (‰) | | soil water $\delta D$ (‰) | | SW (m³ m⁻³) | |
|---|---|---|---|---|---|---|---|
| | | average | CV | average | CV | average | CV |
| *H. rhamnoides* pure plantation | 0–30 cm | -5.61±1.57 | 27.99 | -41.53±11.68 | 28.12 | 0.11±0.02 | 18.7 |
| | 30–100 cm | -7.14±0.92 | 12.89 | -52.37±6.47 | 12.35 | 0.1±0.01 | 10.34 |
| | 100–200 cm | -9.3±0.69 | 7.42 | -68.66±3.53 | 5.14 | 0.088±0.005 | 5.68 |
| *P. tomentosa* pure plantation | 0–30 cm | -5.43±1.69 | 31.12 | -42.08±11.91 | 28.3 | 0.12±0.02 | 16.67 |
| | 30–100 cm | -7.49±0.73 | 9.75 | -51.34±4.56 | 8.88 | 0.09±0.009 | 10.03 |
| | 100–200 cm | -9.39±0.34 | 3.62 | -67.36±3.79 | 5.63 | 0.085±0.005 | 5.88 |
| Mixed plantation | 0–30 cm | -5.68±1.73 | 30.46 | -41.67±10.67 | 25.61 | 0.11±0.019 | 17.28 |
| | 30–100 cm | -6.57±1.08 | 16.44 | -47.8±5.78 | 12.09 | 0.09±0.008 | 9.01 |
| | 100–200 cm | -9.07±0.5 | 5.51 | -64.47±2.45 | 3.8 | 0.089±0.005 | 5.62 |

There are 45, 30, and 30 data for calculated the average water $\delta^{18}O$ and $\delta D$ of shallow, middle, and deep soil layer in each plantation, respectively. The absolute value was used for CVs of soil water $\delta^{18}O$ and $\delta D$ calculation.

To be noticed, although the $\delta^{18}O$ and $\delta D$ values of soil water was more positive in middle (30-100 cm) than in shallow (0-30 cm) soil layer after 24 mm (**red cycle in Figure S7 as follows**), the average $\delta^{18}O$ and $\delta D$ after 5 rainfall events were more negative in middle than in shallow soil layer during the study

period (Table S3).

[Figure]

**Figure S7.** Variation in average (mean ± SD) δ[18]O and δD of rainwater, stem water, and soil water at seven soil depths for *H. rhamnoides* in (a–e) pure and (k–o) mixed plantations and for *P. tomentosa* in (f–j) pure and (k–o) mixed plantations after five rainfall events. Error bars indicate the standard deviation (n = 3). The date of each five selected rainfall events is followed the corresponding rainfall amount value. The average rainwater δ[18]O and δD for each rainfall event is calculated with 3 rainwater subsamples, which was divided from one rainwater sample.

[Figure]

**Figure S1.** The geographic location of (a) study area and (b) plantation sites in the Loess Plateau of China, and (c) monthly average (mean ±SD) rainfall amount and air temperature (Ta) during 2000-2017, and monthly rainfall amount and average Ta in 2018. Plantation types including *H. rhamnoides* pure plantation, *P. tomentosa* pure plantation, and Mixed plantation. Three adjacent plots were selected (16 m ×10 m) for each plantation type, and the schematic diagram of these plantation types is in (b). The China basic map can be obtained from

http://map.geoq.cn/arcgis/rest/services/ChinaOnlineCommunityENG/MapServer.

The size of each three adjacent plots (16 m ×10 m) is the schematic diagram in (b) in Fig. S1. The plantation areas of these types are larger than these selected plots in Fig S1. The plantation areas can be observed in different cycles as follows:

[Figure]

**2)** According to plantation slope, land-use history, and soil and water conservation measures, the relative sentences have been added in "2.1 Study site" subsection in "**2 Materials and methods**" as follows: "Three adjacent plantations were chosen for the study: pure *H. rhamnoides* plantation, pure *P. tomentosa* plantation, and *H. rhamnoides–P. tomentosa* mixed plantation (Fig. S1), with corresponding plantation slope of 5.2, 4.5, and 5.5°. All plantations were planted on abandoned grassland in 2004, where *Bothriochloa ischaemum* was the dominant herbaceous species at that time. Three adjacent plots were selected (16 m × 10 m) for each plantation type, and no soil and water conservation measure was conducted in the plantations." (Page 5 Lines 115-120)

**3)** According to soil physical properties, the relative sentences have been added in "2.1 Study site" subsection in "**2 Materials and methods**" section as follows: "Based on an experiment conducted in July 2018 using the cutting ring (Wu et al., 2016) and constant water head (Reynolds et al., 2002) method, the soil bulk density, total porosity, and saturated hydraulic conductivity at 0–50 cm soil depth were similar in three plantations. The average soil bulk density was 1.34 ±0.04, 1.31 ±0.05, and 1.31 ±

0.05 g cm$^{-3}$ for pure *H. rhamnoides*, pure *P. tomentosa*, and mixed plantations, respectively, and corresponding soil total porosity was 48.25 $\pm$ 0.52, 48.17 $\pm$ 0.48, and 48.03 $\pm$ 0.63%. The average soil saturated hydraulic conductivity was 0.51 $\pm$ 0.15, 0.54 $\pm$ 0.13, and 0.53 $\pm$ 0.11 mm min$^{-1}$ for pure *H. rhamnoides*, pure *P. tomentosa*, and mixed plantations, respectively. The soil is characterized as a silt loam soil according to United States Department of Agriculture soil taxonomy, with average sand (2–0.05 mm), silt (0.05–0.002 mm), and clay (<0.002 mm) compositions were 24.7 $\pm$ 1.6, 62.7 $\pm$ 0.8, and 12.6 $\pm$ 1.8%, respectively, for three plantation types at 0–50 cm soil depth. These compositions were determined using a Mastersize 2000 (Malvern Instruments Ltd., UK)." (Pages 5-6 Lines 130-141).

The method to determine the soil texture can be observed in red arrows in **Figure explain 2** as follows:

[Figure]

**Figure explain 2** The soil texture criterion based on the percent of sand (2-0.05 mm), silt (0.05-0.002 mm), and clay (<0.002 mm) in United States Department of Agriculture soil taxonomy. The red arrow lines represent the percent of sand, silt, and clay of our studied soil.

[Figure]

**Figure 1.** Variation in (a) rainfall amount, reference evapotranspiration ($ET_0$), and average (mean $\pm$ SD) soil water content (SW) in (b) *H. rhamnoides* pure plantation, (c) *P. tomentosa* pure plantation, and (d) mixed plantation from DOY 132 to 273 (11 May to 30 September) (n = 3). Standard deviation bars for SW at each soil layers are not shown to allow clear display of variation of SW for each plantation. Arrows in (a) indicate dates of sample collection at the first day after rainfall events: DOY 157 (6 June), DOY 194 (12 July), DOY 204 (23 July), DOY 249 (6 September), and DOY 265 (22 September).

**Secondly,** combined with the first adaptation type discussed in the previous paragraph (Page 22 Lines 486-495), we suggested that "Thus, both increased rainwater-recharged soil water uptake and decreased water source competition from the deep soil layer were adopted by these species in the mixed plantation to minimize water sources competition under water limited conditions." (Page 23 Lines 512-514)

**Response: Suggestions accepted.** In response to this key point, one figure including the local meteoric water line (LMWL) and the regression between $\delta^{18}O$ and $\delta D$ at three (0-30, 30-100, and 100-200 cm) soil layers has been added into the revised manuscript (see Figure S3). Meanwhile, more information has been given in "2.6 Statistical analysis" subsection in "**2 Materials and methods**" section as follows: "The shallow (0–30 cm) soil layer was vulnerable to rainfall, which exhibited high soil water $\delta^{18}O$ and $\delta D$ values and large water isotope and SW variations (Table S3, Fig. S3). The middle (30–100 cm) soil layer was less vulnerable to rainfall, with moderate soil water isotope values and water isotope and SW variations. The deep (100–200 cm) soil layer was relative stable, with lower soil water isotope values and smaller water isotope and SW variations compared with shallow and middle soil layers." (Page 11 Lines 278-283).

[Figure]

**Figure S3.** The linear regression relationship between $\delta^{18}O$ and $\delta D$ for soil water at three soil layers (0–30, 30–100, and 100–200cm) in (a) *H. rhamnoides* pure plantation, (b) *P. tomentosa* pure plantation, and (c) Mixed plantation. The local meteoric water line (LMWL) is plotted in each panel for reference.

The slope of LMWL is 7.29 based on 19 collected rainfall (**Figure explain 3**), which is slightly lower than the global meter water line (the slope is 8.0), but close to the LMWL (the slope ranges from 7.47 to 7.76) results near the present study region in the semiarid Loess Plateau (Figure **explains 4 and 5**). The slopes between $\delta^{18}O$ and $\delta D$ for three soil layers were smaller than LMWL and slightly decreased with soil layer increased.

The slightly lower slope for each of three soil layer compared with LMWL, may mainly attribute to the mixture of soil water before rainfall event and rainwater, and the soil water before rainfall generally undergo evaporation. In addition, these rainwater samples were collected immediately at the end of each rainfall event to avoid the possible evaporation. More information has been given in "2.4 Rainwater,

plant stem, soil water, and leaf sample collection and measurement" subsection in "**2 Materials and methods**" section as follows: "From April to October 2018, at the end of each rainfall event, 19 rainwater samples were collected immediately using a polyethylene rain gauge cylinder placed in the weather station, and stored at 4 ℃." (Page 7 Lines 185-186).

[Figure]

**Figure explain 3** Linear regression between $\delta^2H$ and $\delta^{18}O$ in rainwater. There are 19 collected rainwater samples and 5 selected rainwater samples in the present study. The rainfall amount of each of 5 selected rainwater samples was added.

[Figure]

**Figure explain 4** Linear regression between $\delta^2H$ and $\delta^{18}O$ in soil water for an (a) apple orchard and (b) black locust forest. LMWL represents the local meteoric water line (y = 7.47x + 3.29, $R^2$ = 0.95, P < 0.01). The isotopic composition values of xylem water from both species are also shown. This plot is from Wu et al. (2016).

[Figure]

**Figure explain 5** The linear regression relationship between $\delta^2H$ and $\delta^{18}O$ in soil water from three species (a) S. bungeana, (b) A. gmelinii, (c) V. negundo during the sampling periods. SWL represents soil water line based on isotopic data of soil water. LMWL represents the local meteoric water line (y = 7.76x + 5.14, $R^2$ = 0.91, p < 0.01). This plot is from Wang et al. (2017).

[Figure]

**Figure S1.** The geographic location of (a) study area and (b) plantation sites in the Loess Plateau of China, and (c) monthly average (mean ± SD) rainfall amount and air temperature (Ta) during 2000-2017, and monthly rainfall amount and average Ta in 2018. Plantation types including *H. rhamnoides* pure plantation, *P. tomentosa* pure plantation, and Mixed plantation. Three adjacent plots were selected (16 m × 10 m) for each plantation type, and the schematic diagram of these plantation types is in (b). The China basic map can be obtained from

http://map.geoq.cn/arcgis/rest/services/ChinaOnlineCommunityENG/MapServer.

*2. When Describing the soil texture (L.105) please indicate it is the texture you refer to and add the correct source (USDA). Besides the soil texture, kindly provide the soil class.*

**Response: Corrected and Added.** In response to this suggestion, the soil texture, United States Department of Agriculture soil taxonomy, and the soil compositions have been written in "2.1 Study site" subsection in "**2 Materials and methods**" section as follows: "The soil is characterized as a silt loam soil according to United States Department of Agriculture soil taxonomy, with average sand (2–0.05 mm), silt (0.05–0.002 mm), and clay (<0.002 mm) compositions were 24.7 ± 1.6, 62.7 ± 0.8, and 12.6 ± 1.8%, respectively, for three plantation types at 0–50 cm soil depth. These compositions were

determined using a Mastersize 2000 (Malvern Instruments Ltd., UK)." (Page 6 Lines 137-141).

Because the "United States Department of Agriculture" occurred only once in the present study, thus the abbreviation named "USDA" was not used. The method to determine the soil texture can be observed in red arrows in **Figure explain 2** as follows:

[Figure]

**Figure explain 2** The soil texture criterion based on the percent of sand (2-0.05 mm), silt (0.05-0.002 mm), and clay (<0.002 mm). The red arrow lines represent the percent of sand, silt, and clay of our studied soil.

*3. L.131: Explain what VPD stands for after equation 1 and give its units (as you have done for the other variable sin equation 1).*

**Response: Added.** In response to this suggestion, the VPD explanation has been added in the revised manuscript in "2.2 Environmental parameter measurements and $ET_0$ calculation" subsection in "**2 Materials and methods**" section as follows: "where $\gamma$, $s$, and *VPD* are the psychrometric constant (kPa $K^{-1}$), the slope between saturation vapor pressure and air temperature (kPa $K^{-1}$), and vapor pressure deficit (kPa), respectively." (Page 6 Lines 156-157)

*4. L.145: What does the abbreviation TDPs mean?*

**Response: Added.** In response to this suggestion, the TDPs has been added at its first appearance sentence in "2.3 Sap flow observation" subsection in "**2 Materials and methods**" section as follows:

"The sap flow was monitored by a pair of Granier-type thermal dissipation probes (TDPs) 10 mm in length and 2 mm in diameter in 36 selected individuals." (Page 7 Lines 164-166).

**Response: Corrected.** Thanks for this suggestion. It should be "RUP"but not "PAP", and this equation has been corrected in "2.6.2 Calculation of RRS uptake proportion and water sources from different soil layers" subsection in "**2 Materials and methods**" section as follows:

"The RRS uptake proportion (RUP, %) after a recent rainfall pulse for plant was calculated as the proportion of rainwater in plant stem as follows (Cheng et al., 2006):

$$\delta^{18}O\ (D)_P = RUP \times \delta^{18}O\ (D)_{rain} + (1 - RUP) \times \delta^{18}O\ (D)_{swb} \tag{5}$$

$$RUP = (\delta^{18}O(D)_p - \delta^{18}O(D)_{swb})/(\delta^{18}O(D)_{swa} - \delta^{18}O(D)_{swb}) \times 100\% \tag{6}$$

" (Page 10 Lines 252-255)

*6. L.208-209: Kindly provide a reference that supports this assumption ('no fractionation was considered during water source uptake by plant roots')*

**Response: Suggestions accepted.** In response to this suggestion, the sentence has been rewritten in "2.6.2 Calculation of RRS uptake proportion and water sources from different soil layers" subsection in "**2 Materials and methods**" section as follows: "No fractionation was considered during water source uptake by these plant roots because none of the plants exhibited xerophytic or halophytic characteristics. Ellsworth and Williams (2007) and Moore and Semmens (2008) suggested that a water stable isotope fractionation generally occurred during root uptake by xerophytic or halophytic plants." (Page 11 Lines 289-292).

[Figure]

**Figure 1.** Variation in (a) rainfall amount, reference evapotranspiration (ET$_0$), and average (mean ±SD) soil water content (SW) in (b) *H. rhamnoides* pure plantation, (c) *P. tomentosa* pure plantation, and (d) mixed plantation from DOY 132 to 273 (11 May to 30 September) (n = 3). Standard deviation bars for SW at each soil layers are not shown to allow clear display of variation of SW for each plantation. Arrows in (a) indicate dates of sample collection at the first day after rainfall events: DOY 157 (6 June), DOY 194 (12 July), DOY 204 (23 July), DOY 249 (6 September), and DOY 265 (22 September).

In addition, the first selected rainfall amount is 35.2 mm, occurred in DOY 155-156, can be observed in the red box in **Figure 1** as follows. The reason that we choose the rainfall amount during DOY 155-156, is mainly because the 35.2mm was continues occurred during these two successive days. DOY 155 only received 5.6 mm rainfall from 23:00-24:00, and DOY156 received 29.6mm. Therefore, the rainfall amount (5.6mm+29.6mm=35.2mm) received during DOY 155-156 was considered as one rainfall event in our present study. The half-hourly rainfall distribution during DOY 155-156 can be observed in **Figure explain 6** as follows:

[Figure]

**Figure 1.** Variation in (a) rainfall amount, reference evapotranspiration ($ET_0$), and average (mean $\pm$ SD) soil water content (SW) in (b) *H. rhamnoides* pure plantation, (c) *P. tomentosa* pure plantation, and (d) mixed plantation from DOY 132 to 273 (11 May to 30 September) (n = 3). Standard deviation bars for SW at each soil layers are not shown to allow clear display of variation of SW for each plantation. Arrows in (a) indicate dates of sample collection at the first day after rainfall events: DOY 157 (6 June), DOY 194 (12 July), DOY 204 (23 July), DOY 249 (6 September), and DOY 265 (22 September).

[Figure]

**Figure explain 6** The half-hour rainfall amount during DOY155-156. The rainfall during DOY 155-156 was considered as one rainfall event in the present study.

*8. Figure 2: It would be good to show the precipitation bars in this plot too.*

**Response: Suggestion accepted.** Both the rainfall amount and arrows indicate dates of sample collection have been added in the revised Figure 2 (a) in the revised manuscript as follows (Page 14 Lines 351-355):

[Figure]

**Figure 2.** Variation in (a) rainfall amount, and average daily normalized $F_d$ for *H. rhamnoides* in (a) pure and (b) mixed plantations and for *P. tomentosa* in (c) pure and (d) mixed plantations (n = 3). Arrows in (a) indicate dates of sample collection at the first day after rainfall events: DOY 157 (6 June), DOY 194 (12 July), DOY 204 (23 July), DOY 249 (6 September), and DOY 265 (22 September).

*9. L.321: in figure S5, kindly add the date of rainfall events. Moreover, the dD signature of rainwater for the 3.4 and 7.9 mm events is very enriched. Could it be that there has been some evaporation of the sample going on? In any case, as I mentioned earlier, it would be really good if the authors could provide a plot of the LMWL.*

**Response: Added and Clarified.** In response to this key point, the date of rainfall events, one figure including the local meteoric water line (LMWL) and the regression between $\delta^{18}$O and $\delta$D at three (0-30, 30-100, and 100-200 cm) soil layers, the positive rainwater $\delta^{18}$O and $\delta$D values in small rainfall events explanation, and the possible rainwater evaporation discussion have been added in the revised manuscript and clarified.

   **1)** The date of five selected rainfall events have been added in Figure S7 in the revised manuscript as follows:

[Figure]

**Figure S7.** Variation in average (mean ± SD) $\delta^{18}$O and $\delta$D of rainwater, stem water, and soil water at

seven soil depths for *H. rhamnoides* in (a–e) pure and (k–o) mixed plantations and for *P. tomentosa* in

(f–j) pure and (k–o) mixed plantations after five rainfall events. Error bars indicate the standard

deviation (n = 3). The date of each five selected rainfall events is followed the corresponding rainfall

amount value. The average rainwater $\delta^{18}$O and $\delta$D for each rainfall event is calculated with 3 rainwater

subsamples, which was divided from one rainwater sample.

**2)** One plot including the local meteoric water line (LMWL) and the regression between $\delta^{18}$O and $\delta$D

at three different soil layers (0-30, 30-100, and 100-200cm) has been added in **Figure S3** as follows.

[Figure]

**Figure S3.** The linear regression relationship between $\delta^{18}O$ and $\delta D$ for soil water at three soil layers (0–30, 30–100, and 100–200cm) in (a) *H. rhamnoides* pure plantation, (b) *P. tomentosa* pure plantation, and (c) Mixed plantation. The local meteoric water line (LMWL) is plotted in each panel for reference.

**3)** In response to positive rainwater $\delta^{18}O$ and $\delta D$ values in small rainfall events, the relative sentences have been added in in "4.1 RRS uptake enhances plant transpiration for *H. rhamnoides* but not *P. tomentosa* in pure plantations" subsection in "**4 Discussion**" section as follows: "Furthermore, the $\delta^{18}O$ and $\delta D$ values in small rainfall events generally exhibit more positive values than those in large rainfall events (Fig. S7). Salamalikis et al. (2016) attribute this phenomenon to the sub-cloud evaporation effect in dry conditions where rainwater in small rainfall event is more vulnerable subject to evaporation during their descent process compared in large rainfall event." (Pages 18-19 Lines 417-421)

Salamalikis et al. (2016) suggested that the positive $\delta^{18}O$ and $\delta D$ values for small rainfall amount than those for large rainfall amount may mainly attribute to the sub-cloud evaporation effect in rainfall

in dry conditions (**Figure explain 7**). In dry conditions, rainwater in small rainfall event is more vulnerable subject to evaporation during their descent process compared in large rainfall event (**Figure explain 7**). In the present study, the 3.4 and 7.9 mm occurred with high $ET_0$ period, which reflects the dry condition (**red cycles in Figure 1 as follows**), although the SW during 7.9 mm rainfall event in different soil layers were higher than those SW during 3.4 mm rainfall event.

[Figure]

**Figure explain 7** Schematic of sub-cloud evaporation effect on rainwater. This plot is from Salamalikis et al. (2016).

[Figure]

**Figure 1.** Variation in (a) rainfall amount, reference evapotranspiration ($ET_0$), and average (mean $\pm$ SD) soil water content (SW) in (b) *H. rhamnoides* pure plantation, (c) *P. tomentosa* pure plantation, and (d) mixed plantation from DOY 132 to 273 (11 May to 30 September) (n = 3). Standard deviation bars for SW at each soil layers are not shown to allow clear display of variation of SW for each plantation. Arrows in (a) indicate dates of sample collection at the first day after rainfall events: DOY 157 (6 June),

DOY 194 (12 July), DOY 204 (23 July), DOY 249 (6 September), and DOY 265 (22 September).

**4**) In response to the possible rainwater evaporation, one plot including the regression between $\delta^{18}O$ and $\delta D$ for 19 collected rainwater samples has been added in **Figure explain 3** as follows. No rainwater would be evaporated during these rainwater samples collection as showed in **Figure explain 3**. The $\delta^{18}O$ and $\delta D$ values of five selected rainwater were well matched the local meteoric water line (LMWL) (**Figure explain 3**). In addition, these rainwater samples were collected immediately at the end of each rainfall event to avoid the possible evaporation. More information has been given in "2.4 Rainwater, plant stem, soil water, and leaf sample collection and measurement" subsection in "**2 Materials and methods**" section as follows: "From April to October 2018, at the end of each rainfall event, 19 rainwater samples were collected immediately using a polyethylene rain gauge cylinder placed in the weather station, and stored at 4 ℃." (Page 7 Lines 185-187).

[Figure]

**Figure explain 3** Linear regression between $\delta^2H$ and $\delta^{18}O$ in rainwater. There are 19 collected rainwater samples and 5 selected rainwater samples in the present study. The rainfall amount of each of 5 selected rainwater samples was added.

The slope of LMWL is 7.29 based on 19 collected rainfall (**Figure explain 3**), which is slightly lower than the global meter water line (the slope is 8.0), but close to the LMWL (the slope ranges from 7.47 to 7.76) results near the present study region in the semiarid Loess Plateau (Figure **explains 4 and 5**). The slopes between $\delta^{18}O$ and $\delta D$ for three soil layers were smaller than LMWL and slightly decreased with soil layer increased (**Figure S3**). The slightly lower slope for each of three soil layer compared with

LMWL, may mainly attribute to the mixture of soil water before rainfall event and rainwater, and the soil water before rainfall generally undergo evaporation.

[Figure]

**Figure S3.** The linear regression relationship between $\delta^{18}O$ and $\delta D$ for soil water at three soil layers (0–30, 30–100, and 100–200cm) in (a) *H. rhamnoides* pure plantation, (b) *P. tomentosa* pure plantation, and (c) Mixed plantation. The local meteoric water line (LMWL) is plotted in each panel for reference.

[Figure]

**Figure explain 4** Linear regression between $\delta^2H$ and $\delta^{18}O$ in soil water for an (a) apple orchard and (b) black locust forest. LMWL represents the local meteoric water line ($y = 7.47x + 3.29$, $R^2 = 0.95$, $P < 0.01$). The isotopic composition values of xylem water from both species are also shown. This plot is

from Wu et al. (2016).

[Figure]

**Figure explain 5** The linear regression relationship between δ²H and δ¹⁸O in soil water from three species (a) S. bungeana, (b) A. gmelinii, (c) V. negundo during the sampling periods. SWL represents soil water line based on isotopic data of soil water. LMWL represents the local meteoric water line (y = 7.76x + 5.14, R² = 0.91, p < 0.01). This plot is from Wang et al. (2017).

**DATE:** March 9, 2022

Compuscript Ltd
T/A International Science Editing
Bay K, Shannon Industrial Park West
Shannon, Co Clare
Ireland
Phone +353 61 472818   Fax +353 61 472688

To whom it may concern,

The paper "Differential response of plant transpiration to uptake of rainwater-recharged soil water for dominant tree species in the semiarid Loess Plateau" by Yakun Tang was edited by International Science Editing. We were asked not to edit the references. Please contact us if you would like to view the edited paper.

Kindest regards,

David Cushley.

If the English and our answers are not meet the standard, please give me another chance, I will revised the language again by another scientific editing service company again .

---

## Referee Report (RR1)

The revised manuscript answered my questions well. However, I have made additional suggestions below to enhance it.

1)    Lines 130-137 The soil bulk density, total porosity, and saturated hydraulic conductivity at 0–200 cm soil depth, rather than 0-50 cm, should be described. Because the similar parameter values were described at 0–200 cm soil depth in Table A1 in Appendix A1.

2) Lines 351-355 The X-axis of Figure 2 should be same as Figure 1, it should be ranged from DOY 132 to 273 (11 May to 30 September).

---

## Author Response (AR2)

Authors' responses to Editor and Reviewers comments on the manuscript of "Differential response of plant transpiration to uptake of rainwater-recharged soil water for dominant tree species in the semiarid Loess Plateau". Manuscript ID: hess-2021-351.

**Dear Editor,**

We deeply appreciate you for giving us an opportunity to revise our manuscript. There are two versions of revised manuscripts, the first version is updated manuscript and without track changes (Updated revised version), the second version is highlighted the changes by using the red colored text in the manuscript (track-changes version). Here are the point-to-point responses (responses in upright Roman) to the Editor and Reviewers comments (*original comment in Itali*).

**Editor**

*Thank you for the revised manuscript. The manuscript has significantly improved already. As you can see in the second round of reviews, there are still some revisions proposed by both reviewers. Indeed I think that it is possible with some minor further revisions to address the points raised by the reviewers in this second round. Therefore I have decided to ask you to do a final round of minor revisions responding to the points raised by the reviewers. Subsequently I will do a final review before the manuscript will be published.*

**Response: Suggestions accepted.** Thanks for these suggestions. The further revision has been made according to the suggestions of two **Reviewers.**

In response to **Reviewer 1,** the values of soil bulk density, total porosity, and saturated hydraulic conductivity at 0–200 cm soil depth have been recalculated and added in "2.1 Study site" subsection in "2 Materials and methods" section (Pages 5-6 Lines 134-141). Meanwhile, the Figure 3 has been revised in "3 Results" section (Page 15 Lines 362-367).

In response to **Reviewer 2, firstly,** the variation of net radiation ($R_n$) and vapor pressure deficit (VPD), and their influence on relative response of normalized sap flow ($SF_R$) have been added in the revised manuscript (Pages 12-13 Lines 320-321). **Secondly,** the possible anisohydric/isohydric behavior

for each species has been added and discussed in the revised manuscript (Pages 20-21 Lines 456-461; Page 23 Lines 512-516). **Thirdly**, the Figure 7 has been revised to clearly exhibit the linear correlation for each species in pure and mixed plantations (Page 19 Lines 416-419).

The detailed and some other revisions can be observed in response to **Comments of Reviewer _1)_ and _2)_** as follows.

**Furthermore**, the language of the revised manuscript has been has been refined by _International Science Editing_.

[Figure]

**International Science Editing Service Certification**

This document certifies that the manuscript titled **Differential response of plant transpiration to uptake of rainwater-recharged soil water for dominant tree species in the semiarid Loess Plateau** has been edited for English language usage, grammar, punctuation and spelling by one or more native English-speaking editors at International Science Editing. Using their scientific and editing backgrounds to highlight passages that were found to be confusing or lacking in clarity, our editors have focused on correcting the language and rephrasing sentences when required. Our editors endeavour to ensure that neither the original research content nor the authors' intentions have been altered in any way during the editing of the manuscript.

Manuscripts receiving this certification are ready for publication in English language journals. Please note that the author may accept or reject any of our suggestions and edits. The final edited version can be verified by visiting our verification page.

Any questions or concerns about the edited manuscript should be directed to International Science Editing at admin@internationalscienceediting.com.

> **Manuscript title: Differential response of plant transpiration to uptake of rainwater-recharged soil water for dominant tree species in the semiarid Loess Plateau**
> **Authors: Yakun Tang**
> **Key: 98139-GOVH1MZXER**
> **Created Date: Aug-30-2022**

This certificate may be verified at https://ise.compuscript.ie/ise/chinese/certificate_download.php

International Science Editing Service is a service provided by Compuscript Limited, one of the world's leading STEM language service companies. Compuscript has been a provider of high-quality editing since 1994. International Science Editing comprises more than 400 language editors from a wide range of academic backgrounds. All our language editors are native English speakers and meet strict selection criteria. All our editors have completed either a PhD or MD and each has undergone extensive training on editing academic papers.

Uploaded manuscripts are reviewed by an editor with the relevant academic background who is familiar with the subject matter involved; the edited paper is then quality assessed by a senior editor before being returned to the author, thus ensuring that our high standards are maintained.

International Science Editing is continually recruiting editors to represent growing and new disciplines of scientific enquiry so that we can meet the needs of those working in science and medicine today.

[Figure]

**Reviewer 1**

**Major Comments:**

*The revised manuscript answered my questions well. However, I have made additional suggestions below to enhance it.*

**Response: Suggestions accepted.** Thanks for these suggestions. The detailed responses to these two suggestions can be observed in response to *Minor Comments 1)* and *2)* as follows.

**Minor Comments:**

*1) Lines 130-137 The soil bulk density, total porosity, and saturated hydraulic conductivity at 0–200 cm soil depth, rather than 0-50 cm, should be described. Because the similar parameter values were described at 0–200 cm soil depth in Table A1 in Appendix A1.*

**Response: Recalculated and Revised.** In response to this meaningful and detailed suggestion, the soil bulk density, total porosity, and saturated hydraulic conductivity at 0–200 cm soil depth has been recalculated in the revised manuscript. The relative sentence has been rewritten in "2.1 Study site" subsection in "2 Materials and methods" section as follows: "Based on an experiment conducted in July 2018 using the cutting ring (Wu et al., 2016), constant water head (Reynolds et al., 2002), and centrifugation (Qiao et al., 2019) method, the soil bulk density, total porosity, saturated hydraulic conductivity, field capacity, and permanent wilting point at 0–200 cm soil depth were found to be similar in the three plantations. The average soil bulk density was 1.38 $\pm$ 0.08, 1.35 $\pm$ 0.11, and 1.35 $\pm$ 0.09 g cm$^{-3}$ for pure *H. rhamnoides*, pure *P. tomentosa*, and mixed plantations, respectively, and corresponding soil total porosity was 48.2 $\pm$ 0.6, 48.1 $\pm$ 0.4, and 48.1 $\pm$ 0.7%. The average soil saturated hydraulic conductivity was 0.44 $\pm$ 0.08, 0.46 $\pm$ 0.09, and 0.46 $\pm$ 0.08 mm min$^{-1}$ for pure *H. rhamnoides*, pure *P. tomentosa*, and mixed plantations, respectively. The average field capacity was 0.26 $\pm$ 0.02, 0.25 $\pm$ 0.03, and 0.25 $\pm$ 0.02 m$^3$ m$^{-3}$ for pure *H. rhamnoides*, pure *P. tomentosa*, and mixed plantations, respectively, and corresponding permanent wilting point was 0.06 $\pm$ 0.02, 0.06 $\pm$ 0.01, and 0.06 $\pm$ 0.02 m$^3$ m$^{-3}$." (Pages 5-6 Lines 134-144).

To be noticed, the "field capacity" and "permanent wilting point" parameters at 0–200 cm soil depth

have also been added in these sentences mentioned above according to the suggestion by **Reviewer** *2)*.

*2) Lines 351-355 The X-axis of Figure 2 should be same as Figure 1, it should be ranged from DOY 132 to 273 (11 May to 30 September).*

**Response: Added.** The X-axis of Figure 2 has been extended form DOY 132 to 273. The revised Figure 2 can be observed in "3.2 Variations in sap flow" subsection in "3 Results" section as follows:"

[Figure]

**Figure 3.** Variation in (a) rainfall amount, and average daily normalized $F_d$ for *H. rhamnoides* in (a) pure and (b) mixed plantations and for *P. tomentosa* in (c) pure and (d) mixed plantations from DOY 132 to 273 (11 May to 30 September) (n = 3). Arrows in (a) indicate dates of sample collection at the first day after rainfall events: DOY 157 (6 June), DOY 194 (12 July), DOY 204 (23 July), DOY 249 (6 September), and DOY 266 (23 September)." (Page 15 Lines 362-367)

**Reviewer 2**

**Major Comments:**

*The study on which the manuscript is based is well structured, with clear and important objectives and a vast array of methodologies to investigate them and support the results. I think the manuscript benefited in readability and quality from the previous reviews.*

*1) The first major comment I have to regard the choice to base all the considerations about the behavior of the two tree species on a few days after different rainfall amounts. If this is acceptable in dry periods when minor rainfall events occur after a relatively long period without rain and thus we can consider the trees under water stress conditions, which presupposes that there is an active physiological control on tree transpiration, the same may not be valid in presence of higher water availability. Looking at Fig. 1 it seems that after DOY 200 rainfall events are quite frequent and SWC keeps at relatively high values (at least up to 50 cm depth). This means that in this situation water availability may not be the most influential factor on SF, but a stronger role may be played by the environmental variables (determining the atmospheric demand for water). Although the effect of ETo was discussed with apparently no effect on SF (lines 455-463), I think it would be worth analyzing the SF response to key climatic variables (such as net radiation and VPD) on days with relatively good water availability, to have a clearer picture of the transpiration variability of the two tree species in the pure and mixed plantation.*

**Response: Suggestions accepted and Added.** Thanks for this meaningful and detailed suggestion, the variation of net radiation ($R_n$) and vapor pressure deficit (VPD), and their influence on relative response of normalized sap flow ($SF_R$) have been added in the revised manuscript.

   **Firstly**, the variation of $R_n$ and VPD has been added in the "3.1 Variation in environmental parameters and plant fine root vertical distribution" subsection in "**3 Results**" section as follows: "The $R_n$ and VPD also exhibited higher and lower values during the low and high rainfall event periods, respectively (Fig. S4)" (Pages 12-13 Lines 320-321).

[Figure]

**Figure S4.** Variation in rainfall amount, net radiation ($R_n$), and vapor pressure deficit (VPD) from DOY 132 to 273 (11 May to 30 September). Arrows indicate dates of sample collection at the first day after rainfall events: DOY 157 (6 June), DOY 194 (12 July), DOY 204 (23 July), DOY 249 (6 September), and DOY 266 (23 September).

The relative low (DOY 132–202) and high rainfall event (DOY 203–273) period has been mentioned before this sentence in "3.1 Variation in environmental parameters and plant fine root vertical distribution" subsection in "**3 Results**" section as follows: "The $ET_0$ (554.7 mm) was approximately twice the rainfall amount during the study period, with the higher and lower values during the low (DOY 132–202) and high (DOY 203–273) rainfall event periods, respectively (Fig. 1)." (Page 12 Lines 317-320).

**Secondly**, the influence of $R_n$ and VPD on $SF_R$ in pure plantation during the observation (DOY 132–273) and relative high rainfall event (DOY 203–273) periods have been added in the revised manuscript in "4.1 RRS uptake enhances plant transpiration for *H. rhamnoides* but not *P. tomentosa* in pure plantations" subsection in "**4 Discussion**" section as follows: "The $ET_0$ and VPD represent the atmospheric evaporative demand factors and $R_n$ represents the energy factor, and these factors have been observed to influence plant transpiration (Du et al., 2011; Iida et al., 2016; Li et al., 2021).

However, in the present study, none of $ET_0$, $R_n$, and VPD after rainfall or relative response of $ET_0$, $R_n$, and VPD significantly influenced $SF_R$ for either species in pure plantations (Table S7)." (Page 21 Lines 472-476).

**Thirdly**, the influence of $R_n$ and VPD on $SF_R$ in mixed plantation has been added in the revised manuscript in "4.2 RRS uptake enhances plant transpiration for coexisting species in mixed plantation" subsection in "**4 Discussion**" section as follows: "Furthermore, no significant relationship of $SF_R$ with $ET_0$, VPD, and $R_n$ after rainfall and of $SF_R$ with relative response of $ET_0$, VPD, and $R_n$ was observed for these species in the mixed plantation from DOY 132 to 273 and from DOY 203 to 273 (Table S7)." (Page 24 Lines 521-524)

**Table S7.** The linear regression relationship between relative response of normalized sap flow ($SF_R$) and reference evapotranspiration ($ET_0$), net radiation ($R_n$), and vapor pressure deficit (VPD) after rainfall, and between $SF_R$ and relative response of $ET_0$, $R_n$, and VPD from DOY 132 to 273 and from DOY 203 to 273.

| Period | Independent factors | *H. rhamnoides* in pure plantation | | *H. rhamnoides* in mixed plantation | | *P. tomentosa* in pure plantation | | *P. tomentosa* in mixed plantation | |
|---|---|---|---|---|---|---|---|---|---|
| | | $R^2$ | p | $R^2$ | p | $R^2$ | p | $R^2$ | p |
| | $ET_0$ | 0.18 | 0.47 | 0.11 | 0.59 | 0.44 | 0.22 | 0.39 | 0.26 |
| | VPD | 0.09 | 0.62 | 0.02 | 0.83 | 0.26 | 0.38 | 0.22 | 0.43 |
| **DOY** | $R_n$ | 0.06 | 0.68 | 0.04 | 0.74 | 0.04 | 0.75 | 0.03 | 0.8 |
| **132–273** | Relative response of $ET_0$ | 0.35 | 0.32 | 0.61 | 0.12 | 0.12 | 0.56 | 0.25 | 0.4 |
| | Relative response of VPD | 0.3 | 0.34 | 0.48 | 0.2 | 0.06 | 0.7 | 0.12 | 0.57 |
| | Relative response of $R_n$ | 0.08 | 0.74 | 0.02 | 0.84 | 0.1 | 0.61 | 0.07 | 0.66 |

| DOY | | | | | | | | | |
|---|---|---|---|---|---|---|---|---|---|
| | ET$_0$ | 0.15 | 0.75 | 0.25 | 0.67 | 0.009 | 0.98 | 0.003 | 0.97 |
| | VPD | 0.14 | 0.76 | 0.24 | 0.67 | 0.008 | 0.99 | 0.002 | 0.97 |
| **DOY** | R$_n$ | 0.31 | 0.63 | 0.44 | 0.54 | 0.04 | 0.87 | 0.06 | 0.84 |
| **203–273** | Relative response of ET$_0$ | 0.06 | 0.84 | 0.01 | 0.93 | 0.35 | 0.59 | 0.31 | 0.63 |
| | Relative response of VPD | 0.79 | 0.3 | 0.67 | 0.39 | 0.29 | 0.64 | 0.34 | 0.6 |
| | Relative response of R$_n$ | 0.03 | 0.9 | 0.09 | 0.81 | 0.05 | 0.86 | 0.03 | 0.89 |

The regression equation is y=ax+b for all equations. Relative responses of R$_n$, VPD, and ET$_0$ are respectively calculated as for SF$_R$ in Eq. (4), corresponding to before and the first day after rainfall event parameters for R$_n$, VPD, and ET$_0$.

There are three selected rainfall events (7.9, 15.4 and 24 mm) after DOY 203, when the SWC at 0-50 cm become at relatively high values. The non-significant influence of ET$_0$, R$_n$, and VPD after rainfall on SF$_R$, and non-significant influence of relative response of ET$_0$, R$_n$, and VPD on SF$_R$ were detected from DOY 132 to 273 and from DOY 203 to 273.

For example, the influence of ET$_0$ and relative response of ET$_0$ on SF$_R$ from DOY 203 to 273 can be observed in **Figure explain 1** as follows**.**

[Figure]

**Figure explain 1**. Relationship of (a, b) reference evapotranspiration (ET$_0$) and (c, d) relative response of ET$_0$ with relative response of normalized F$_d$ (SF$_R$) for *H. rhamnoides* and *P. tomentosa* in both

plantation types.

The influence of VPD and relative response of VPD on $SF_R$ from DOY 203 to 273 can be observed in **Figure explain 2** as follows.

[Figure]

**Figure explain 2**. Relationship of (a, b) vapor pressure deficit (VPD) and (c, d) relative response of VPD with relative response of normalized $F_d$ ($SF_R$) for *H. rhamnoides* and *P. tomentosa* in both plantation types.

The influence of $R_n$ and relative response of $R_n$ on $SF_R$ DOY 203 to 273can be observed in **Figure explain 3** as follows.

[Figure]

**Figure explain 3**. Relationship of (a, b) net radiation ($R_n$) and (c, d) relative response of VPD with relative response of normalized $F_d$ ($SF_R$) for *H. rhamnoides* and *P. tomentosa* in both plantation types.

**References:**

Du, S., Wang, Y. L., Kume, T., Zhang, J. G., Otsuki, K., Yamanaka, N., and Liu, G. B.: Sapflow characteristics and climatic responses in three forest species in the semiarid Loess Plateau region of China, Agr Forest Meteorol, 151, 1-10, 2011.

Iida, S., Shimizu, T., Tamai, K., Kabeya, N., Shimizu, A., Ito, E., Ohnuki, Y., Chann, S., and Keth, N.: Interrelationships among dry season leaf fall, leaf flush and transpiration: insights from sap flux measurements in a tropical dry deciduous forest, Ecohydrology, 9, 472-486, 10.1002/eco.1650, 2016.

Li, H. Q., Zhang, F. W., Zhu, J. B., Guo, X. W., Li, Y. K., Lin, L., Zhang, L. M., Yang, Y. S., Li, Y. N., Cao, G. M., Zhou, H. K., and Du, M. Y.: Precipitation rather than evapotranspiration determines the warm-season water supply in an alpine shrub and an alpine meadow, Agr Forest Meteorol, 300, ARTN 108318, 10.1016/j.agrformet.2021.108318, 2021.

*2) A second major comment regards the different sensitivity of H. rhamnoides and P. tomentosa with respect to rainfall pulses. Mention a possible anisohydric vs. isohydric behavior of the two species, respectively, I think will help to contextualize these findings with respect to the existing literature.*

**Response: Suggestions accepted and Added.** In response to this meaningful suggestion, the possible anisohydric/isohydric behavior for each species has been added and discussed in "**1 Introduction**", "**4 Discussion**", "**5 Conclusions**", and "**Abstract**" sections in the revised manuscript.

**Firstly**, the characteristics of anisohydric/isohydric plant behavior related to leaf $\Psi_{pd} - \Psi_m$ and plant transpiration has been added in the "**1 Introduction**" section as follows: "For example, plant species that show isohydric behavior generally maintain relative small $\Psi_{pd} - \Psi_m$ to protect stem hydraulic architecture, which is vulnerable to cavitation and limited plant transpiration under varied soil water conditions (Franks et al., 2007; McDowell et al., 2008). However, plant species that show anisohydric behavior are generally less vulnerable to cavitation and adopt relative large $\Psi_{pd} - \Psi_m$ to allow high plant transpiration after rainfall pulses (West et al., 2007; Klein, 2014; Ding et al., 2021)." (Page 3 Lines 61-66)

**Secondly**, the definition of anisohydric/isohydric behavior for each species (*H. rhamnoides* and *P.*

*tomentosa*) in pure plantation has been discussed and added in "4.1 RRS uptake enhances plant transpiration for *H. rhamnoides* but not *P. tomentosa* in pure plantations" subsection in "**4 Discussion**" section as follows: "Meanwhile, the $\Psi_{pd}-\Psi_m$ was significantly higher for *H. rhamnoides* (0.54 $\pm$ 0.26 MPa) compared to *P. tomentosa* (0.2 $\pm$ 0.06 MPa) (P<0.01), indicated that *H. rhamnoides* and *P. tomentosa* exhibited anisohydric and isohydric behavior, respectively, based on definitions of Franks et al. (2007) and Klein (2014). Previous studies demonstrated that isohydric plants generally exhibit more conservative transpiration than anisohydric plants when contending with varied soil water conditions (West et al., 2007; McDowell et al., 2008; Ding et al., 2021). The significantly higher (P < 0.001) $SF_R$ for *H. rhamnoides* (56.9 $\pm$ 43.9 %) than *P. tomentosa* (35.19 $\pm$ 26.9 %) indicated that plant transpiration for *H. rhamnoides* was more sensitive to rainfall pulses than *P. tomentosa*." (Pages 20-21 Lines 456-463)

Franks et al. (2007) and Klein (2014) suggested that isohydric behavior species generally exhibited small $\Psi_{pd}-\Psi_m$, meanwhile, anisohydric behavior species generally exhibited large $\Psi_{pd}-\Psi_m$. In the present study, significantly larger $\Psi_{pd}-\Psi_m$ (0.54 $\pm$ 0.26 MPa) was observed for *H. rhamnoides* compared to the value (0.2 $\pm$ 0.06 MPa) for *P. tomentosa* (P<0.01) (**Table explain 1**). Thus, *H. rhamnoides* and *P. tomentosa* cab be considered anisohydric and isohydric behavior plant species, respectively.

**Table explain 1.** The average (mean $\pm$ SD) and coefficients of variation (CVs, SD/mean) of gradient of leaf water potential ($\Psi_{pd}-\Psi_m$) and normalized sap flow ($SF_R$) in *H. rhamnoides* pure plantation, *P. tomentosa* pure plantation, and *H. rhamnoides–P. tomentosa* mixed plantation.

| | $\Psi_{pd}-\Psi_m$ | | | $SF_R$ | |
| --- | --- | --- | --- | --- | --- |
| | Average (MPa) | CV (%) | *p* | Average (%) | *p* |
| *H. rhamnoides* in pure plantation | 0.54 $\pm$ 0.26 | 48.2 | | 56.9 $\pm$ 43.9 | |
| | | | <0.01 | | <0.001 |
| *P. tomentosa* in pure plantation | 0.2 $\pm$ 0.06 | 30 | | 35.1 $\pm$ 26.9 | |

| | | | | | |
|---|---|---|---|---|---|
| *H. rhamnoides* in mixed plantation | $0.72 \pm 0.32$ | 44.4 | | $89.2 \pm 80.2$ | |
| | | | <0.01 | | <0.001 |
| *P. tomentosa* in mixed plantation | $0.39 \pm 0.09$ | 23.6 | | $50.7 \pm 38.1$ | |

The *p* value is the significant detect result for these two plantation species in pure and mixed plantations, respectively. These results were contained in Tables S4 and S6 in the manuscript.

In addition, this study mainly focused on the response of plant transpiration to uptake of rainwater-recharged soil water after 5 rainfall events. West et al. (2007) and McDowell et al. (2008) also suggested that isohydric plant generally exhibited more conservative transpiration than anisohydric plant to contend with varied soil water conditions. Meanwhile, the $SF_R$ value for *H. rhamnoides* (anisohydric behavior species) was significantly higher than that for *P. tomentosa* (isohydric behavior species) (P < 0.001). Thus, we suggested that the *H. rhamnoides* was more sensitive to rainfall pulses than *P. tomentosa* in pure plantation based on significant higher $SF_R$ value for former compared with latter plant species (P < 0.001).

**Thirdly**, the anisohydric/isohydric behavior for each species in mixed plantation has been discussed and added in "4.2 RRS uptake enhances plant transpiration for coexisting species in mixed plantation" subsection in "**4 Discussion**" section as follows: "Similar to the results in pure plantations, the significant higher $\Psi_{pd}-\Psi_m$ ($0.72 \pm 0.32$ MPa) and $SF_R$ ($89.2 \pm 80.2\%$) for *H. rhamnoides* compared to *P. tomentosa* ($0.39 \pm 0.09$ MPa and $50.7 \pm 38.1\%$, respectively) in mixed plantation (Figs. 3 and 6), suggested that *H. rhamnoides* and *P. tomentosa* exhibited anisohydric and isohydric behavior in mixed plantation, respectively, and the former plant species was more sensitive to rainfall pulses than *P. tomentosa*." (Page 23 Lines 512-516).

**Fourthly**, these discussions and results have been added in the "**5 Conclusions**" section as follows: "In pure and mixed plantations, the large $\Psi_{pd}-\Psi_m$ was consistent with high $SF_R$ for *H. rhamnoides*

suggesting that this species exhibited anisohydric behavior and sensitivity to rainfall pulses. Meanwhile, the small $\Psi_{pd}-\Psi_m$ was consistent with low $SF_R$ for *P. tomentosa* in both plantation types, and indicated that this species exhibited isohydirc behavior and less sensitivity to rainfall pulses." (Page 25 Lines 563-567).

**Finally**, we summarized these results and discussions, the relative sentences have been added in the "**Abstract**" section as follows: "In pure and mixed plantations, the large $\Psi_{pd}-\Psi_m$ was consistent with high $SF_R$ for *H. rhamnoides*, and the small $\Psi_{pd}-\Psi_m$ was consistent with low $SF_R$ for *P. tomentosa*, in response to rainfall pulses. Therefore, *H. rhamnoides* and *P. tomentosa* exhibited anisohydric and isohydric behavior, respectively, and the former plant species was more sensitive to rainfall pulses than *P. tomentosa*." (Page 1 Lines 23-26).

**References:**

Ding, Y. L., Nie, Y. P., Chen, H. S., Wang, K. L., and Querejeta, J. I.: Water uptake depth is coordinated with leaf water potential, water-use efficiency and drought vulnerability in karst vegetation, New Phytol, 229, 1339-1353, 2021.

Franks, P. J., Drake, P. L., and Froend, R. H.: Anisohydric but isohydrodynamic: Seasonally constant plant water potential gradient explained by a stomatal control mechanism incorporating variable plant hydraulic conductance, Plant, Cell and Environment, 30, 19–30, 2007.

Klein, T.: The variability of stomatal sensitivity to leaf water potential across tree species indicates a continuum between isohydric and anisohydric behaviours, Funct Ecol, 28, 1313–1320, 2014.

McDowell, N. G., Pockman, W. T., Allen, C. D., Breshears, D. D., Cobb, N., Kolb, T., Plaut, J., Sperry, J., West, A., and Williams, D. G.: Mechanisms of plant survival and mortality during drought: why do some plants survive while others succumb to drought? New Phytol, 178, 719– 739, 10.1111/nph.16971, 2008.

West, A. G., Hultine, K. R., Jackson, T. L., and Ehleringer, J. R.: Differential summer water use by Pinus edulis and Juniperus osteosperma reflects contrasting hydraulic characteristics, Tree Physiol, 27, 1711-1720, 10.1093/treephys/27.12.1711, 2007.

*3) Finally, I think that considering the relevance for the conclusion of the job, the figure of root distribution, currently in the supplemental material (S4), should be included in the main body of the*

*manuscript.*

**Response: Added.** The Figure S4 in previous manuscript has been added as ***Figure 2*** in the revised manuscript in "3.1 Variation in environmental parameters and plant fine root vertical distribution" subsection in "**3 Results**" section as follows: "The *H. rhamnoides* and *P. tomentosa* in pure plantations exhibited different fine root vertical distributions, with more than 40% of fine roots observed in shallow and deep soil layers, respectively (Fig. 2)." (Page 13 Lines 330-332; Page 14 Lines 343-345)

[Figure]

**Figure 2.** Variation in average surface area of fine root at different soil depths for *H. rhamnoides* and *P. tomentosa* in (a) pure and (b) mixed plantations. Error bars indicate the standard deviation (n = 3).

**Minor Comments:**

*1) L137-141: Please add indications of the value of Soil water content at Field capacity and at the permanent wilting point. This would help the interpretation of Figure 1.*

**Response: Suggestions accepted.** The soil water content at field capacity and permanent wilting point have been added in the revised manuscript.

   **Firstly**, the method detect field capacity and permanent wilting point has been added in the "2.1 Study site" subsection in "**2 Materials and methods**" section as follows: "Based on an experiment conducted in July 2018 using the cutting ring (Wu et al., 2016), constant water head (Reynolds et al., 2002), and centrifugation (Qiao et al., 2019) method, the soil bulk density, total porosity, saturated

hydraulic conductivity, field capacity, and permanent wilting point at 0–200 cm soil depth were found to be similar in the three plantations." (Pages 5-6 Lines 134-137)

**Secondly**, the value of field capacity and permanent wilting point has been added in the "2.1 Study site" subsection in "2 Materials and methods" section as follows: "The average field capacity was 0.26 $\pm$ 0.02, 0.25 $\pm$ 0.03, and 0.25 $\pm$ 0.02 m$^3$ m$^{-3}$ for pure *H. rhamnoides*, pure *P. tomentosa*, and mixed plantations, respectively, and corresponding permanent wilting point was 0.06 $\pm$ 0.02, 0.06 $\pm$ 0.01, and 0.06 $\pm$ 0.02 m$^3$ m$^{-3}$." (Page 6 Lines 142-144)

**References:**

Wu, G. L., Yang, Z., Cui, Z., Liu, Y., Fang, N. F., and Shi, Z. H.: Mixed artificial grasslands with more roots improved mine soil infiltration capacity, J Hydrol, 535, 54-60, 2016.

Qiao, J. B., Zhu, Y. J., Jia, X. X., Huang, L. M., and Shao, M. A.: Pedotransfer functions for estimating the field capacity and permanent wilting point in the critical zone of the Loess Plateau, China, Journal of Soils and Sediments, 19, 140–147, 10.1007/s11368-018-2036-x, 2019.

Reynolds, W.D., Elrick, D.E., Youngs, E.G., Booltink, H.W.G., and Bouma, J.: Saturated and field-saturated water flow parameters, in: Methods of soil analysis, edited by: Dane, J.H., Topp, G.C., Soil Science Society of America, Madison, Wisconsin, USA, 797-878, 2002.

*2) L323: you could simplify the phrase stating: ..." in the shallow soil layer, with no significant changes in mixed plantations for H. rhamnoides…"*

**Response: Suggestions accepted.** This sentence has been rewritten in "3.1 Variation in environmental parameters and plant fine root vertical distribution" subsection in "3 Results" section as follows: "In the shallow soil layer, no significant changes in fine root proportion were observed for *H. rhamnoides* in pure and mixed plantations (P > 0.05)." (Page 13 Lines 332-333)

*3) L327: I suggest using only 1 decimal when expressing the percentage: 21.94%    21.9% and so on in the rest of the manuscript (also for CV)*

**Response: Suggestions accepted and Revised.** In response to meaningful suggestion, one decimal was

used when expressing the percentage (for example, Page 6 Line 140; Page 6 Lines 146; Page 13 Lines 324; Page 13 Lines 334;) and CV (for example, Page 13 Line 324; Page 17 Lines 400-401) throughout the revised manuscript.

For example, the corrected for percentage expressing in "3.1 Variation in environmental parameters and plant fine root vertical distribution" subsection in "3 **Results**" section can be observed as follows: "However, the fine root proportion of *P. tomentosa* in the shallow soil layer was significantly increased from 21.9% in pure plantation to 31.3% in the mixed plantation ($P < 0.05$)." (Page 13 Lines 333-335)

For example, the corrected for CV expressing in "3.1 Variation in environmental parameters and plant fine root vertical distribution" subsection in "**3 Results**" section can be observed as follows: "The coefficients of variation (CVs, SD/mean) for SW in the shallow soil layer were 18.2%, 16.7%, and 17.3% in *H. rhamnoides* and *P. tomentosa* pure plantations and the mixed plantation, respectively." (Page 13 Lines 323-325)

*4) L389: what does it mean "positive ψpd"? Not clear*

**Response: Revised.** The "positive $\psi_{pd}$" has been revised to "higher $\Psi_{pd}$" in the revised manuscript in "3.4 Variations in plant leaf water potential" subsection in "3 Results" section as follows: "Compared with *P. tomentosa*, *H. rhamnoides* exhibited significantly higher $\Psi_{pd}$ in the pure plantation, lower $\Psi_m$ in the mixed plantation, and larger $\Psi_{pd}-\Psi_m$ in both plantation types ($P < 0.05$) (Table S6)." (Page 17 Lines 401-403)

*5) L403: the linear correlation for P. tomentosa in pure plantation is assessed on 4 points only (instead of 5) and the p-value is not far from 0.05... I would avoid such a conclusion of the paragraph.*

**Response: Clarified and Revised the Figure and relative sentence.** Thanks for this detailed and meaningful suggestion. The Figure 7 has been revised to clearly exhibit the linear correlation for each species in pure and mixed plantations, and the relative sentence has also been revised in the revised manuscript.

**Firstly**, the linear correlation for *P. tomentosa* in pure plantation is assessed based on 5 rather than 4

points. In original Figure 7(b) (**the blue colour point in the red cycle in Original Figure. 7b as follows**), one point for *P. tomentosa* in pure plantation was covered by the point for *P. tomentosa* in mixed plantation.

[Figure]

**Original Figure 7.** Relationship between average relative response of normalized $F_d$ ($SF_R$) and (a, b) rainwater-recharged soil water uptake proportion (RUP), and (c, d) leaf water potential gradient ($\Psi_{pd}-\Psi_m$) for *H. rhamnoides* and *P. tomentosa* in both plantation types (n = 3)."

In the revised **Figure 7**, we changed the symbol type and rearranged the sequential of the symbol to clearly exhibit the linear correlation for each species in pure and mixed plantations. The revised **Figure 7** can be observed in "3.5 Influence of water sources and $\Psi_{pd}-\Psi_m$ on plant transpiration" subsection in "**3 Results**" section as follows: "

[Figure]

**Figure 7.** Relationship between average relative response of normalized $F_d$ ($SF_R$) and (a, b) rainwater-recharged soil water uptake proportion (RUP), and between $SF_R$ and (c, d) leaf water potential gradient ($\Psi_{pd}-\Psi_m$) for *H. rhamnoides* and *P. tomentosa* in both plantation types (n = 3)." (Page 19 Lines 416-419)

**Secondly,** because the *p*-value (p=0.07) was close to 0.05 for the linear correlation for *P. tomentosa* in pure plantation, the relative sentence has also been rewritten in "3.5 Influence of water sources and $\Psi_{pd}-\Psi_m$ on plant transpiration" subsection in "**3 Results**" section as follows: "However, a significant relationship between $SF_R$ and RUP was observed for *P. tomentosa* in the mixed plantation (P < 0.05) (Fig. 7)." (Page 18 Lines 414-415)

*6) L422-423: "exhibiting plasticity in water sources" is not clear. Try to reformulate it in a clearer way (i.e. plant water sources in relation to soil depth"). Plasticity is a rather qualitative description*

**Response: Rewritten.** In the revised sentence, we directly described that these plants uptake water from different soil layers after different rainfall pulses. The revised sentence can be observed in "4.1 RRS uptake enhances plant transpiration for *H. rhamnoides* but not *P. tomentosa* in pure plantations"

subsection in "**4 Discussion**" section as follows: "Similar to *Salix psammophila* and *Caragana korshinskii* in the studied region (Zhao et al., 2021), both *H. rhamnoides* and *P. tomentosa* take up water from different soil layers under varied soil water conditions following rainfall pulses in pure plantations (Fig. 5)." (Page 20 Lines 433-435)

*7) L424: Why is it so obvious? This paragraph (starting from "in pure…" and ending with "… of no rainfall" need to be reformulated*

**Response: Rewritten.** In response to this meaningful suggestion, this sentence has been rewritten in "4.1 RRS uptake enhances plant transpiration for *H. rhamnoides* but not *P. tomentosa* in pure plantations" subsection in "**4 Discussion**" section as follows: "In pure plantations, large water uptake proportion from the deep soil layer after 3.4 mm of rainfall for *H. rhamnoides* (52.5 $\pm$ 8.7%) and *P. tomentosa* (64.1 $\pm$ 5.1%) (Fig. 5), suggested that this rainfall amount did not relieve the drought caused by 36 days (DOY 157–192) of no rainfall." (Page 20 Lines 435-438)

In the revised manuscript, the "obviously lower SWC at all soil depths (Fig. 1)" has been deleted to correctly express the result, according to this meaningful suggestion.

*8) L489: Again positive ψpd , please check this sentence*

**Response: Revised.** The "positive $\psi_{pd}$" has been revised to "higher $\Psi_{pd}$" in the revised manuscript in "4.2 RRS uptake enhances plant transpiration for coexisting species in mixed plantation" subsection in "4 Discussion" section as follows: "Although mixed afforestation did not significantly alter the $\Psi_{pd}$ and $\Psi_m$ for *H. rhamnoides* and *P. tomentosa*, respectively, significantly lower $\Psi_m$ and higher $\Psi_{pd}$ were observed for corresponding species ($P < 0.01$) (Table S6)." (Page 23 Lines 505-507).

In addition, the expression of "positive $\psi_{pd}$" has been corrected to "higher $\Psi_{pd}$" throughout the revised manuscript (such as in Page 17 Line 402).